# Linking spatial patterns of terrestrial herbivore community structure to trophic interactions

**Jakub Witold Bubnicki\*, Marcin Churski, Krzysztof Schmidt, Tom A Diserens, Dries PJ Kuijper**

Mammal Research Institute, Polish Academy of Sciences, Białowieża, Poland

**Abstract** Large herbivores influence ecosystem functioning via their effects on vegetation at different spatial scales. It is often overlooked that the spatial distribution of large herbivores results from their responses to interacting top-down and bottom-up ecological gradients that create landscape-scale variation in the structure of the entire community. We studied the complexity of these cascading interactions using high-resolution camera trapping and remote sensing data in the best-preserved European lowland forest, Białowieża Forest, Poland. We showed that the variation in spatial distribution of an entire community of large herbivores is explained by species-specific responses to both environmental bottom-up and biotic top-down factors in combination with human-induced (cascading) effects. We decomposed the spatial variation in herbivore community structure and identified functionally distinct landscape-scale herbivory regimes ('herbiscapes'), which are predicted to occur in a variety of ecosystems and could be an important mechanism creating spatial variation in herbivory maintaining vegetation heterogeneity.

DOI: https://doi.org/10.7554/eLife.44937.001

## Introduction

Spatial patterns in species distribution, abundance and community composition are manifestations of underlying ecological mechanisms operating at a range of spatial scales (*Levin, 1992*). These patterns emerge from dynamic interactions between environmental bottom-up, biotic top-down and biotic parallel factors in combination with stochastic effects. Although the role of bottom-up factors in shaping species distributions has been intensively studied in recent decades (*Elith and Leathwick, 2009*; *Guisan and Thuiller, 2005*), we still know little about the importance of the biotic factors underlying most spatial patterns. Especially, the role of species interactions, within and across trophic levels, including those involving humans, remain largely unexplored (*Darimont et al., 2015*; *Schmitz et al., 2017*; *Wiens, 2011*; *Worm and Paine, 2016*). It has recently been proposed that community ecology should be 'rediscovered' as an integrative study of species interactions and spatial distributions (*Schmitz et al., 2017*), while accounting for direct and indirect anthropogenic effects on species distributions and behavior (*Berger, 2007*; *Worm and Paine, 2016*).

Large mammalian herbivores influence terrestrial ecosystem structure and functioning (*Gordon et al., 2004*; *Hobbs, 1996*; *Schmitz, 2008*) via their direct effects on vegetation structure (*Charles-Dominique et al., 2016*; *Churski et al., 2017*; *Didion et al., 2009*; *Hempson et al., 2015*; *Kuijper et al., 2010a*) and indirect effects on nutrient cycling (*Murray et al., 2013*). In this way, herbivory influences vegetation at large spatial scales, from the local landscape up to the biome level (*Moncrieff et al., 2016*; *Woodward et al., 2004*), and can lead to herbivory-mediated cascading effects on other trophic levels (*Gordon et al., 2004*; *Palmer et al., 2015*; *Schmitz, 2008*). There is often strong spatial variation in herbivory impact, resulting from the space use of different functional groups of herbivores. This spatial variation is driven primarily by the interactive effects of abiotic

**\*For correspondence:**
kbubnicki@ibs.bialowieza.pl

**Competing interests:** The authors declare that no competing interests exist.

**eLife digest** In almost every ecosystem on Earth, communities of herbivores are kept in check by both predators and the availability of the plants they eat. As herbivores move in response to these pressures, they shape local plant communities and impact vegetation across entire landscapes. Yet the role of large plant-eating mammals in structuring ecosystems is often overlooked. Indeed, most research on this topic has looked at African ecosystems, like open savannahs, and fewer researchers have studied temperate forests like those found across Europe, Asia and North America.

Bubnicki et al. have now examined factors influencing the distribution of five large herbivore species and resulting plant communities in Białowieża Forest in eastern Poland, the best-preserved European lowland forest. Their method involved measuring the cascading interactions of plants and animals in the forest using cameras set at nearly 900 locations, satellite images and other remote sensing technologies, and on-the-ground surveys. Added to this were patterns of human activity inferred from the available data for the study area. This approach allowed Bubnicki et al. to explore how humans are influencing the forest ecosystem, too.

The analysis revealed that humans are the main factor influencing the movements of carnivorous predators in Białowieża Forest, but not the herbivores directly. Wolves and lynxes avoided areas heavily used by humans whereas large herbivores responded primarily to different environmental factors. Wild boar and bison are influenced by the availability of plant food and preferred habitat for foraging; moose and roe deer by the features of the landscape, like elevation or openness. The red deer was the only large herbivore species whose distribution was strongly linked to that of its main predator, the wolf.

From this, Bubnicki et al. identified distinct areas in the forest which have emerged from the interactions at play, describing these areas as 'herbiscapes' for the herbivores that shaped them. These findings provide new understanding of the complex ecological processes shaping the Białowieża Forest and serve as a model to help understand other ecosystems around the world. The knowledge will also contribute to the ongoing management and conservation of this UNESCO World Heritage Area.
DOI: https://doi.org/10.7554/eLife.44937.002

factors, disturbances, forage quality and quantity in combination with life-history traits, such as herbivore body mass (*Anderson et al., 2016*; *Cromsigt et al., 2009*; *Hempson et al., 2015*; *Hopcraft et al., 2010*; *Ogutu et al., 2010*). However, the actual distribution of many herbivores often differs from the expected distribution derived purely from interactions with bottom-up factors. This discrepancy results from herbivores also responding to landscape gradients induced by biotic top-down interactions (*Anderson et al., 2010*; *Hopcraft et al., 2010*; *Kauffman et al., 2007*). In effect, this landscape of interacting ecological gradients, both bottom-up and top-down, creates spatial heterogeneity in the availability and suitability of habitats for different large herbivore species within a community (*Cromsigt et al., 2009*; *Fryxell, 1991*; *Hopcraft et al., 2010*). Thus, to assess the ecosystem-level impact of large herbivore communities requires full understanding of the factors driving spatial heterogeneity in their community structure across a landscape (*Gordon et al., 2004*; *Weisberg and Bugmann, 2003*).

Recently, there has been much attention given to the role of large carnivores in structuring ecosystems via their effects on herbivore communities (*Estes et al., 2011*; *Ripple et al., 2014*; *Terborgh et al., 2006*). In addition to their density-mediated effects (i.e. impact on prey population size), behaviorally mediated effects (i.e. impact on prey behavior) on prey species are a crucial mechanism explaining the trophic cascades driven by large carnivores (sensu *Ripple et al., 2016*). Prey species react to the presence of large carnivores by adjusting their spatio-temporal patterns of landscape use (*Creel et al., 2005*; *Kohl et al., 2018*; *Laundré et al., 2001*; *Valeix et al., 2009*). These spatial interactions between trophic levels are usually context-dependent and are shaped by the biophysical characteristics of a landscape (*Kauffman et al., 2007*; *Schmitz et al., 2017*; *Valeix et al., 2009*). Many studies have addressed how carnivores affect their prey species, but these have generally used a single carnivore - single prey species approach, whereas many ecosystems host multiple carnivore and multiple prey species. In such systems, different carnivore species can create

contrasting risk effects (*Creel et al., 2017*; *Preisser et al., 2007*; *Thaker et al., 2011*). Moreover, in multi-species communities, some prey species perceive more risk than others from a carnivore species (*Anderson et al., 2016*; *Laundré et al., 2001*; *Valeix et al., 2009*).

This suggests that spatial distributions of predation-sensitive prey species may be mainly driven by carnivore top-down effects, whereas distributions of predation-insensitive prey or non-target species by gradients in resources availability (*Hopcraft et al., 2010*). When a community of prey species consists of ecologically or functionally similar species, changes in the abundance and distribution of one species may be buffered by another species (*Ford et al., 2015*; *Rosenfeld, 2002*). These so-called redundancy effects, can prevent apex predators from creating trophic cascading effects when taking the response of the entire herbivore community into account, despite them significantly impacting one or more prey species (*Ford et al., 2015*; *Liu et al., 2016*).

There is a growing awareness that including humans in community studies is critical for improving our understanding of ecosystem functioning (*Darimont et al., 2015*; *Worm and Paine, 2016*) and for predicting species distributions in increasingly anthropogenic environments. Due to the recovery of some large carnivore populations and expansion of human populations, carnivores are increasingly sharing landscapes with humans world-wide (*Carter and Linnell, 2016*; *Chapron et al., 2014*). Humans are also increasingly being considered a coherent part of complex trophic interaction chains (*Darimont et al., 2015*; *Strong and Frank, 2010*; *Kuijper et al., 2016*). The resulting complex, cascading interactions urgently need to be considered when studying the spatial distributions of herbivores, their effects within the landscape and the functional role large carnivores can play in landscapes that are becoming increasingly anthropogenic (*Kuijper et al., 2016*).

In this study, we investigated how the interactive effects of bottom-up and natural top-down factors (two large carnivore species and humans), determine the landscape distribution and community composition of five native ungulate species in Białowieża Forest (BF, Poland; *Figure 1*). BF is regarded to be one of the best preserved temperate European lowland forest systems and is inhabited by a natural community of large mammals (*Jędrzejewska and Jędrzejewski, 1998*). In addition, BF is also embedded within an anthropogenic landscape typical for many terrestrial systems. We hypothesized that spatial variation in the composition of the large herbivore community is explained by the interactive effects of species-specific responses to major environmental and risk gradients operating at the landscape level. We aimed to answer the following questions: 1) Do large carnivores have species-specific effects on the distributions of ungulates in our multiple predator-prey system?, 2) How does human activity mediate predator-prey interactions at the landscape scale?, 3) Does this lead to ecologically distinct herbivory regimes (sensu *Hempson et al., 2015*) with differential vegetation impact at the landscape scale? Using detailed data on species distributions (894 camera trap locations), landscape structure (high-resolution GIS and remote sensing data) and detailed woody vegetation surveys (385 study plots) along with a novel spatially-explicit hierarchical modelling approach we decomposed the spatial variation in herbivore community structure into ecologically distinct landscape-scale herbivory regimes. With data from a complex, multi-species and human-influenced system, we aimed to exemplify the functional role that large carnivores and their herbivorous prey can play in increasingly human-affected ecosystems.

## Materials and methods

### Study area

The study was carried out in Białowieża Forest (BF) in Poland (c. 580 km2; *Figure 1*). This harbors a natural assemblage of central European ungulate species, with red deer (*Cervus elaphus*) being the most abundant (6.0 individuals/km2), followed by wild boar (*Sus scrofa*; 5.4/km2) and roe deer (*Capreolus capreolus*; 2.0/km2); and European bison (*Bison bonasus*; 0.5/km2) and moose as the rarest ungulates (*Alces alces*; 0.08/km2) (*Borowik et al., 2016*). Two large carnivores occur in BF: the Eurasian lynx (*Lynx lynx*; c. 15 individuals) and wolf (*Canis lupus*; 4 packs of 7–12 individuals) (*Jędrzejewski et al., 2002*; *Schmidt and Kuijper, 2015*). Part of BF, the core of Białowieża National Park (BNP; c. 47 km2), has been strictly protected since 1921. Since this time, human activities such as hunting and forestry have been banned. In 1996 BNP was enlarged to c. 100 km$^2$. Outside the national park, the forest is managed by the State Forest Holding, hence timber production takes place and ungulate hunting is allowed, but here also exists a network of nature reserves (c. 130 km$^2$,

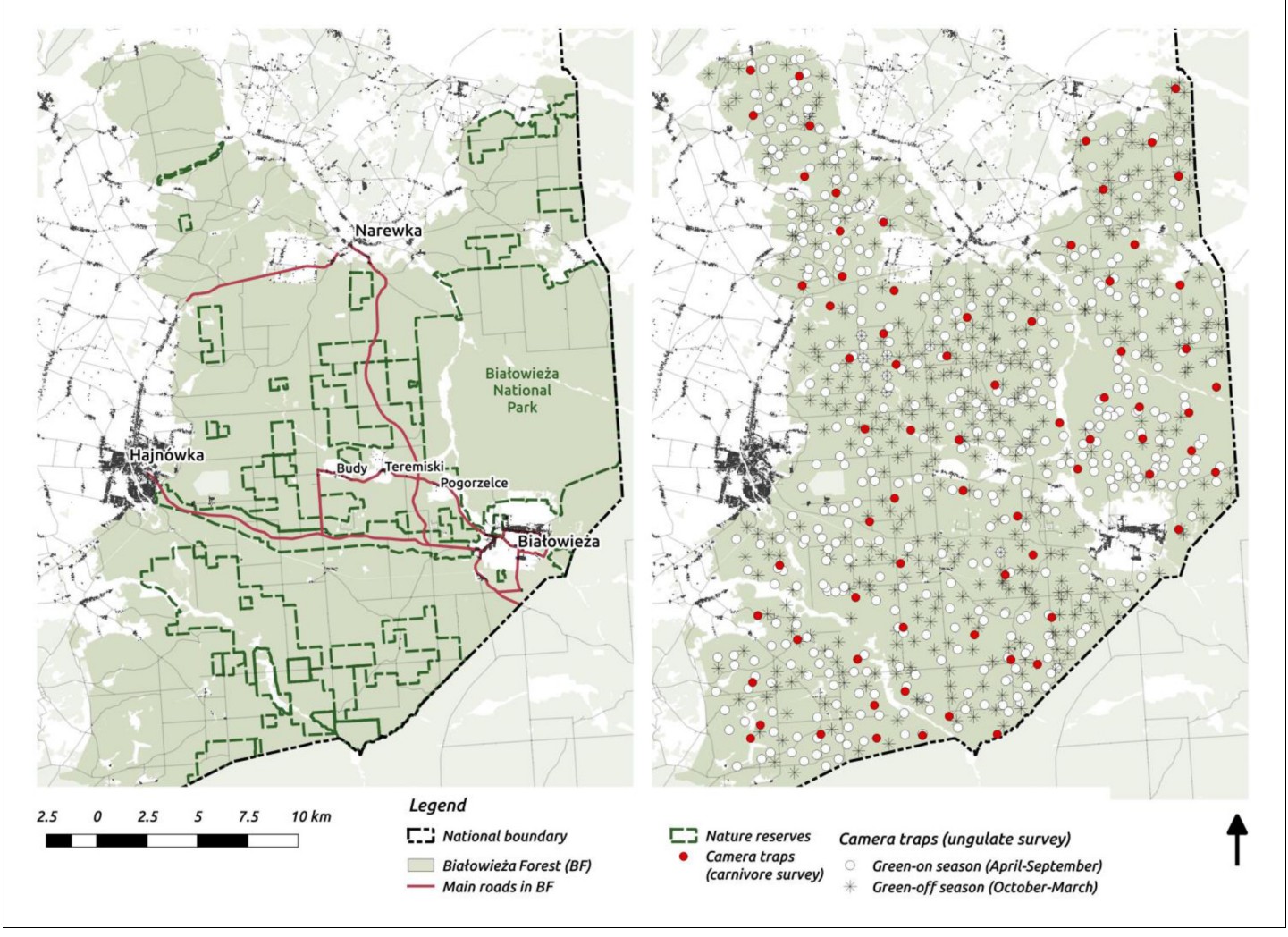

**Figure 1.** Maps of the study area (left) and camera trapping sampling design (right). On the study area map we marked the major settlements connected by public roads open for cars.

DOI: https://doi.org/10.7554/eLife.44937.003

see *Figure 1*). For a more detailed description of this area see *Faliński and Falińska (1986)* and *Jędrzejewska and Jędrzejewski (1998)*. From a landscape perspective, it is important that the boundaries of BF ecosystem are well defined. From the west, BF is surrounded by agricultural fields, and from the north and south by a mosaic of agricultural and fragmented forest landscapes. In the east, a tall wire-fence along the border with Belarus (built in 1981) prevents movements by ungulates. These conditions create an opportunity to study the spatial distribution of a whole community of ungulates in a spatially restricted, complex ecosystem with natural predators present and varying levels of human management and impact.

## Camera trapping

In contrast to telemetry, which allows the study of both predator and prey movement patterns by placing sensors on individual animals, we used point-based camera trapping to record and reconstruct the distributional patterns of large carnivores and ungulates. We sampled the continuous BF landscape with camera traps placed randomly with respect to species movements and hypothetical mechanisms driving their distributions (habitat structure, humans, predator-prey interactions). We used an intensive, large-scale and high-resolution camera trap network covering the entire study area to collect detailed, spatially-explicit information on species distributions. We argue that camera

trapping is the most objective and efficient method for collecting this type of spatially-explicit community data. This kind of study would be inherently impossible to do with telemetry: to record the spatial patterns we were looking for, the entire populations of each species had to be observed simultaneously. Moreover, because of logistical and financial limitations and ethical issues (related to live-trapping of protected species in Europe) it would be practically impossible to obtain sufficient data for all the studied species of large herbivores and carnivores using telemetry. We used digital trail cameras (Ecotone SGN-5210A) triggered by passive infrared sensors with a detection angle of c. 35˚ and range of c. 20 m. After detection, with a time lag of c. 1 s, a photograph was taken and the camera recorded a 60 s video. When an animal stayed, this procedure was repeated without trigger delay. During low-light conditions, cameras switched to a stealth infrared mode. Cameras were attached to a tree at a height of c. 1 m at locations with a clear view of at least 20 m. Whenever possible, we randomly chose an acceptable place to mount a camera as close as possible to the locations given by coordinates pre-computed prior to the field work.

## Ungulate survey

We quantified the spatial distribution of the ungulate community by using a spatially extensive, high-resolution network of camera traps. The data were collected over 2 years (May 2012 - May 2014) of intensive camera trapping in all seasons, except for days with the strongest winter conditions (snowing heavily or temperatures below −20°C). We conducted 34 trapping sessions, each lasting 14 days on average. During each session between 30 to 40 camera traps were pseudo-randomly deployed in different parts of the forest and within a minimum distance of 100 m from the nearest roads (both paved and unpaved), large clearings and settlements. Additionally, we kept a minimum distance of 100 m between all camera locations. The coordinates for all locations were pre-computed prior to the field work in QGIS software (*QGIS Development Team, 2017*). In total, we collected data at >1 k sites. However, because of logistical errors, camera failures, stealing of equipment and heavy snowing in winter (leading to blocked view and uninterpretable gaps in data) we had to exclude >100 sites (~10%) from further analysis. Finally, we used data collected at 894 sites, covering the whole BF landscape (*Figure 1*).

## Carnivore survey

Large carnivores generally tend to have a higher detection probability at forest roads and trails (*Cusack et al., 2015*) because they often prefer to move along linear landscape structures (*Zimmermann et al., 2014*) and/or to increase the probability of encountering prey (*Whittington et al., 2011*). This specific space use resulted in a very low trapping rates of both carnivores (wolf and lynx) during the ungulate survey (*Appendix 1—table 1*), when camera traps were placed randomly in the forest. Hence, to better quantify the space use of the two large carnivores we ran an additional camera trapping survey in September-October 2015. We deployed 73 camera traps on the sides of forest roads across the whole landscape for one month (*Figure 1*). The core areas of wolf pack territories are related to the locations of breeding dens (*Jędrzejewski et al., 2001*). During the reproductive season (spring-summer) the spatial distribution of a wolf pack is restricted to their core area, whereas outside this period they regularly return to it (*Jędrzejewski et al., 2001*). In August-September, pups begin to travel with other pack members and move more widely through their territory, returning to the core of their territory on a regular basis (*Jędrzejewski et al., 2001*). Lynx have a similar, typical pattern of movements whereby females restrict their movements in May-July, while tending their kittens (*Schmidt, 1998*). Therefore, late summer-autumn is the best period to quantify the space use of both carnivores at the landscape scale.

The ungulate and carnivore surveys were conducted during different time periods, with different sampling intensities (May 2012 to May 2014 versus September to October 2015, respectively) and using different camera placement strategies. To check if our results were not spatio-temporally confounded because of these differences, we ran the same model for all ungulate species using only a subset of the camera trap data covering a 3-month period (August - October) matching the period of the carnivore survey as closely as possible (the models did not converge for data from only one or two months). The obtained results were similar to those based on the full dataset (Appendix 1: Figs. S36-S37); we thus chose to use the full dataset, which provided a larger sample size and better

spatial coverage of the study area, both of which are needed for making robust inferences using complex hierarchical spatial models such as ours (see *Statistical model* section). Moreover, between-season variation in the trapping rate of ungulates was directly accounted for in the model. Lastly, all the studied ungulate species are non-migratory; in other words, there is no large (landscape)-scale seasonal movement of ungulates in BF (*Jędrzejewska and Jędrzejewski, 1998*; *Kamler et al., 2008*; *Podgórski et al., 2013*). The winter distribution of European bison is to a large extent driven by the location of supplementary feeding sites. This results in concentration of bison at these locations during the winter season.

## Data processing

After downloading camera trap data, both ungulates and wolf datasets were organized and classified using TRAPPER software (*Bubnicki et al., 2016*). Species, sex, age and group size were determined for every recorded image or video containing an observation of focal species. We defined the independence interval between successive captures (i.e. event; see *Meek et al., 2014*) as five minutes.

## Statistical model

### General description

We developed a hierarchical multi-scale spatial model to quantify landscape use by large carnivores and ungulates. Our model was built upon the previous work of *Royle (2004)* and *Royle et al. (2007)*, who described a class of Binomial-Poisson N-mixture models for spatially replicated counts collected during multiple (discrete) surveys at each site (*Royle, 2004*) and further applied these models in a spatially explicit context (*Royle et al., 2007*). Following the later work of *Guillera-Arroita et al. (2011)*; *Guillera-Arroita et al., 2012*), we expanded this approach to a continuous case where counts are described using the Poisson instead of binomial distribution. This approach is more suitable for camera traps and unmarked populations (as in our case), as multiple detections of the same individuals are allowed, accounting within a single modeling framework for both false-negatives ('imperfect detection') and false-positives ('double counts').

Moreover, the Poisson distribution intensity parameter ($\gamma$ in the next section), when multiplied by the number of (arbitrarily defined) sampling occasions, can be interpreted as the trapping rate, and also corrects for unequal effort amongst sampling locations (offset). Both the Binomial-Poisson N-mixture model and its continuous variations allow for estimating species abundances at monitored sites ($N_j$ in the next section), assuming sampling locations are independent and that the observed system is closed (i.e. no changes in abundance at the site during repeated surveys). In other words, site abundances are treated as independent random variables distributed according to some mixing distribution, for example Poisson or Negative Binomial (*Royle, 2004*).

Usually, to ensure independence between sampling locations (i.e. no shared individuals) a distance larger than the average home range of focal species is preferred. This does not apply to our high-resolution camera trapping study as both the average distance between neighbouring sampling locations and size of landscape grid cells chosen for predictions (500 x 500 m) are much smaller than home and daily ranges of all ungulates and both carnivores. This specific sampling strategy was designed to capture and model the fine-scale, continuous variation in the use of the landscape by all studied species. Thus, following *Royle (2004)* we view the site-specific abundance as a random effect and relax the assumption of sampling site independence by interpreting $N_j$ as the relative density that is the number of individuals *using* a given landscape grid cell during a sampling period rather than an absolute value of abundance. A similar interpretation has been used in many occupancy studies (*Cusack et al., 2017*; *Efford and Dawson, 2012*; *Latif et al., 2016*), where the probability of site occupancy has been interpreted as the probability of site use. We used the Negative Binomial distribution as the prior distribution for $N$ to accommodate extra-Poisson variation not explained by the included covariates and spatial random effects (see Formal description below). Lastly, we assumed the system to be closed (i.e. no significant changes in ungulate and carnivore distributions and demographies) during the sampling period (2 years for ungulates and 1 month for carnivores). This is a reasonable assumption since the landscape surrounding BF creates natural boundaries (see description of the study area) and the hunting intensity is relatively low and constant over the years (*Zbyryt et al., 2018*).

## Formal description

Let us consider a landscape divided into $j = 1, 2, ..., G$ grid cells of equal sizes and $i = 1, 2, ..., S$ sampling locations monitored with camera traps for $d_i$ days each, where each grid cell can contain zero, one or multiple camera trap locations. This results in a multi-scale design where smaller subunits (camera trap sites) are nested within larger units (grid cells) (*Kery and Royle, 2016*). Counts, $y_i$, are Poisson random variables given by

$$y_i | N_{j[i]} \sim Poisson\left(N_{j[i]} \gamma_i d_i\right) \tag{1}$$

where the intensity parameter is a product of $N_{j[i]}$, the number of individuals using a landscape grid cell $j$ during a study period, $\gamma_i$, the expected detection (trapping) rate per sampling occasion (here 1 day) and $d_i$, the number of sampling occasions (days) during a survey at a camera trap location $i$. In the state-space formulation of our hierarchical model $N_j$ is the Negative Binomial distributed latent variable

$$N_j \sim NegBin\left(\lambda_j, \phi\right) \tag{2}$$

with the parameter $\lambda_j$ being the expected number of individuals using a landscape grid cell $j$ and $\phi$ the dispersion parameter. Both parameters, $\lambda_j$ and $\gamma_i$ can depend on covariates describing for example environmental gradients (at different scales), biotic interactions and seasonal differences in species activity level. This variation can be modelled using standard generalized linear regression techniques using log-link functions

$$log(\gamma_i) = x'_{\gamma i} \beta_\gamma + \epsilon_i \tag{3}$$

$$log(\lambda_j) = x'_{\lambda j} \beta_\lambda \tag{4}$$

where $x'_{\gamma \cdot}$ and $x'_{\lambda \cdot}$ are transposed rows of design matrices $X_\gamma$ and $X_\lambda$, respectively, $\beta_\cdot$ are vectors of linear predictor coefficients and $\epsilon_i$ are identically and independently distributed (*iid*) camera trap measurement errors. It is necessary to note that, while the linear predictor of $\lambda_j$ explains the variation in data arising from the ecological processes, the linear predictor of $\gamma_i$ (detection/trapping rate) deals with both the ecological (e.g. habitat selection, movement, seasonal activity levels) and observational processes (e.g. detection issues, camera failures). In the latter case, informative and ecologically meaningful covariates are needed to disentangle these otherwise confounded sources of variation. In order to directly account for the potential spatial dependence between landscape grid cells (i.e. to capture all the spatial variation not explained by the included covariates) we introduced spatial random effects ("spatial residuals") into the linear predictor of $\lambda_j$:

$$log(\lambda_j) = x'_{\lambda j} \beta_\lambda + \omega_j \tag{5}$$

where $\omega = (\omega_1, ..., \omega_G)$ is the realization of a Gaussian spatial process on a discrete (gridded) spatial domain. Specifically, we implemented a Restricted Spatial Regression model (RSR) (*Hughes and Haran, 2013*; *Johnson et al., 2013*), which is the restricted version of the intrinsic Conditional Autoregressive (iCAR) model. The RSR model was developed to solve the issue of confounding between the spatial process and the fixed-effects covariates in spatial regression models. The byproduct of this solution is its computational efficiency, as the RSR model is a reduced dimension model.

## Specification

The model as described above was fitted to our camera trapping data of the wolf and lynx and each of the five ungulate species. To model the intensity of landscape use ($\lambda_j$ in *Equation 5*), we overlaid a grid of 2303 cells 500 m per side (25 ha each) over the study area. Based on this grid we compiled a set of spatial (raster) covariates describing the ecologically relevant (sensu *Elith and Leathwick, 2009*) environmental and human-induced gradients (*Figure 2*). For GIS data processing, we used QGIS (QGIS Development *QGIS Development Team, 2017*) and GRASS GIS (*Neteler et al., 2012*) open source software. We standardized (scaled) all covariates, by subtracting the mean and dividing the result by the SD of the original variables. Finally, we ensured that the Pearson correlation

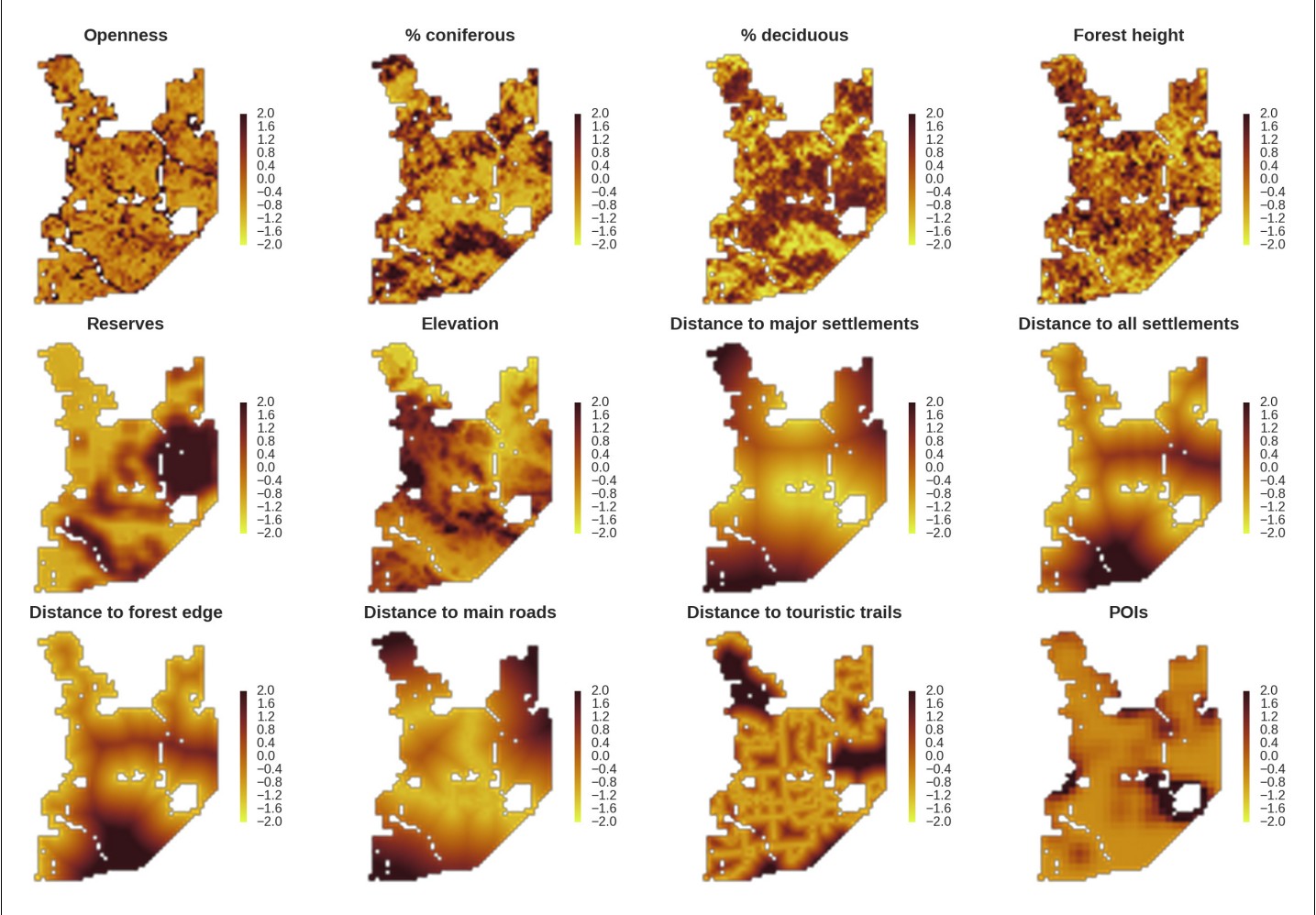

**Figure 2.** Maps of candidate landscape-scale covariates for the ungulate and carnivore models representing environmental and human-related landscape gradients. All covariates were scaled and zero-centered prior to modelling. POIs – density of tourist infrastructure. When selecting covariates for the final models we ensured that the Pearson correlation value for all pairs of included covariates was lower than 0.7 (*Appendix 1—figure 1*).
DOI: https://doi.org/10.7554/eLife.44937.004

coefficient for all pairs of included covariates was lower than 0.7 (*Appendix 1—figure 1*). For more details about all spatial (raster) covariates and the processing of GIS and remote sensing data see Appendix 1.

## Specification - wolf and lynx model

Based on existing knowledge (*Kuijper et al., 2015*; *Schmidt et al., 2009*; *Theuerkauf et al., 2003*) both carnivore species in the Białowieża forest utilize the entire landscape, although with clear spatial patterns in the intensity of use. Previous work in this study area has shown that this is primarily determined by human related factors (*Theuerkauf et al., 2003*). In BF tourist traffic concentrates mainly within the central parts of the forest where roads (open for the public and cars) connect the three major settlements in the area, that is Hajnówka, Białowieża and Narewka (see *Figure 1* and *Appendix 1—figure 3*). For the reasons above, the following raster layers were chosen as landscape covariates likely to influence carnivore space-use: distance to major settlements, distance to touristic trails, density of touristic infrastructure (POIs), density of protected areas (BNP and nature reserves) and elevation (*Figure 2*). Elevation was included as in this flat landscape the lowest, often swampy areas are the least accessible for humans and could therefore be preferred by the wolf and lynx. We interpreted the parameter $\lambda$ as large carnivore space use intensity and included rasters with predicted values of $\lambda_j$ for both species as covariates in all models for ungulates. We assumed that from

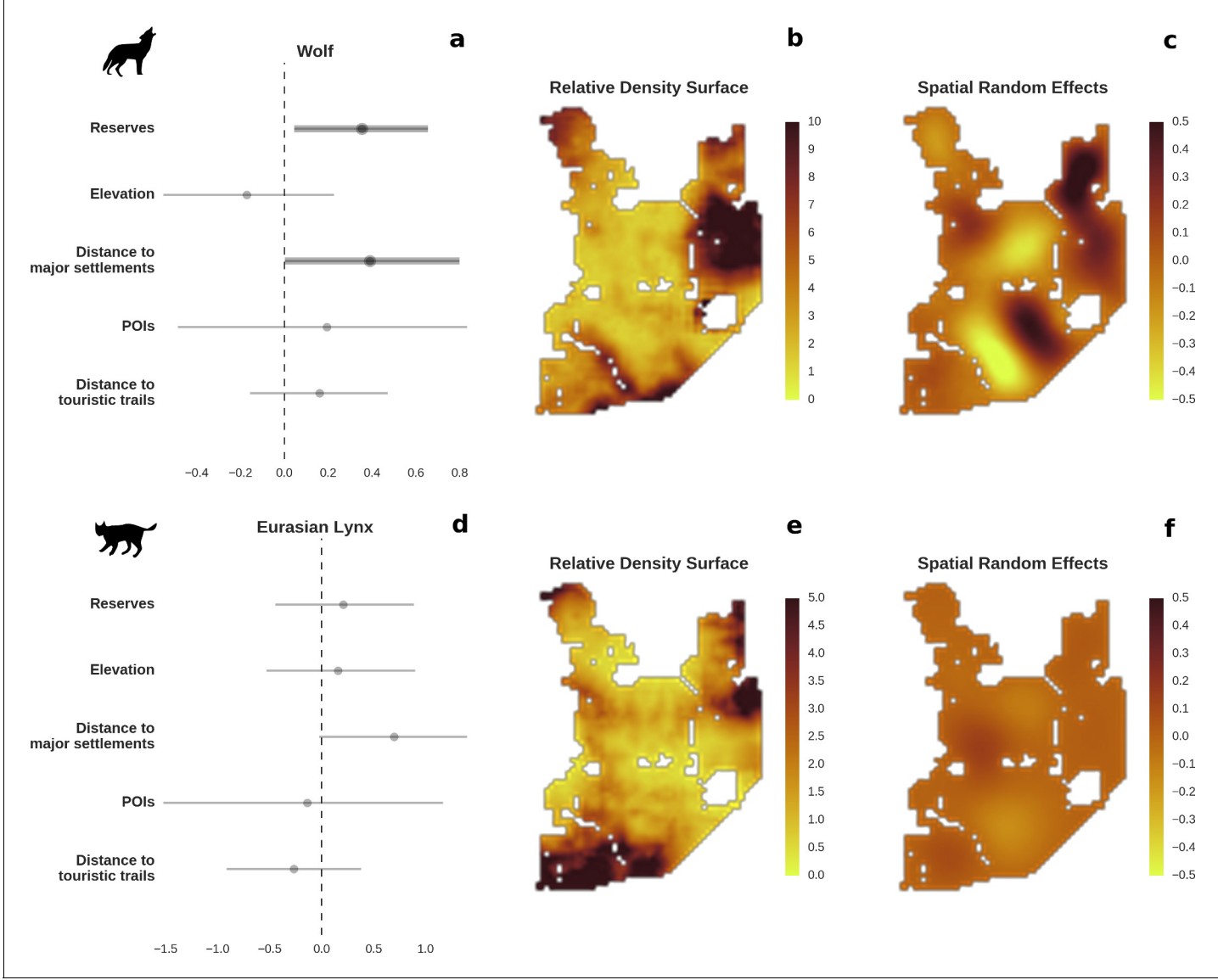

**Figure 3.** Spatial distribution of large carnivores is determined by human activity. (**a, d**) Estimated parameters for the wolf and lynx models, (**b, e**) maps of the predicted variation in their landscape use (relative density surfaces) and (**c, f**) fitted spatial random effects (SRE). Specifically, panels (**b**) and (**e**) present the spatial predictions for the parameter $\lambda$, which is the expected number of individuals using a given landscape grid cell (25 ha pixel) during the sampling period (see the general and formal description of the model in the Materials and methods section). The fitted values of SRE are deviations from 0 at log-scale. The SRE captured all spatial variation not explained by the covariates included in the model, indicating parts of the landscape for which higher ("hot-spots") or lower ("cold-spots") activities of species were observed in the data than was predicted by the "fixed" part of the model. On panels (**a**) and (**b**) we additionally marked in bold the credibility intervals (quantile based) at which the estimated parameter values differed from 0 (for the level 2.5–97.5%, there is a 0.95 probability that the true parameter value lies within this range). The full posterior distribution for each parameter, together with its numerical description can be found in *Appendix 1—figures 26–27*).
DOI: https://doi.org/10.7554/eLife.44937.005

a prey perspective, $\lambda_j$ is proportional to the predator encounter rate, hence it quantifies potential risk as perceived by ungulates.

## Specification - ungulate model

To explain the variation in the landscape-scale distribution of ungulates, we considered three environmental gradients primarily related to major biophysical properties of the forest environment that are known to affect space use of ungulates at multiple scales: percentage of landscape openness,

tree stand canopy height and percentage share of coniferous species (*Churski et al., 2017*; *Jedrzejewska et al., 1994*; *Kuijper et al., 2009*; *Kuijper et al., 2010a*; *Figure 2*). For landscape openness and percentage share of coniferous species, we additionally included their quadratic effects, allowing for the existence of an optimum value for each variable (e.g. species preference for a mixed forest or for intermediate levels of canopy closure). The other covariates included were potential predation risk variables (space use of large carnivores) and two human-related landscape gradients, namely distance to all settlements and density of protected areas (BNP and nature reserves; *Figure 2*). Additionally, we computed the same set of environmental covariates but with a higher resolution (100 m) and used them to model variation in detection rate ($\gamma_i$ in *Equation 4*) at camera trap sites. Here, we assumed that a part of this variation comes from a resource selection process operating at a scale smaller than the landscape unit we defined (see e.g. *Johnson, 1980*), influencing at-site detection rates and observed counts in the end. Another source of variation is species movement behavior and activity level, which can both change between seasons. To control for these temporal effects, we considered a quadratic function of temperature and snow cover as covariates for detection rate.

## Implementation

The models were implemented within a Bayesian framework in Python using PyMC 2.3.6 software (*Patil et al., 2010*). We used Markov chain Monte Carlo (MCMC) for inference and sampled from the posterior distributions with Metropolis-Hastings and Adaptive Metropolis step methods, both available in PyMC. To speed up the model and improve the MCMC convergence, we marginalized out the latent variable $N$ and implemented the integrated likelihood function (*Guillera-Arroita et al., 2012*; *Royle, 2004* ). To make the integration over $N$ values finite, we assumed 100 as the maximal possible number of individuals using a single grid cell $j$ (25 ha). We defined the priors for the linear predictor coefficients $\beta_.$ as diffuse normal priors $N(0, 10^{-3})$. The priors for the measurement errors $\epsilon_i$ were given by $N(0, 1/\sigma^2)$ with the hyper-parameter $\sigma \sim U(0, 100)$ (*Gelman and Hill, 2006*). We followed *Royle et al. (2007)* and we chose the gamma distributed prior $G(0.1, 0.1)$ for the precision parameter of the RSR model.

The reason for not choosing a vague prior for this parameter (as e.g. in *Johnson et al., 2013*) was that we expected the spatial covariates included in the models would not account for all the spatial dependence alone. Part of this (unexplained) variation is likely related to species movement behavior occurring at multiple spatial scales. It is also commonly known that the variance components are poorly identified in these types of models (*Royle et al., 2007*). To obtain posterior distributions of parameters, we ran a MCMC sampler with three chains for 500,000 iterations each (removing the first 400.000 as a burn-in phase of the sampling process) and with the thinning parameter set to 20 to avoid autocorrelation between samples. The convergence was assessed through visual inspection of MCMC trace plots and Gelman–Rubin diagnostics provided by the PyMC software. We evaluated the fit of the models through visual inspection of standard model diagnostics plots (see *Appendix 1—figures 28–35*). We used posterior predictive distribution and Bayesian 'p-value' to assess the goodness of fit of each model (*Kery and Royle, 2016*). The source code of our models is available at https://github.com/mripasteam/herbiscapes/ (*Bubnicki et al., 2019*; copy archived at https://github.com/elifesciences-publications/herbiscapes).

## Ungulate community-level analysis

We explored the distribution of ungulates across the landscape, as predicted by species-specific models, in the context of the community. First, the spatial overlap between species was evaluated by means of pairwise Pearson correlations of their relative density surfaces. Next, we converted ungulate relative densities to herbivore biomass using the following average weights per species: red deer female 90 kg, red deer male 150 kg, roe deer 20 kg, wild boar 80 kg, moose 200 kg and European bison 400 kg (*Borowik et al., 2016*; *Jedrzejewska et al., 1994*). Total biomass was estimated as the sum of each species' biomass for each landscape cell (25 ha pixel) in our prediction grid. We further calculated and mapped the index of the functional diversity of the entire ungulate community (FDis; *Laliberté and Legendre, 2010*) using the R statistical software (*Core Team, 2019*) and the R package FD v.1.0–12. The FDis was calculated based on the predicted relative densities of all ungulates and the following species-specific traits: body mass, diet type and gut type. The FDis is

the mean distance in multidimensional trait space of individual species to the centroid of all species. It can account for species abundances by shifting the position of the centroid toward the more abundant species and weighting distances of individual species by their relative abundances (*Laliberté and Legendre, 2010*). The input data for a FDis calculation are 1) a matrix with species traits and 2) a matrix with relative densities that describe how much weight to assign to each individual observation. We compiled a six row (species) by three column (traits) matrix with one quantitative and two qualitative traits, namely body mass, gut type and diet type (*Appendix 1—table 2*).

By means of hierarchical cluster analyses (*Lê et al., 2008*), we grouped all landscape grid cells (25 ha each) with similar values of FDis, and total and species-specific biomass into clusters representing ecologically distinct landscape-scale herbivory regimes, or 'herbiscapes'. Specifically, following the approach of *Hempson et al. (2015)*, we used the R package FactoMineR v.1.39 (*Lê et al., 2008*) and its HCPC (hierarchical clustering on principle components) function. The HCPC requires that PCA (principal component analysis) is performed on variables prior to clustering, which limits the impact of covariance amongst variables on the subsequent clustering algorithm. To build the cluster tree, we used HCPC default values for the metric (Euclidean distance) and method (Ward's) parameters. Similarly to *Hempson et al. (2015)*, the number of clusters was determined by assessing the inertia (i.e. change in within cluster homogeneity) gained by cutting the tree at different levels and the ecological interpretability of the resulting clusters. Eventually, we split the cluster tree into five independent clusters (*Appendix 1—figure 2*).

The hierarchical clustering analysis allowed us to learn how the combined effect of predation risk and resource quality translates into the composition and abundance of the ungulate community and, in consequence, into the diversification of herbivory pressure on the ecosystem.

## Vegetation analysis

Using an independent vegetation dataset from a large-scale inventory of tree regeneration (part of the LIFE+ ForBioSensing project, contract number LIFE13 ENV/PL/000048), we tested if the predicted variation in the landscape-scale distribution of large herbivores, synthesized into ecologically distinct herbivory regimes that is herbiscapes, affects tree browsing intensity and regenerating tree species composition. We used data collected in 2017 at 385 plots spread randomly across the entire BF. Each plot contained two concentric sub-plots: 1) with a radius of 1.3 m (area of 5 m$^2$) at which all trees with height <30 cm excluding seedlings were recorded, and 2) with a radius of 2.52 m (area of 20 m$^2$) at which all trees with height $\geq$30 cm and diameter at breast height <2 cm were recorded. Additionally, each individual tree was checked for any sign of fresh or 1-year-old browsing of its main shoot. The <30 cm tree sapling community is structured mainly by bottom-up factors (and/or forest management practices) and only minimally influenced by ungulate herbivory (*Kuijper et al., 2010a*), whereas the $\geq$30 cm tree sapling community is within the foraging height class preferred by ungulate herbivores (*Kuijper et al., 2013*) and is therefore largely structured by ungulate top-down factors (*Kuijper et al., 2010a*; *Kuijper et al., 2010b*).

Based on this data, for each plot we calculated the cumulative browsing intensity index, expressed as the proportion of browsed individual trees out of all tree saplings $\geq$ 30 cm, and the difference between the two tree height classes in the proportional shares of *Carpinus betulus* (*Carpinus*) and *Acer platanoides* (*Acer*). The latter two parameters represent a measure of recruitment from the sapling-bank (<30 cm) to the taller size-class ($\geq$30 cm). This process is to a great extent driven by large herbivores in the studied system (see *Kuijper et al., 2010a* and *Churski et al., 2017*). We specifically focused on the response of two contrasting species, *Carpinus* and *Acer*. While both species are palatable and strongly selected by the ungulate community (see *Churski et al., 2017*), *Carpinus* is highly browsing-tolerant (a typical 'brown-world' species sensu *Churski et al., 2017*) and *Acer* is highly-sensitive to ungulate browsing (a typical 'green-world' species). Long-term exclosure studies have also shown that *Carpinus* typically increases while *Acer* decreases in dominance in response to ungulate herbivory (*Kuijper et al., 2010a*). As both species are very common throughout the forest, we see them as suitable indicator species for the impact of ungulate herbivory on tree species composition. We explained the variation in the calculated parameters by fitting simple linear models with two interacting factors: *herbiscape* and *reserves*. The latter was added to account for potential differences in forest structure (see *Jedrzejewska et al., 1994*) and ungulate behaviour (see *Kamler et al., 2008*) between protected and unprotected areas in BF.

## Results

### Large carnivores

The main factor associated with the space use of both large carnivore species was human activity, as indicated by the positive effect of distance to major settlements on their spatial distributions (*Figure 3*). For the wolf, the density of protected areas was another important variable related to its landscape use. Wolves more often used large nature reserves (including BNP) than parts of the landscape dominated by managed forest. There was no statistically important effect of elevation, distance to touristic trails and density of touristic infrastructure on landscape distribution of neither the wolf nor lynx. However, wolves tended to use lower areas more intensively. For the lynx there was a clear tendency to use parts of the landscape further away from major human settlements; however, this effect was less evident than for wolf (the credible intervals overlapped at zero, *Figure 3D*). The density of protected areas had no effect on lynx distribution.

To test the quality of our predictions, we compared them with existing radio-tracking data from collared wolves collected over 20 years ago (1994–1999) in BF. This showed that our model fitted to the camera trapping dataset (471 wolf detections) not only conforms to the general pattern of the wolves' space use determined by telemetry 20 years ago, but also reflects the wolves' response to ongoing environmental changes in the study area. See the Appendix 1 and *Figure 4* for more details.

### Ungulates – landscape use

In contrast to the other ungulates, the red deer, the main prey of the wolf (*Jędrzejewski et al., 2002*), was the only species whose landscape use was associated with that of wolves (*Figures 4* and *5*). The predicted relative densities of both red deer females and males were lowest in parts of the landscape intensively used by wolves, and this effect was more pronounced for females. Lynx distribution was not related to the landscape use of any ungulate species. Red deer females were also positively associated with protected areas (BNP and nature reserves) and mixed deciduous forest with an intermediate level of landscape openness (*Figure 4*). Red deer males showed similar tendencies, but these factors were not statistically important predictors of their landscape distribution. Instead, red deer males showed a negative association with distance to human settlements, which could have resulted from their more intensive use of open meadows surrounding settlements, especially during the rutting period. Forage habitat availability was the main factor associated with wild boar distribution, as indicated by its clear association with closed-canopy and deciduous species dominated forest stands (as indicated by the strong negative effect of coniferous species dominated forest stands; *Figure 4*). The presence of deciduous forest was also the main factor positively associated with the distribution of bison, followed by proximity to human settlements (as indicated by the negative association with distance to human settlements). The latter is likely related to the presence of meadows surrounding settlements, which provide optimal foraging habitats for bison (*Bocherens et al., 2015*; *Cromsigt et al., 2012*). The moose was the only species whose landscape-scale distribution was positively associated with low elevation areas (river valleys and wetlands). Roe deer were positively associated with intermediate levels of landscape openness. However, no other predictors that we used were related to the distribution of roe deer, which may be due to the low density of this species in our study system. All ungulate species except red deer males showed some level of remaining spatial auto-correlation not explained by the raster covariates included in the models (*Figure 6*). The spatial random effects were particularly large for the bison, whose spatial distribution in BF is strongly influenced by supplementary winter feeding and use of open areas outside the forest throughout the year (*Kowalczyk et al., 2011*). An interesting hot-spot of unexplained variation in distribution of bison and red deer females was the south of the national park, in an area without hunting and rich old-growth deciduous stands and with known higher densities of deer (*Jędrzejewska et al., 1997*).

### Ungulates – detection rates

At the scale of single camera trap sites (1 ha), increased canopy openness led to higher detection rates of red deer and moose (*Figure 7*). This could be explained by cameras having better detection in open areas (*Marcus Rowcliffe et al., 2011*) and/or a preference of these species to forage in

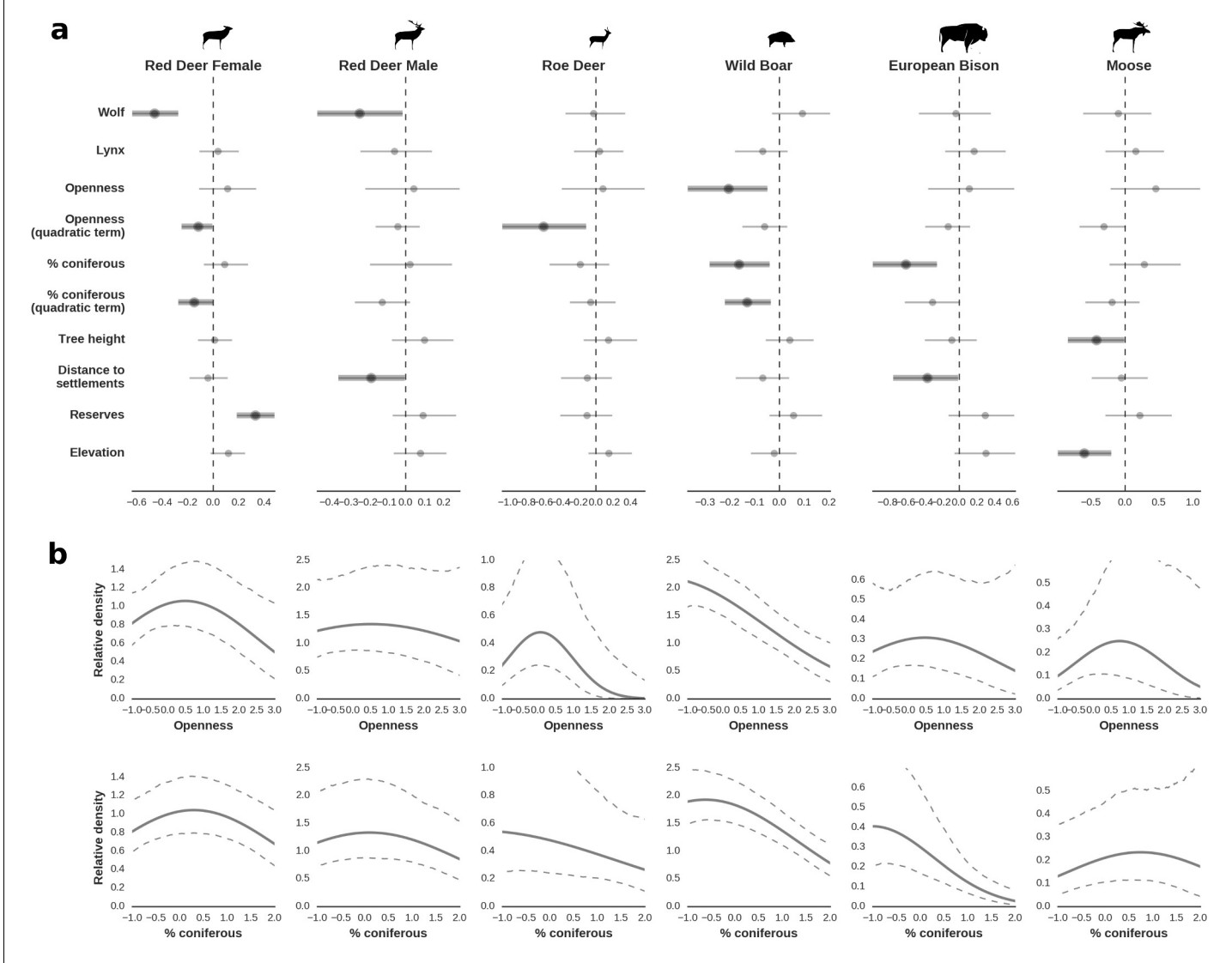

**Figure 4.** Spatial distribution of each ungulate species was related to a unique combination of bottom-up and top-down factors. (**a**) Estimated effects of covariates explaining the spatial variation in the parameter $\lambda$, which is the expected number of individuals using a given landscape grid cell (25 ha pixel) during the sampling period (i.e. the relative density; see the general and formal description of the model in the Materials and methods section). In bold, we marked the credibility intervals (quantile based) at which the estimated parameter values differed from 0 (for the level 2.5–97.5%, there is 0.95 probability that the true parameter value lies within this range). The full posterior distribution for each parameter, together with its numerical description can be found in Appendix 1. (**b**) For easier interpretation of the effects of landscape openness and percentage share of coniferous species (together with their quadratic terms and assuming an average level for the other covariates), we plotted predictions of the relative density for each species and for each variable. The values of the covariates were scaled and zero-centered before making predictions; thus 0 corresponds to the average value of a given covariate. The full posterior distribution for each parameter, together with its numerical description can be found in *Appendix 1—figures 14–25*).

DOI: https://doi.org/10.7554/eLife.44937.006

canopy gaps (*Churski et al., 2017*; *Kuijper et al., 2009*). Red deer females were associated with even larger canopy gaps, whereas for red deer males and moose there was an optimum value of canopy openness, above which the detection rate started to decrease. Roe deer followed a similar pattern as red deer but there was no statistical support for this result. At-site detection rate of wild boar was highest in deciduous forest patches with relatively closed canopies (in line with *Kuijper et al., 2009*). However, wild boar, as well as red deer, roe deer and bison were also relatively frequently detected at forest patches dominated by coniferous tree species, indicating a

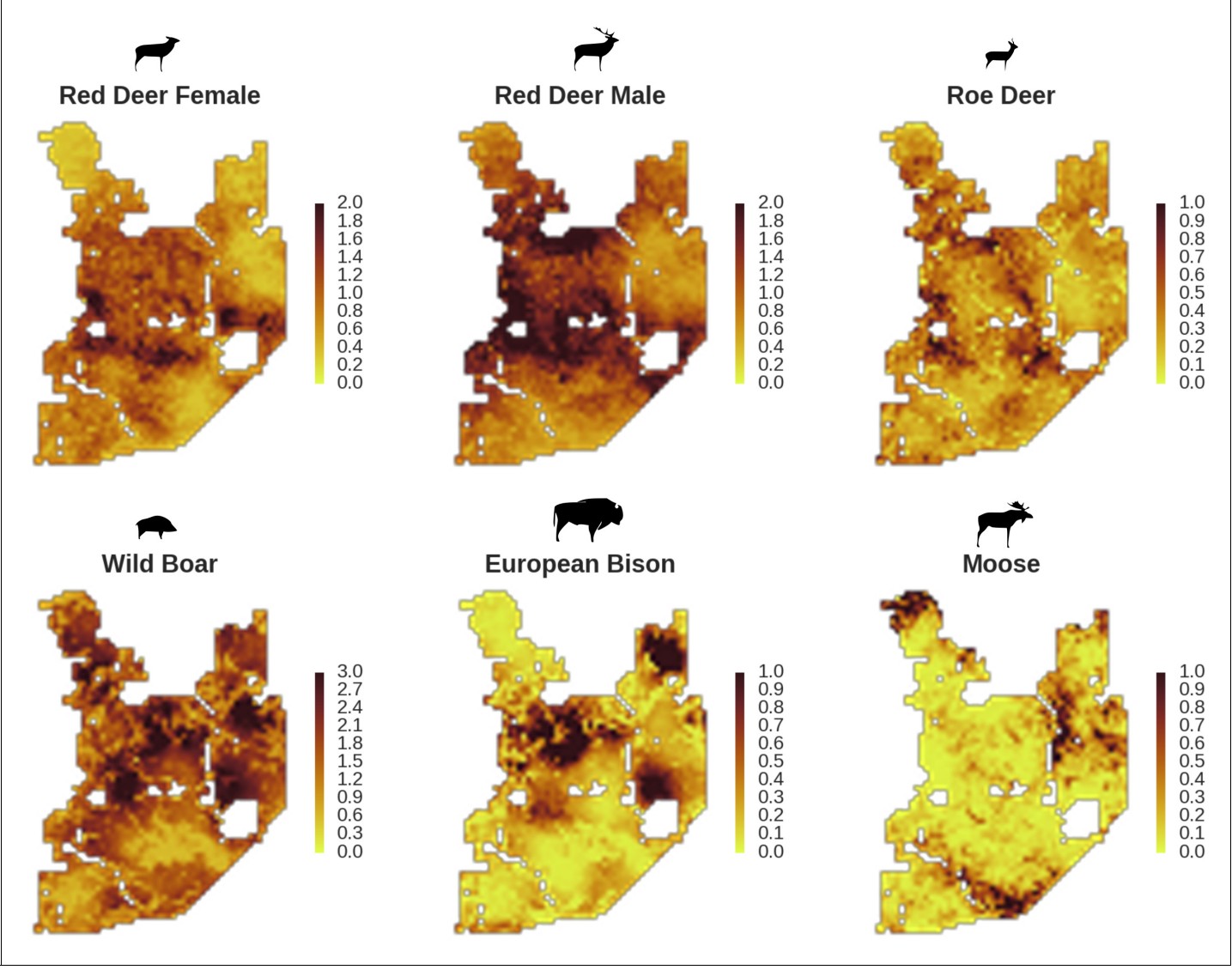

**Figure 5.** Only red deer (the major wolf prey) showed a negative spatial association with parts of the landscape intensively used by wolves, whereas distributions of other species were shaped by bottom-up factors. The maps show substantial variation in predicted landscape use for the five studied ungulate species (relative density surfaces). Specifically, this figure presents the spatial predictions for the parameter which is the expected number of individuals using a given landscape grid cell (25 ha pixel) during the sampling period (see the general and formal description of the model in the Materials and methods section). The predictions are based on the parameter estimates presented in *Figure 3* and include the spatial random effects (see *Figure 4*).

DOI: https://doi.org/10.7554/eLife.44937.007

context-dependence in the selection of small habitat patches. For example, as forage availability changes between seasons, deciduous patches may be preferred in the green season while coniferous patches in winter. Temperature affected detection rate non-linearly by influencing the activity level of red deer males, with the optimum at the yearly average temperature. In the case of the wild boar the highest detection rates were found at high temperatures, coinciding with their reproductive period in summer. And in case of the moose, the highest activity level was found at moderately high temperatures. The bison was the only species whose detection rate was strongly affected by snow cover. Winter supplementary feeding causes bison to aggregate near feeding stations or outside BF (*Kowalczyk et al., 2011*), hence dramatically decreasing their detection rate in other parts of the forest.

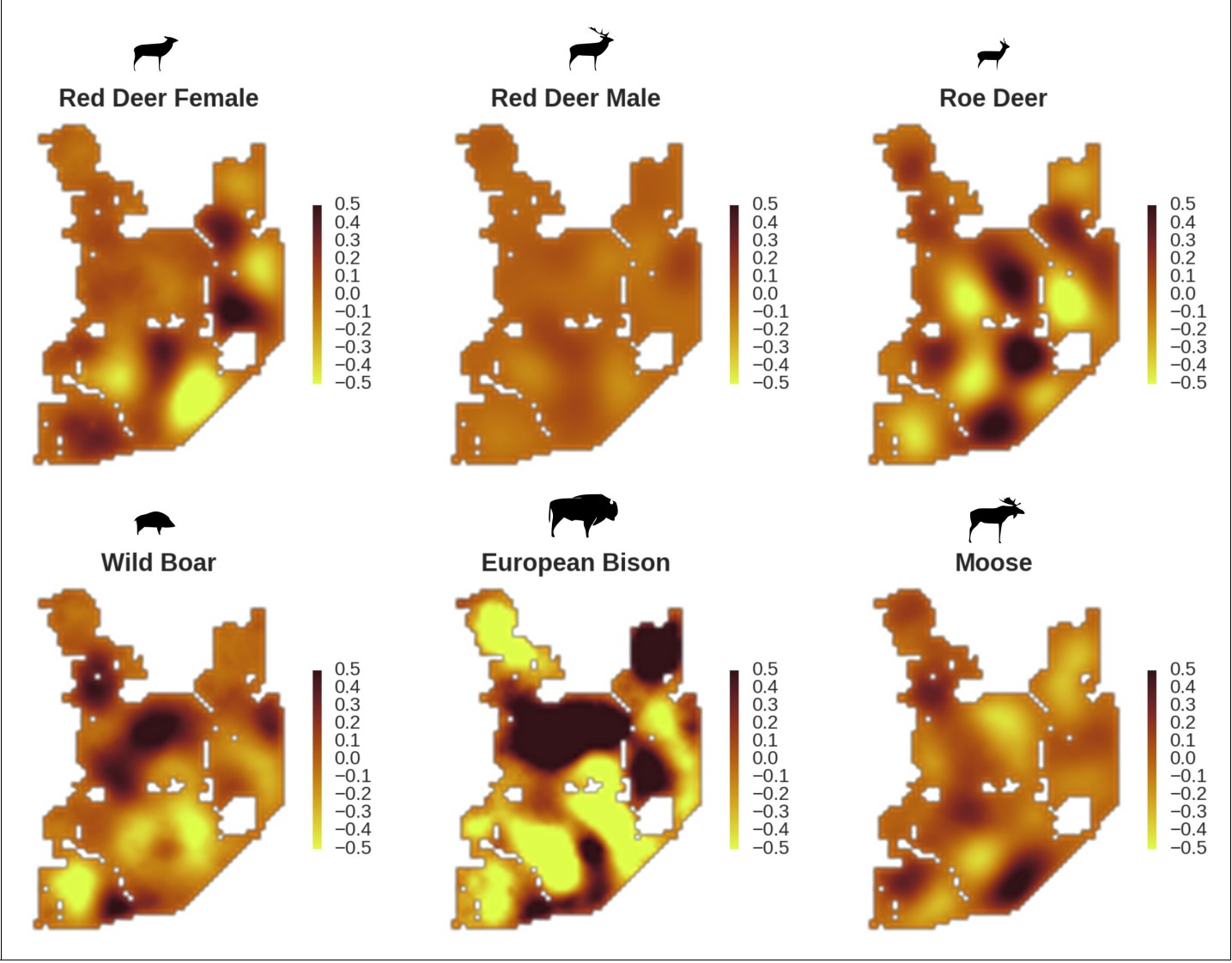

**Figure 6.** All ungulate species except red deer males showed some level of remaining spatial auto-correlation not explained by the raster covariates included in the models. The maps show the fitted spatial random effects (SRE) for the models of five studied ungulate species. The SRE are deviations from 0 at log-scale. The SRE captured all spatial variation not explained by covariates included in the model, indicating parts of the landscape for which higher ('hot-spots') or lower ('cold-spots') activity of species was observed in the data than predicted by the 'fixed' part of the model. The SRE were particularly large for the bison, whose spatial distribution in BF is strongly influenced by supplementary winter feeding and the use of open areas outside the forest during the year.

DOI: https://doi.org/10.7554/eLife.44937.008

## Ungulates – spatial overlap

All ungulate species except moose showed some level of pairwise positive spatial associations as indicated by the Pearson correlation of their relative density surfaces predicted by the models (*Figure 8*). However, the strength of these associations was relatively low, indicating substantial spatial variation in the structure of the whole community. Unsurprisingly, the strongest overlap in space was between red deer females and males. However, the estimated correlation (0.71) was far from a 'perfect' overlap, indicating red deer are sexually segregated in space in BF (see *Kamler et al., 2008*). The relatively high spatial overlap between red deer males and roe deer (0.58) likely resulted from their utilization of similar parts of the landscape, often close to the forest edge and large clearings with human settlements. The moose was a clear exception showing a negative spatial association with all other ungulate species. This indicates that the moose has a specific (spatial) niche in our

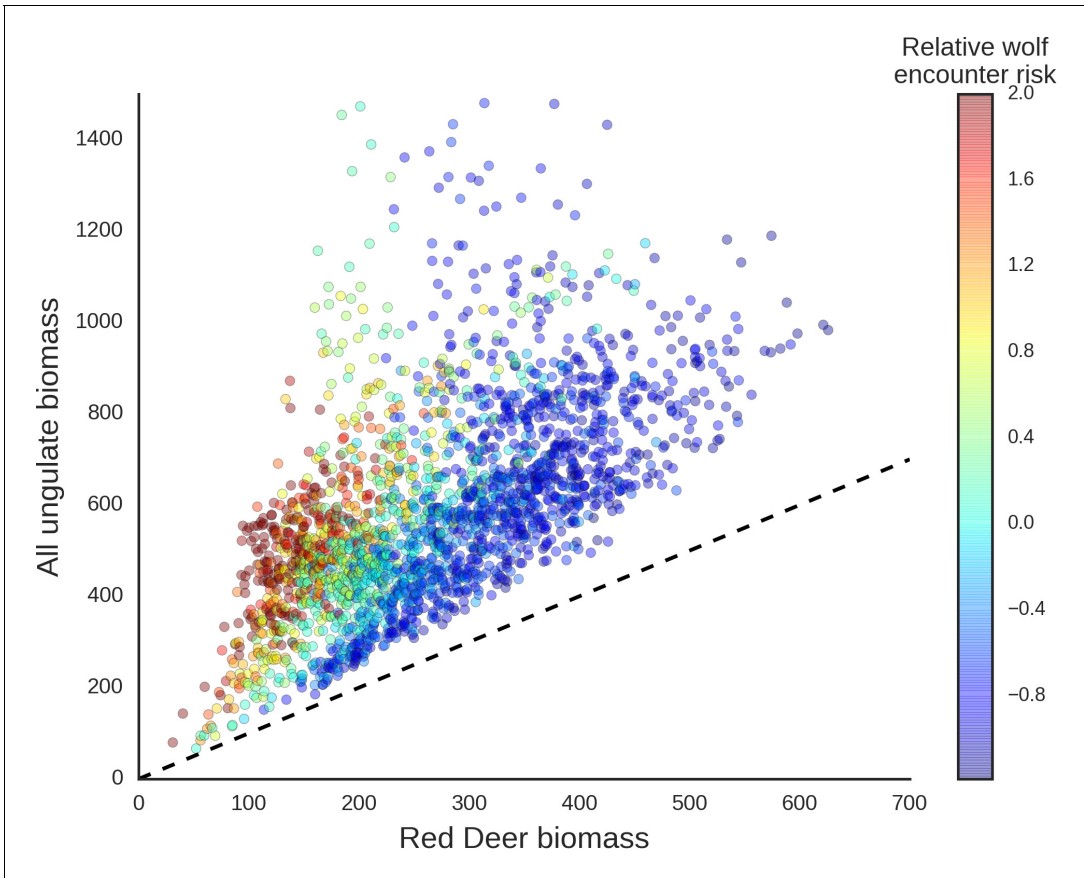

**Figure 14.** Scatter plot showing how wolves, by affecting the spatial distribution of red deer, re-structured the composition of the entire community of large herbivores. The predicted biomass of red deer (both sexes, X axis) is shown as a share of the total biomass of the entire community of ungulates. Each dot on the plot is a 25 ha landscape pixel. In color, we marked relative wolf encounter risk (scaled values of the parameter $\lambda$) as predicted by the model fitted to the wolf camera trapping data. We interpreted the parameter $\lambda$ as large carnivore space use intensity and assumed that from a prey perspective, $\lambda$ is proportional to the predator encounter rate. The dashed line is a reference line indicating the y=x relationship.
DOI: https://doi.org/10.7554/eLife.44937.016

system, strongly associated with low lying areas like river valleys and wetlands (see *Figures 2*, *4* and *5*). Interestingly, when comparing the surfaces of spatial random effects, there was a relatively strong positive spatial association (0.59) between bison and wild boar (*Figure 6*). A possible explanation for this pattern may be that wild boar are attracted to supplementary food at bison feeding stations, where next to hay and silage, beetroots are provided for bison (*Kowalczyk et al., 2011*).

## Functional diversity and hierarchical clustering analysis

When mapped, both the estimated total biomass of the ungulate community and the functional diversity index (FDis) showed distinct patterns across the studied landscape (*Figure 9*). Interestingly, the lowest values of both parameters were associated with coniferous-dominated tree stands at higher elevations (*Figures 2* and *5*) belonging to the least productive parts of this landscape (*Faliński and Falińska, 1986*; *Kwiatkowski, 1994*). By means of hierarchical cluster analyses, we further grouped all landscape grid cells (25 ha each) with similar values of FDis, and the total and species specific biomasses. We identified five clusters, characterized by different sets of risk-related and environmental factors and composed of different sets of ungulate species (*Figure 10*). The 'red' cluster (id = 1) was characterized by high wolf and lynx use and low quality foraging habitat (low elevation, mixed tree stands with large shares of coniferous tree species, *Figure 10B*). In terms of ungulate biomass, this cluster was dominated by moose, red deer and wild boar (*Figure 10C*). However, red deer relative density was the lowest out of all five clusters. The total ungulate biomass in this cluster was low but the FDis value was relatively high (*Figure 10D*). A large part of the 'red'

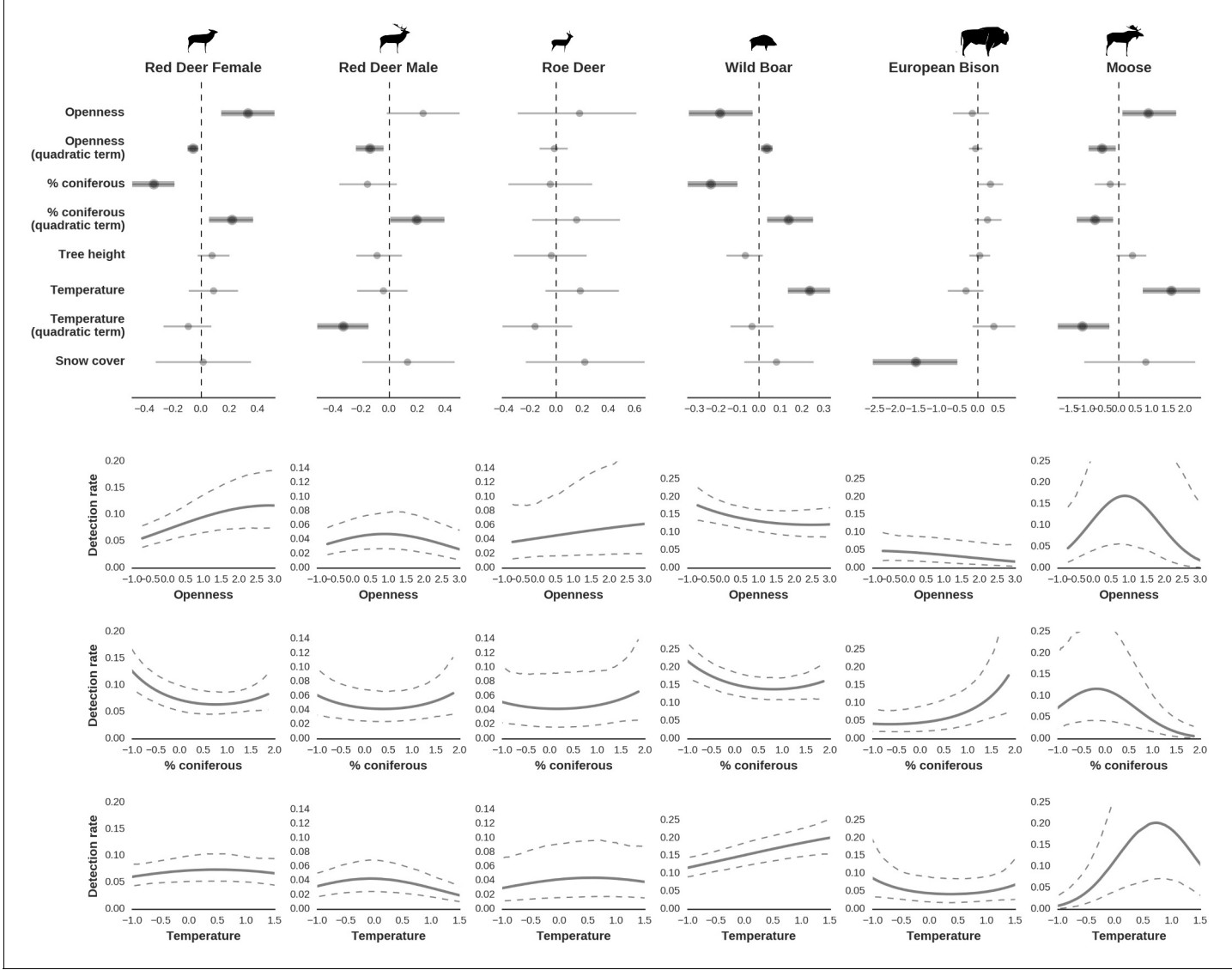

**Figure 7.** Factors influencing detection rates of ungulate species. (**a**) Estimated effects of covariates explaining the variation in the parameter γ, which is the expected (daily) detection (trapping) rate at a single camera trap site (see the general and formal description of the model in the main text in the Materials and methods section). The spatial covariates were calculated at a pixel resolution of 100 m (1 ha) and the temporal covariates were calculated as the average value for the whole period of camera trapping at a given location. In bold, we have marked the credibility intervals (quantile based) at which the estimated parameter values differ from 0 (for the level of 2.5–97.5 %, there is a 0.95 probability that the true parameter value lies within this range). The full posterior distribution for each parameter, together with its numerical description can be found in Appendix 1. (**b**) For easier interpretation of the effects of canopy openness, the percentage share of coniferous species and temperature (together with their quadratic terms and assuming an average level for other covariates), we plotted predictions of the detection rate for each species and for each variable. The values of the covariates were scaled and zero-centered before making predictions; thus 0 corresponds to the average value of a given covariate. The full posterior distribution for each parameter, together with its numerical description can be found in (*Appendix 1—figures 14–25*).

DOI: https://doi.org/10.7554/eLife.44937.009

cluster was spatially associated with the lowest elevated areas, that is marshlands and river valleys (compare *Figure 10A* with *Figure 2*). The 'green' cluster (id = 3) was characterized by high predator presence and high-quality foraging habitat (moderate elevation, mixed tree stands with large shares of deciduous tree species), which increased the numbers of seemingly risk-insensitive species like wild boar and bison. Red deer males, which had a less pronounced negative spatial association with wolves than females (*Figure 4*), also had a higher biomass in this cluster than in the 'red' one. More-over, the 'green' cluster was characterized by low values of total biomass and high values of FDis.

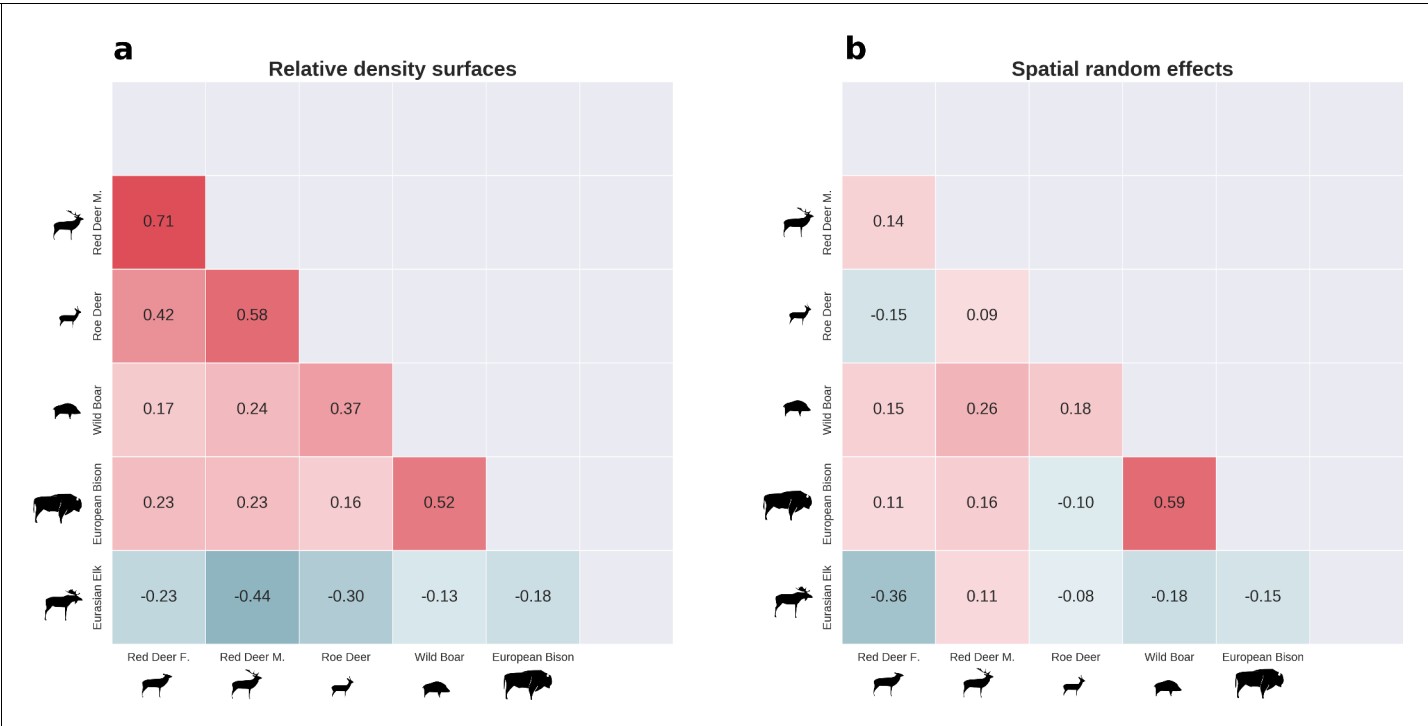

**Figure 8.** There was substantial spatial variation in the structure of the ungulate community, as indicated by the weak spatial associations between species. The pairwise spatial overlap between the studied ungulate species is presented as the Pearson correlation heatmap of their relative density surfaces predicted by the models (**a**) and fitted spatial random effects (**b**).
DOI: https://doi.org/10.7554/eLife.44937.010

This cluster in the landscape, mainly covered remote areas of low human activity and high use by wolf, and occurred in protected areas (BNP and nature reserves). The 'blue' cluster (id = 2) was characterized by moderate use by predators and low-quality foraging habitat (high elevation, coniferous

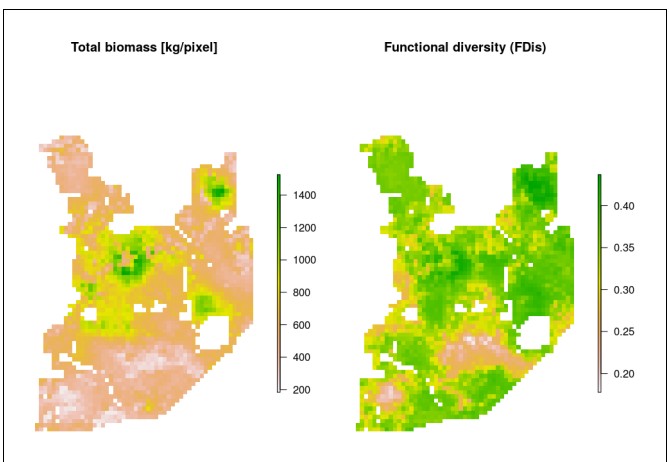

**Figure 9.** Maps of estimated total biomass [kg/pixel] and functional diversity (FDis). Both variables, together with species-specific biomass raster layers, were used in the hierarchical clustering analysis. Interestingly, the lowest values of both parameters were largely associated with coniferous-dominated tree stands at higher elevations (compare with *Figure 2*), often re-planted after extensive clear-cuts from the past and belonging to the least productive parts of this landscape (*Faliński and Falińska, 1986*; *Kwiatkowski, 1994*).
DOI: https://doi.org/10.7554/eLife.44937.011

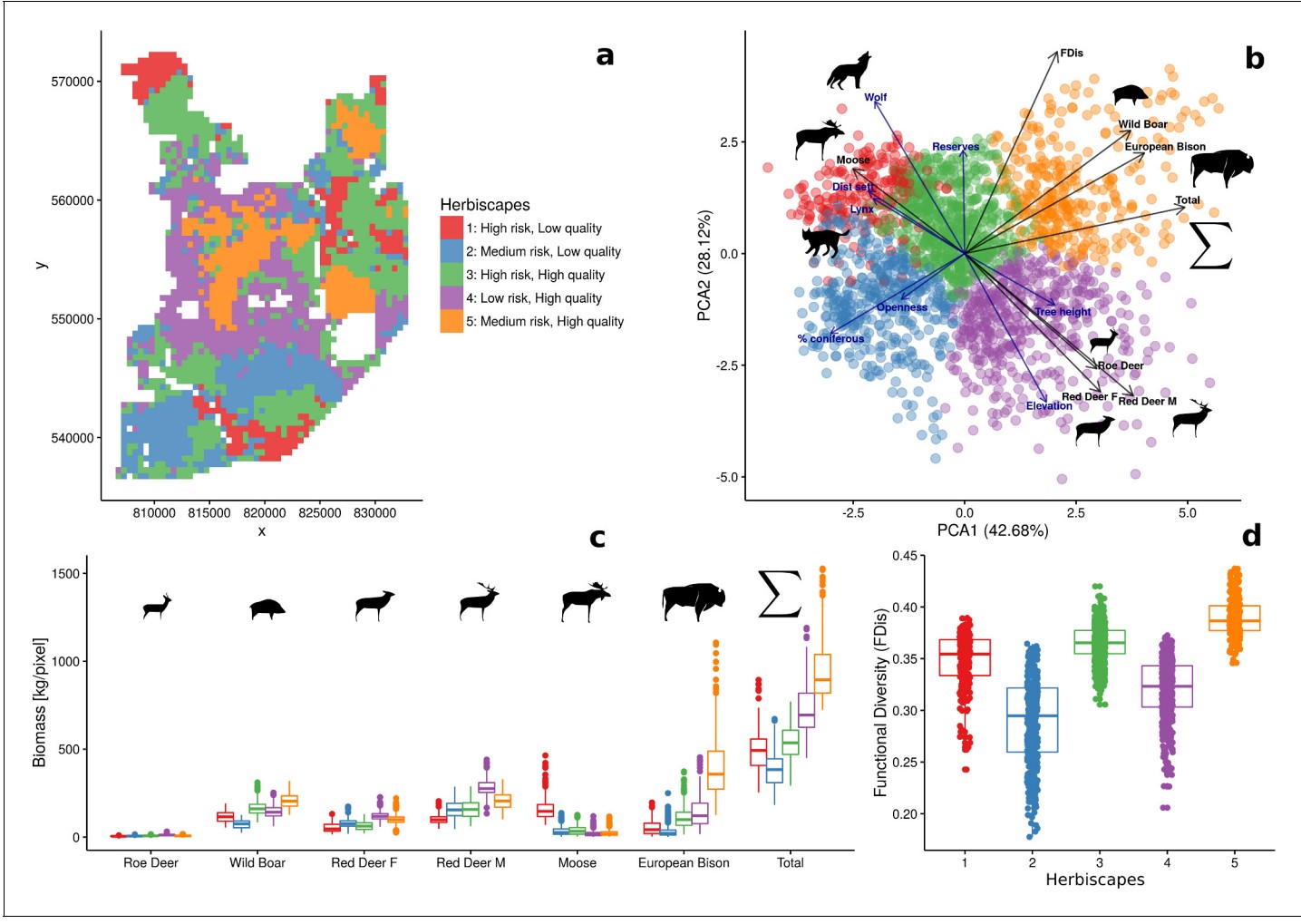

**Figure 10.** Spatial variation and functional diversity of a herbivore community within a temperate forest herbivome, decomposed into landscape-scale herbivory regimes – herbiscapes. (a) The spatial distribution of the five identified clusters ('herbiscapes') based on hierarchical clustering on the principal components analysis. (b) The distribution of identified clusters in the space of the first two principal components. (c) Tukey-like boxplot comparing the species-specific and total biomass in each of the identified clusters. (d) Tukey-like boxplot comparing the functional diversity index (FDis) of the ungulate community between the identified clusters. The boxplots show median values and the lower and upper hinges correspond to the first and third quartiles. The colors of clusters are consistent amongst all figures. Each data point on panel (b) corresponds to a 25 ha pixel mapped on figure (a). On panel (b), for ease of interpretation, we additionally plotted (blue arrows) the major environmental and risk-related gradients in the study area which were used as covariates in the ungulate models.

DOI: https://doi.org/10.7554/eLife.44937.012

dominated tree stands), and had the lowest values of both total biomass and FDis. However, the wolf used these areas less intensively than the 'red' and 'green' clusters, and the biomass of red deer was higher than in those high wolf use clusters. The 'purple' cluster (id = 4) was characterized by low use by predators and high-quality foraging habitat (high elevation, mixed tree stands with large share of deciduous tree species), and was dominated by red deer. Both sexes of red deer were at their most abundant in this cluster, which covered areas of high human activity and the lowest use by wolf. These seemingly 'safe' parts of the landscape were also characterised by high-quality habitat for red deer – a mosaic of deciduous and mixed tree stands at higher elevations and relatively close to forest edges and large clearings surrounding human settlements. The roe deer was also spatially associated with this cluster. The 'orange' cluster (id = 5) was characterized by moderate use by predators and high-quality foraging habitat (moderate elevation, deciduous dominated tree stands) and had the highest values of total biomass and FDis. This cluster was dominated by the bison - the main grazer and largest species in our system. Also, the wild boar was at its highest relative density

in this cluster. It is worth mentioning that a large part of this cluster was in the southern part of Biało-wieża National Park, comprising some of the best preserved parts of this forest.

## Spatial patterns in herbivores and impact on woody vegetation

The highest proportions of browsed trees were found in herbiscapes 3, 4 and 5 (*Figure 11c*), which are characterized by high values of total ungulate biomass and/or high functional diversity index

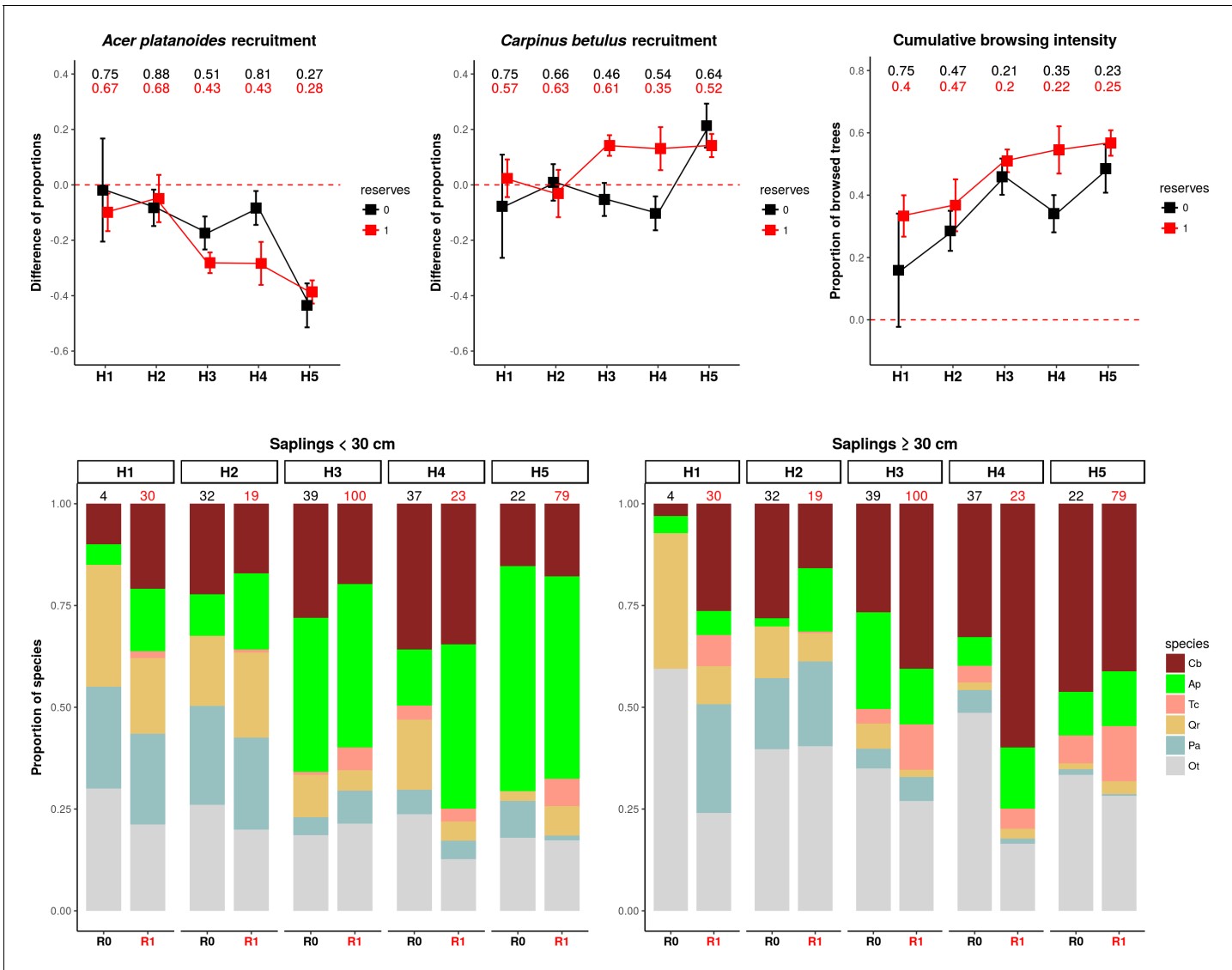

**Figure 11.** The patterns of browsing intensity and the composition of regenerating tree species differ between herbiscapes and between the protected and managed parts of the BF landscape. (a, b) The mean differences between the two tree height classes (saplings < 30 cm and ≥30 cm) in the proportional shares of *Acer platanoides* (*Acer*) and *Carpinus betulus* (*Carpinus*). The numbers on top of both figures are the proportions of plots without a single individual of each species. (c) The cumulative browsing intensity index, expressed as the proportion of browsed individual trees out of all tree saplings ≥ 30 cm. The numbers on top of this figure are the proportions of plots with no browsing. Panels (a), (b) and (c) show the values predicted by the fitted linear models with two interacting factors: *herbiscape* (six levels) and *reserves* (two levels coded as 0: unprotected areas and 1: protected areas). The error bars are standard errors of the mean. (ᴅ) The mean proportions of selected tree species (saplings < 30 cm and ≥30 cm) out of the entire pool of the main regenerating trees recorded at the sampled plots, showed for the herbiscapes and protected (R1) and unprotected (R0) areas separately. The value on top of each stacked bar is the number of plots used to calculate the mean proportion. Cb: *Carpinus betulus*, Ap: *Acer platanoides*, Tc: *Tilia cordata*, Qr: *Quercus robur*, Pa: *Picea abies*, Ot: other species (*Betula pendula, Sorbus aucuparia, Fraxinus excelsior, Alnus glutinosa, Populus tremula, Pinus sylvestris, Ulmus*).
DOI: https://doi.org/10.7554/eLife.44937.013

(FDis). These three herbiscapes covered the more fertile parts of the landscape dominated by productive deciduous and mixed forests (*Figure 10B*) with abundant tree regeneration dominated by *Acer platanoides* and *Carpinus betulus* (*Acer* and *Carpinus* hereinafter, *Figure 11d*). The variation in browsing intensity between the herbiscapes was reflected in the recruitment patterns towards taller tree sapling size classes (>30 cm) of *Acer* and *Carpinus*. In accordance with previous experimental exclosure studies in BF (*Churski et al., 2017*; *Kuijper et al., 2010a*; *Hedwall et al., 2018*), the proportion of the palatable but browsing intolerant *Acer* in the community of regenerating trees decreased as herbivore pressure increased (*Figure 11a,d*). In contrast, the palatable but highly browsing-tolerant *Carpinus*, showed the opposite pattern (*Figure 11b,d*). These supposedly herbivore-driven shifts in the species composition of regenerating trees were most pronounced in nature reserves, whereas areas outside the reserves broadly showed the same patterns but less closely followed the observed patterns in cumulative browsing intensity.

## Discussion

Our results show how variation in spatial distribution in a community of large herbivores in a temperate forest ecosystem can be explained by species-specific associations with major ecological gradients operating at the landscape scale. This creates ecologically distinct landscape-scale herbivory regimes ('herbiscapes'), which are interactively driven by both environmental bottom-up and biotic top-down (large carnivores) factors in combination with human-driven (cascading) effects. In addition, our analyses suggest that these herbiscapes differ in browsing intensity and impact on vegetation, indicated by the changing proportions of recruitment of browsing-sensitive versus browsing-tolerant tree species (*Figure 12*).

### Spatial heterogeneity in the landscape distribution of large herbivores

All ungulates in our study showed specific, non-uniform distributional patterns that were associated with species-specific combinations of bottom-up and/or top-down forces, including predation, human presence, availability of resources and the (bio)physical properties of the landscape (*Figures 4* and *5*). The substantial spatial variation in each of these landscape components, when combined, resulted in an aggregated, non-uniform pattern of landscape distribution of all ungulates, as predicted by *Fryxell (1991)* and *Hopcraft et al. (2010)*. The spatial distribution of each species (*Figure 5*) followed a characteristic shape of a hollow or sigmoidal curve (*Figure 13*) when plotted as a graph of ranked relative densities predicted for each 25 ha landscape pixel. This pattern of spatial variation in abundance of different populations has been shown to be universal over large spatial domains and for different taxa (*Brown et al., 1995*). Our study contributes to this finding by showing that similar spatial patterns can be observed within a population and within a local landscape. The shape of these ranked-abundance curves is informative for the properties of a given continuous (landscape) surface (*Rocchini and Neteler, 2012*), for example the sigmoidal curve of wild boar indicates a highly heterogeneous distribution with many hot and cold spots (high and low use), whereas the hollow curve of moose indicates a rather homogeneous distribution at low density with only some hot-spots.

Interestingly, the degree of spatial aggregation of different ungulate species in our study system was not related to their body size. This contrasts with studies from African savannas where larger herbivore species were more evenly distributed over the landscape than the smaller species (*Cromsigt et al., 2009*; *du and Owen-Smith, 1989*). The two largest herbivores in our study system, the European bison and moose, showed highly aggregated landscape distributions. The moose was strongly associated with lower elevation habitats (i.e. river valleys and wetlands), whereas landscape use by bison was strongly related to locations of supplementary winter feeding and open areas inside the forest, which are distributed sparsely within the study area and often associated with anthropogenic activity. The distribution pattern of bison could also be the result of partial migrations outside the forest in winter (*Kowalczyk et al., 2013*; *Kowalczyk et al., 2011*). This mainly human-driven distribution of bison was also manifested in the largest values for unexplained spatial variation in the model output (fitted spatial random effects, *Figure 6*).

The distributions of all species, when combined, revealed substantial spatial variation in the composition of the entire ungulate community (*Figure 10*). To our knowledge this is the first empirical study presenting a synthesized, high-resolution and spatially explicit approach (sensu *Royle et al.,*

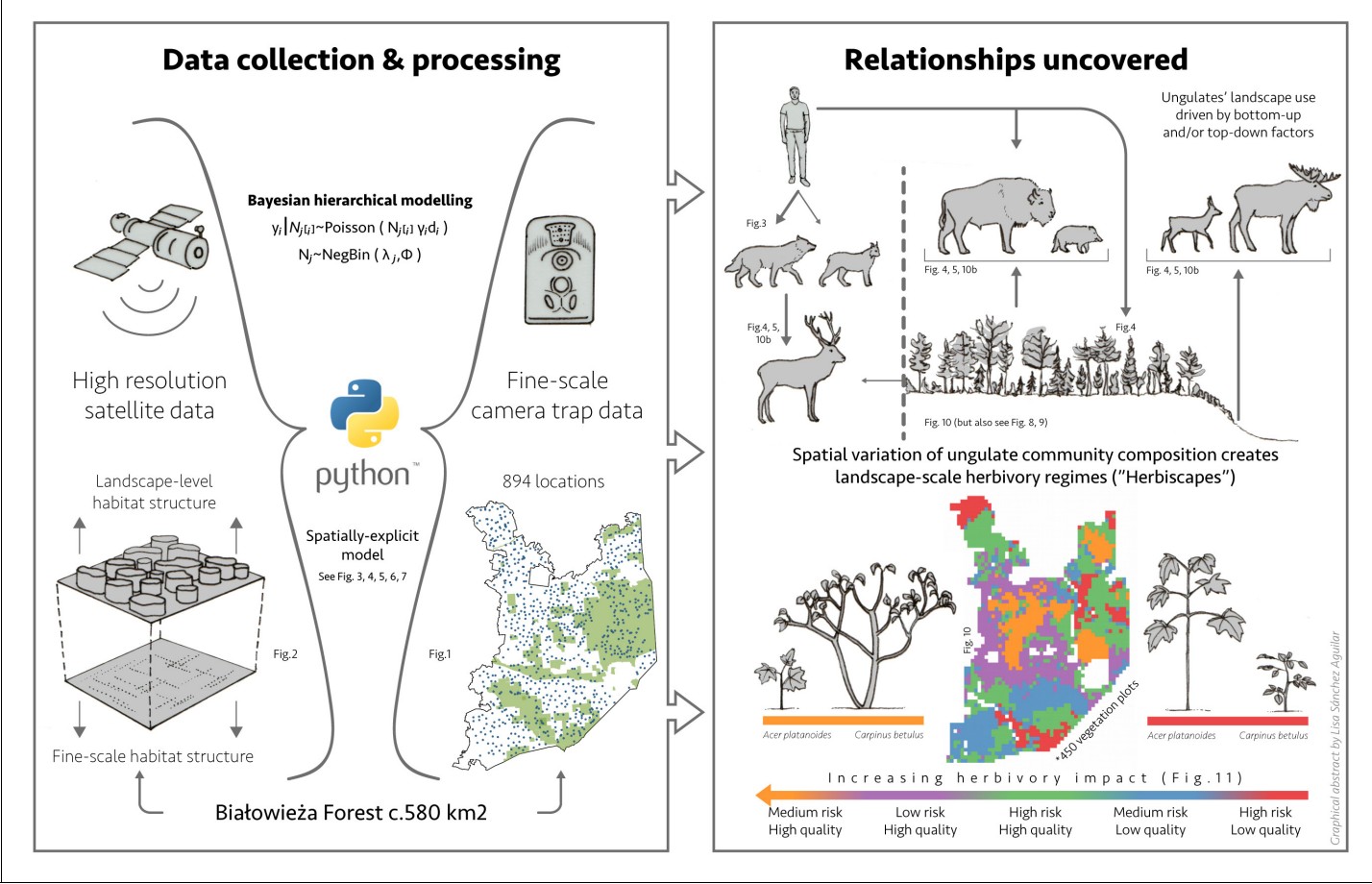

**Figure 12.** Graphical abstract presenting the steps taken to reveal the relationships uncovered in this study.

Image credit: Lisa Sánchez Aguilar

DOI: https://doi.org/10.7554/eLife.44937.014

*2007*) combining bottom-up and top-down factors to explain the landscape-scale variation of an entire large herbivore community in a temperate forest ecosystem. Including both resource- and predator-related factors was critical for achieving this goal as they can operate simultaneously and interactively (*Anderson et al., 2010*; *Fryxell, 1991*; *Hopcraft et al., 2010*). Similar studies in African ecosystems have shown that the distribution and diversity of a community of large herbivores can be driven by bottom-up (e.g. habitat heterogeneity; *Cromsigt et al., 2009*), top-down (e.g. anthropogenic fire; *Klop and Prins, 2008*) or interactive effects of both factors (e.g. distance to water and settlements; *Ogutu et al., 2010*). More recent studies, also from African ecosystems, have extended this approach by including predation-related factors, revealing the trade-offs native ungulates make to cope with changes in forage availability, human disturbance and predation risk (*Schuette et al., 2016*) and showing that a top predator can have species-specific spatial associations with herbivores (*Anderson et al., 2010*). For example, the latter study showed that lions were positively associated with large-bodied migratory ungulates but negatively associated with smaller non-migrants. Most comparably to our study, *Anderson et al. (2016)* used an extensive network of camera traps and a spatially-explicit occupancy modeling framework to quantify the spatial distribution of African savanna herbivores. Interestingly, they quantified pairwise interactions between all modeled species demonstrating the emergence of strong positive spatial associations among a diverse group of savannah herbivores. This is in line with our results where all ungulates except moose showed positive spatial associations. However, in our study, these spatial associations were rather weak, indicating substantial spatial variation in the structure of the ungulate community. Our contribution to all the above studies is a high-resolution picture of the spatial structure of an entire community of large

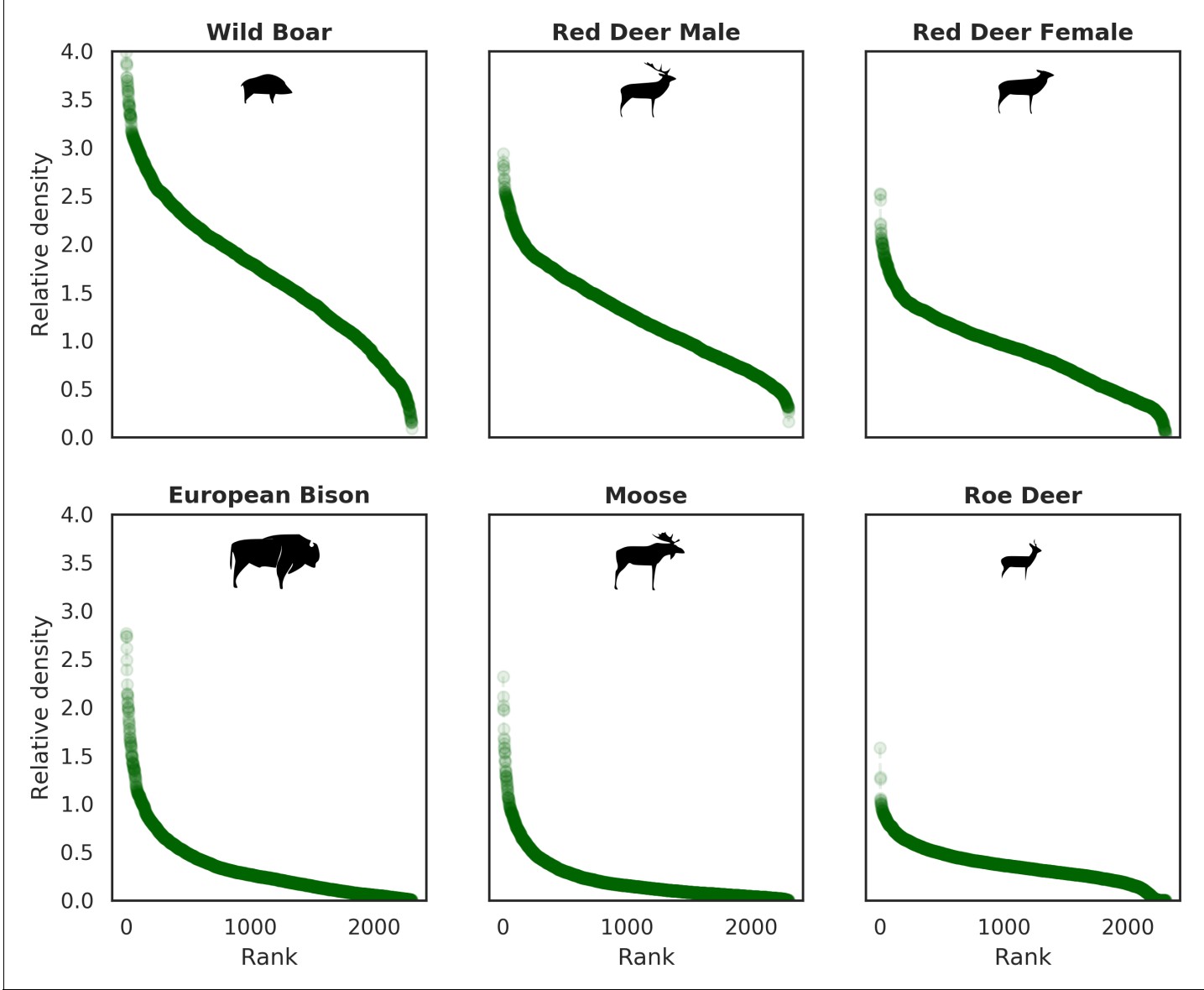

**Figure 13.** Ranked relative densities predicted for all ungulate species and for each 25 ha landscape pixel. All plots follow a characteristic shape of a hollow or sigmoidal curve (*Brown et al., 1995*; *Rocchini and Neteler, 2012*) indicating a non-uniform, heterogenous distribution of all studied species.
DOI: https://doi.org/10.7554/eLife.44937.015

herbivores that incorporates both bottom-up and predation- and human-related top-down factors. We believe one of our major points of novelty is in providing information on these spatial interactions for a temperate ecosystem.

## The effect of large carnivores on the spatial structure of the large herbivore community

The red deer was the only species whose landscape use was associated with that of large carnivores, with lower red deer presence in areas with higher wolf use. This is in line with other studies that have shown that predator top-down effects can work selectively on some members of large herbivore communities (*Sinclair et al., 2003*; *Valeix et al., 2009*) and have a complex and context-specific nature especially in multiple-prey and multiple-predator systems (*Davies et al., 2016*; *Moll et al., 2016*). However, wolves, by seemingly affecting the spatial distribution of red deer (see

below), one of the dominant species in BF, re-structured the composition of the entire community of large herbivores (*Figure 14*) and increased the degree of its spatial heterogeneity (*Figure 10*).

The red deer is the main prey species for wolves in our study area and experiences the highest predation pressure by wolves, in contrast to the European bison, moose and roe deer, which comprise only a small proportion of their diet, and wild boar, which is a secondary prey species (*Jędrzejewska et al., 1997*; *Jędrzejewski et al., 2002*). Hence, this might explain why the the red deer was also the species that seemed to react most strongly to the space use of its main predator. The most intensively used part of a wolf territory (on an annual basis) is related to the location of their dens during the reproductive period, and is generally far from human settlements (*Jędrzejewski et al., 2001*; *Kuijper et al., 2015*). These high wolf use areas likely only have higher predation rates during the reproductive period (*Kuijper et al., 2013*). Over the rest of the year, wolves move more widely across their territories (*Jędrzejewski et al., 2001*) and kills may be distributed more widely across annual wolf territories. Hence, the areas that seemed to be avoided by red deer are not necessarily areas with the highest predation risk on an annual basis. This finding adds to the growing recognition that prey species perceive risk based on various factors such as the space use of large carnivores and physical landscape and not necessarily by kill site distribution (*Kohl et al., 2018*; *Gaynor et al., 2019*).

The distributional pattern of red deer was similar for both sexes (one independent model was fitted for each), but females had a stronger negative association with wolf space use than males. This indicates that females are more sensitive to wolf presence than males and is consistent with the selective killing of this sex and juveniles by wolves in our study system (*Jędrzejewski et al., 2000*; *Jędrzejewski et al., 2002*). These apparent effects of wolf space use on red deer distribution could have resulted from both non-lethal (behaviorally mediated) as well as lethal (density-mediated) effects. With our data we were not able to distinguish between these two mechanisms, although previous studies have shown that non-lethal risk effects play an important role in affecting the responses of red deer at both fine- and large-scales in this system (*Kuijper et al., 2015*; *Kuijper et al., 2013*; *Kuijper et al., 2014*).

In contrast to the red deer, environmental bottom-up factors, particularly landscape topography and resource availability (natural or supplemented by humans), had the strongest associations with the spatial distribution of the other, less predation-sensitive ungulate species (*Figures 4* and *5*). Although the wild boar is a secondary prey species for wolves (*Jędrzejewski et al., 2002*), its high abundance during t study period means that predation by wolves contributed little to its annual mortality (*Jędrzejewska et al., 1997*). This may explain why we did not observe a negative association between wild boar and wolf space use. Moreover, a previous study of ours found that the wild boar displayed behaviour suggesting it perceived no predation risk in response to the presence of fresh wolf or lynx scats (*Kuijper et al., 2014*; *Wikenros et al., 2015*).

Our results seem contrary to those of *Theuerkauf and Rouys (2008)*, who carried out a similarly focused study in the same study area with use of pellet counts. They concluded that habitat alteration by forest exploitation and hunting by humans influenced the density distribution of ungulates, including red deer, more than predation risk by wolves. Despite our results indicating a spatial mismatch between the red deer and wolves' landscape use, this is likely to be caused by a cascading effect of humans. Moreover, the patterns of ungulate space use shown by our analyses revealed an additional source of variation, which involves a species-specific response to the inter-relationships between human and predator space use.

We were surprised to find that the Eurasian lynx – the other large carnivore in our study area – had no apparent effect. As ungulates constitute the bulk of the lynx's diet, with the roe deer being the major prey (60% of the diet) and the red deer being the alternative prey (22%) (*Okarma et al., 1997*), it is an important apex predator in this system. Lynx predation is a major mortality factor for both cervids, taking 21–36% of roe deer and 6–13% of red deer population numbers (*Okarma et al., 1997*; *Jędrzejewski et al., 1993*; *Jędrzejewska et al., 1997*; *Jędrzejewska and Jędrzejewski, 2005*). Moreover, it was recently revealed that red deer clearly react with anti-predatory behavior to olfactory cues of the lynx in BF (*Wikenros et al., 2015*). It is thus striking that these highly sensitive behavioral responses do not lead to changes in the spaces use of ungulates at the landscape scale. This may be a result of the combined effects of the low densities of both the roe deer and the lynx and the particular hunting mode of lynx, which, typically for felids, relies on fine-scale habitat characteristics that allow the predator to exploit prey independently of their spatial distribution

(*Podgórski et al., 2008*; *Schmidt, 2008*). Although there are many studies on the impact of large carnivores on the space use of their prey species in temperate systems, the majority of these have focused on single carnivore - single prey relationships (see e.g. *Creel et al., 2005*; *Kauffman et al., 2007*; *Lima and Dill, 1990*; *Mao et al., 2005*). Our study is the first we are aware of to show how these carnivore top-down effects structure an entire ungulate community in a temperate landscape. This knowledge is relevant as most terrestrial ecosystems are characterized by a diversity of herbivorous prey species and the differential responses of functionally similar prey species to apex predators can to a large extent determine the potential for trophic cascading effects of large carnivores (*Ford and Goheen, 2015*; *Ford et al., 2015*; *Rosenfeld, 2002*).

## Mediating predator-prey interactions at a landscape scale by humans

Humans can drive complex interactions between species, particularly by affecting keystone species like large carnivores (*Worm and Paine, 2016*). We found that humans were the main factor associated with the spatial distributions of both the wolf and lynx, which had lower activities in parts of the landscape heavily used by humans (in line with *Theuerkauf et al., 2003*). In this way, human presence can be beneficial for ungulate prey species as large carnivores generally avoid human presence and activity more strongly than their ungulate prey species (*Rogala et al., 2011*), leading to so-called 'human shields' (*Berger, 2007*). The observed spatial (re)distribution of red deer in our system seems to result from human-induced shifts in space use of the wolf. These kinds of three-way trophic cascades (following the definition of *Ripple et al., 2016*) involving humans, large carnivores and ungulates have been found in different systems, mainly in landscapes with moderate human activity (*Berger, 2007*; *Hebblewhite et al., 2005*) and are likely much more pronounced in highly human-dominated landscapes (*Kuijper et al., 2016*). We carried out our study in the forest considered to be the best preserved lowland forest ecosystem in Europe, which is replete with natural wildlife communities and ecological processes (including trophic interactions) still operating at the landscape scale. Thus, we believe it provides new knowledge on the fundamental structuring and functioning of European forest ecosystems that will help to predict the ecological effects of the ongoing large carnivore recolonisation of more human-altered habitats across Europe (*Chapron et al., 2014*).

## Landscape scale herbivory regimes and their potential consequences for vegetation

How the consumptive off-take by large mammal herbivores varies in space at a continental-scale has recently been shown by *Hempson et al. (2015)*, who classified African herbivore communities into ecologically distinct herbivory regimes, or *herbivomes*. In our study, we down-scaled this approach to the landscape level, exploring the spatial variation and functional diversity of the herbivore community within a single herbivome, decomposing it into ecologically distinct landscape-scale herbivory regimes. We refer to these as *herbiscapes*. Additionally, we extended the herbivome concept by including the direct and indirect top-down effects of higher trophic levels: large carnivores (keystone species) and humans (hyper-keystone species). Thus, similarly to a spatial 'profile' of each single species, the major dimensions of a herbiscape are oriented along both environmental (landscape feature, resource availability) and trophic interaction induced (direct and perceived predation risk, human activity) gradients operating over a landscape (*Figure 10*).

As large herbivores influence many important processes of ecosystem functioning (*Hobbs, 1996*), the observed landscape-scale variation in the structure of the large herbivore community (i.e. herbiscapes), when stable enough, could result in a differential impact of herbivores on (woody) vegetation. Our tree regeneration analysis revealed that the patterns of ungulate browsing intensity followed the variation in the modeled community-level distribution of ungulates in BF. The highest browsing intensities occurred in herbiscapes characterized by high ungulate biomass and/or functional diversity. Compositional changes in regenerating trees across herbiscapes were in accordance with our earlier experimental studies (*Churski et al., 2017*; *Kuijper et al., 2010a*): with higher levels of browsing intensity the proportion of a common browsing-tolerant tree increased and the proportion of a browsing-intolerant tree decreased. These patterns strongly suggest that the impact of ungulate communities on tree regeneration varies greatly at the landscape-scale, even in relatively homogeneous forest landscapes like BF. Earlier experimental studies showed that the species composition of the small tree sapling community in BF is shaped by environmental bottom-up factors,

whereas herbivores are the main factor limiting recruitment towards taller size classes (above 50 cm; *Kuijper et al., 2010a*). We therefore interpret the observed shift in dominance towards browsing-tolerant species in the taller height class (>30 cm) as the result of herbivore top-down effects, as indicated by the clear differences in browsing intensity (*Kuijper et al., 2010a*; *Churski et al., 2017*). Humans are an inherent part of these multi-trophic interactions as they affect both carnivore and species-specific herbivore space use, cascading down in a *complex* but *predictable* way to the lower trophic levels. Our study suggests, that the existence of more stable herbiscapes at larger spatial scales could create a mosaic of differential impact on woody plant communities and consequently spatial variation in expressed tolerance traits. Herbiscapes could therefore strongly contribute to the creation of a heterogeneous patchwork of green (less browsing tolerant) and brown (more browsing tolerant) worlds (sensu *Churski et al., 2017*) driven by variation in herbivory pressure at multiple spatial scales in temperate forest systems.

Recent views on the biome concept have emphasized the role of biotic interactions (e.g. herbivory) in creating multiple biome stable states under similar climatic conditions (*Moncrieff et al., 2016*; *Woodward et al., 2004*). This approach has proven to be very useful in explaining the spatial distributions of different vegetation communities maintained by functionally distinct guilds of herbivores at continental-scales (*Charles-Dominique et al., 2016*; *Hempson et al., 2015*). On the basis of the high-resolution data we collected on ungulate species distribution, our study revealed the presence of a large continuous variation in herbivore community structure at a much finer, within-landscape scale. Although European assemblages of wild herbivores are not as species-rich as those found in African savannas, the species we studied are clearly functionally diversified (see for example *Hofmann, 1989*). These differences in resource use and foraging behavior among species within the community could differentially impact the vegetation if certain parts of the landscape are dominated by different functional types of herbivores for a long enough period.

Studies from African savannas have indicated that the stability of these relationships is poorly understood, but is likely maintained by strong large-scale positive feedbacks between vegetation, abiotic resources and consumers (*Charles-Dominique et al., 2016*). This long-term stability likely provides the basis for co-evolutionary dynamics between functional types of herbivores and plants in these systems. Whether such feedbacks could also be operating at smaller spatial scales (i.e. in herbiscapes within a herbivome) and play a role in creating small-scale variation in vegetation structure is an intriguing question. While we do not believe they could be stable on an evolutionary time-scale, there could be sufficient stability over several decades to significantly impact woody plant communities (*Kuijper et al., 2010a*). In our system, these feedbacks have been controlled by landscape scale anthropogenic factors that have been present in a similar spatial arrangement for decades or even centuries (i.e. villages and roads have their origins in historical times; see for example *Samojlik et al., 2016*). As a result, the distribution of herbiscapes, driven by human-induced carnivore space use is arguably stable enough to create differential impact on vegetation in different herbiscapes. As humans are a crucial factor determining and restricting space use of large carnivores worldwide (*Tucker et al., 2018*), they likely contribute toward stabilizing herbiscapes in many human influenced landscapes across the globe.

Similarly, the results of our study have implications for patterns in vegetation structure that have been observed at large spatial scales. Herbiscapes will likewise also occur within the herbivomes described for the African continent. Our study indicates that predators can be a main factor structuring herbivore communities by redistributing predation-sensitive species over the landscape. As African systems harbor one of the most carnivore-rich communities in the world (*Ripple et al., 2014*), large carnivores are expected to greatly influence variation in herbivore community structure within African landscapes (*Thaker et al., 2011*; *Valeix et al., 2009*). A major difference between our, and African study systems, is that humans supposedly play a less pronounced role in determining the space use of large carnivores and herbivores (and hence the spatial arrangement of herbiscapes) in the latter (but see *Tucker et al., 2018*). The importance of this landscape-scale (or within-herbivome) variation in herbivore community structure in creating clear spatial patterns in vegetation, have already been illustrated for African systems. For example, *Ford et al. (2014)* showed that both predation risk and plant defenses enabled plants to thrive in different parts of a landscape; consequently, the thorniness of tree communities decreased across the landscape, contributing to intrabiome variation driven by predator-prey interactions (*Ford et al., 2014*).

The knowledge obtained in this study on spatial variation in the densities of local wildlife populations dependent on species-specific responses to habitat and disturbance factors gives us a better understanding of wildlife communities and may be relevant for effective wildlife management. In contrast to the common assumption that for management purposes wildlife populations are uniformly distributed, our study emphasizes the existence of large spatial variation in the landscape scale densities of ungulates. Such an approach fits particularly well in the recently developed concept of 'hunting for fear', which promotes the spatio-temporal diversification of management techniques based on perception of risk in wildlife (see *Cromsigt et al., 2012*). Moreover, maps visualizing cold- and hot-spots of ungulates across landscapes, together with information about community compositions, could also be useful tools for wildlife managers.

Our results also introduces valuable knowledge relevant to the conservation of natural habitats. We argue that a unique, and unfortunately still undervalued character of Białowieża Forest is that ecosystem processes are still operating here at a landscape scale, as shown by our study. The 'herbiscapes' proposed in this paper are another unique aspect of the ecological processes that should be preserved within this area and will contribute to the ongoing debate on future conservation strategies for Białowieża Forest as UNESCO World Heritage (*Mikusiński et al., 2018*).

## Conclusions

In conclusion, our study has illustrated that spatial variation in the structure of an entire large herbivore community results from interactive effects of species-specific responses to major ecological gradients operating at the landscape scale. Humans were a crucial factor associated with the landscape use of wolves and lynx. While lynx were not associated with the space use patterns of any ungulate, wolves were strongly (negatively) associated with the spatial distribution of their main prey species (red deer), affecting the ungulate co-occurrence patterns at the landscape scale. The space use of European bison, moose, roe deer and wild boar was related to food resources. These processes led to the distribution of different functional types of herbivores over the landscape and created a clear spatial structure in the herbivore community, which we referred to as herbiscapes. Vegetation analyses suggested that herbivore impact measured by browsing intensity and regeneration of browsing-tolerant tree species consistently differed between herbiscapes. When these herbiscapes are stable enough they could be an important mechanism driving variation in herbivore impact on woody vegetation and thus maintain heterogeneity in a wide range of ecosystems.

## Acknowledgements

We are grateful to Roman Kozak, Andrzej Waszkiewicz and Tomasz Kamiński for their field and lab work, and to Lisa Sanchez for preparing a graphical abstract of the article. We thank Bogumiła Jędrzejewska for her contribution at the initial stage of the project. We thank the administration of the Białowieża National Park, the administrations of Białowieża, Hajnówka and Browsk Forest Districts, and Polish authorities (Ministry of Environment, Polish General Directorate of Environment Conservation and Regional Directorate of Environment Conservation in Białystok) for permissions to work in Białowieża Forest. We thank European Space Agency (ESA) and Planet Labs Germany GmbH for providing RapidEye satellite imageries (ESA TPM PI project no. 28377). LIDAR and vegetation data used in the analysis were kindly provided by the team of the 'LIFE+ ForBioSensing' project which is co-funded by the EU Life Plus programme (contract number LIFE13 ENV/PL/000048) and the National Fund for Environmental Protection and Water Management in Poland (contract number 485/2014/WN10/OP-NM-LF/D). The work of JWB was supported by the National Science Centre, Poland (grant no. 2012/07/N/NZ8/02651). The work of DPJK and MC was supported by the National Science Centre, Poland (grant no. 2015/17/B/NZ8/02403), and the work of KS by European Nature Heritage Fund EURONATUR, Germany (grant no. PL-15-500-28). In addition, the work of DPJK and TAD was supported by funding from the National Science Centre, Poland (grant no. 2015/17/B/NZ8/02403 and grant no. 2017/25/B/NZ8/02466.

## Additional information

### Funding

| Funder | Grant reference number | Author |
|---|---|---|
| National Science Centre, Poland | 2012/07/N/NZ8/02651 | Jakub Witold Bubnicki |
| National Science Centre, Poland | 2015/17/B/NZ8/02403 | Marcin Churski<br>Dries PJ Kuijper |
| EuroNatur | PL-15-500-28 | Krzysztof Schmidt |
| National Science Centre, Poland | 2017/25/B/NZ8/02466 | Tom A Diserens<br>Dries PJ Kuijper |

The funders had no role in study design, data collection and interpretation, or the decision to submit the work for publication.

### Author contributions

Jakub Witold Bubnicki, Conceptualization, Resources, Data curation, Software, Formal analysis, Funding acquisition, Validation, Investigation, Visualization, Methodology, Writing—original draft, Project administration, Writing—review and editing; Marcin Churski, Conceptualization, Supervision, Validation, Investigation, Methodology, Writing—review and editing; Krzysztof Schmidt, Conceptualization, Resources, Writing—review and editing; Tom A Diserens, Writing—review and editing; Dries PJ Kuijper, Conceptualization, Supervision, Funding acquisition, Investigation, Writing—review and editing

### Author ORCIDs

Jakub Witold Bubnicki ⓘD https://orcid.org/0000-0002-2064-3113
Marcin Churski ⓘD https://orcid.org/0000-0001-8727-0203
Krzysztof Schmidt ⓘD http://orcid.org/0000-0002-9043-291X

### Decision letter and Author response

Decision letter https://doi.org/10.7554/eLife.44937.061
Author response https://doi.org/10.7554/eLife.44937.062

## Additional files

### Supplementary files

• Source data 1. Source data tables and Python code of the analysis.
DOI: https://doi.org/10.7554/eLife.44937.017

• Supplementary file 1. MCMC trace plots.
DOI: https://doi.org/10.7554/eLife.44937.018

• Transparent reporting form DOI: https://doi.org/10.7554/eLife.44937.019

### Data availability

All data generated or analysed during this study are included in the manuscript and supporting files. The source code of our analyses, together with the source data files, are available in our github repository at https://github.com/mripasteam/herbiscapes/ (copy archived at https://github.com/eli-fesciences-publications/herbiscapes).

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

## Appendix 1

DOI: https://doi.org/10.7554/eLife.44937.020

# Linking spatial patterns of terrestrial herbivore community structure to trophic interactions

Jakub W. Bubnicki, Marcin Churski, Krzysztof Schmidt, Tom A. Diserens, Dries J.P. Kuijper

## Methods

### Spatial covariates – data sources and processing

The GIS layers used as spatial covariates in our models were prepared using QGIS (**QGIS Development Team, 2017**) and GRASS GIS (**Neteler et al., 2012**) open source software. We compiled a set of high-resolution (5 m) rasters to describe environmental and human-related gradients within the study area. All raster covariates were summarized at two focal resolutions: 100 m (camera trap site scale) and 500 m (landscape scale). The density-based covariates, that is density of protected areas and density of touristic infrastructure were calculated for each pixel using $5 \times 5$ km kernel windows. The density of protected areas included Białowieża National Park and all strictly protected nature reserves in the study area. The density of tourist infrastructure was calculated based on locations of so-called 'Points of Interest' (POIs; e.g. hotels, restaurants, agrotourism farms, museums, visitors centers, camp sites but also smaller touristic infrastructure like car parking and picnic sites). The distance-based covariates (distances to all and major human settlements, touristic trails and forest edge) were calculated as raster proximity maps using the Euclidean distance to the nearest focal feature. The percentage of landscape (canopy) openness and average forest height were calculated based on the Digital Surface Model derived from LIDAR data (six points per m2) acquired in July 2015 within the framework of the LIFE+ ForBioSensing project (http://www.forbiosensing.pl) and provided us with a raster layer with resolution 5 m. We assumed all pixels with a vegetation height lower than 2 m were open areas and calculated their percentage share within 100 m (canopy openness at a camera trap site) and 500 m (landscape openness) grid cells. This data source was also used to describe variation in landscape topography (elevation). The percentage share of coniferous/deciduous tree species was derived through a supervised classification of a Rapid Eye satellite image acquired in June 2013 and provided by the European Space Agency (ESA; https://www.esa.int) as a raster layer with resolution 5 m. We performed the supervised classification of the Rapid Eye image using the Support Vector Machine (SVM) classifier (**Mountrakis et al., 2011**) as implemented in the scikit-learn Python package (**Pedregosa et al., 2011**). Two land cover classes were defined: coniferous dominated forest and deciduous dominated forest. All open (i.e. non-forest) areas were excluded from the classification based on the mask developed using the LIDAR data (see above). To collect spectrally homogeneous reference data for each land cover class we first ran a Large Scale Mean Shift (LSMS) segmentation as implemented in Orfeo Toolbox version 5.8.0 (**Grizonnet et al., 2017**). Next, the training and testing polygons were selected by image interpretation methods using very high spatial resolution data, such as the most recent available images in Google Earth (2014/10/02) and one Spot-6 scene (2015/06/27). Additionally, we used auxiliary information on tree species composition of each tree stand extracted from the State Forests inventory database (available at https://www.bdl.lasy.gov.pl). Next, we trained the SVM classifier using two-thirds of the reference dataset and the default settings in the scikit-learn implementation (**Pedregosa et al., 2011**). After assuring good performance of the classifier (>95% of test pixels correctly classified) we ran it for the whole study area. In the last step, we re-sampled the output raster to our focal resolutions (100 m and 500 m), calculating the percentage of coniferous and deciduous forest covering each landscape pixel. *Appendix 1—figure 2* presents the correlation heatmap of all candidate spatial covariates for the ungulate models. When selecting covariates for the final models we ensured that the Pearson correlation coefficient for all pairs of included covariates was lower

than 0.7. As there was a strong negative correlation (−0.97) between the percentages of coniferous and deciduous forest covering each landscape pixel we decided to include the former. For the same reason, we included distance to all settlements and excluded distance to forest edge (*Appendix 1—figure 2*). Also there was a strong correlation between distance to major settlements and distance to main roads (0.87), so we excluded the latter. The landscape use of lynx was determined by distance to major settlements (as indicated by the fitted model; see *Figure 1* in the main text), and hence the correlation between both variables was also high (0.82). Again for the reason discussed above, we excluded the distance to major settlements from the final models.

## Temporal covariates

The temporal covariates, that is temperature and snow cover, were calculated as the average values for the whole period of camera trapping at a given location. The source of the data were measurements acquired at the meteorological station in Białowieża.

**Appendix 1—table 1.** Summary of the two camera trapping surveys conducted during the study period. Days – the total number of trapping days (effort), Counts – the total number of individuals recorded during independent visits (events), Trate – trapping rate. Notice the dramatic differences in the trapping rates of the carnivores between the two types of surveys. Counts of red deer female (973) and red deer male (547) do not sum to the total count (2025) as there were 505 records of individuals of undefined gender.

| Species | Days | Counts | Trate (daily) | Trate sd | Trate max |
|---|---|---|---|---|---|
| *Ungulates survey (05.2012–05.2014)* | | | | | |
| Wild Boar | 9813 | 4818 | 0.49507 | 0.87881 | 10.16667 |
| Red Deer | 9813 | 2025 | 0.20255 | 0.36340 | 3.90909 |
| Red Deer Female | 9813 | 973 | 0.09648 | 0.20391 | 2.00000 |
| Red Deer Male | 9813 | 547 | 0.05544 | 0.12867 | 1.25000 |
| European Bison | 9813 | 440 | 0.04522 | 0.23212 | 4.50000 |
| Roe Deer | 9813 | 194 | 0.02043 | 0.06611 | 0.54545 |
| Eurasian Elk | 9813 | 82 | 0.00866 | 0.04714 | 0.66667 |
| Wolf | 9813 | 51 | 0.00541 | 0.02960 | 0.36364 |
| Eurasian Lynx | 9813 | 3 | 0.00027 | 0.00458 | 0.08333 |
| *Carnivores survey (09–10.2015)* | | | | | |
| Wolf | 3093 | 471 | 0.19129 | 0.49534 | 6.00000 |
| Eurasian Lynx | 3093 | 35 | 0.01268 | 0.04997 | 0.33333 |

DOI: https://doi.org/10.7554/eLife.44937.021

**Appendix 1—table 2.** Matrix of the foraging-related traits used for the estimation of the functional diversity index (FDis) of the entire ungulate community: body mass (based on *Jedrzejewska et al., 1994*) and *Borowik et al., 2016*), diet type (based on *Hofmann, 1989*); B: browsers-concentrate selectors, BG: browsers/grazers-intermediate types, G: grazers-grass roughage eaters, O: omnivores) and gut type (R: ruminants, NR: non-ruminants).

| | Roe deer | Wild boar | Red deer female | Red deer male | Moose | European bison |
|---|---|---|---|---|---|---|
| Body mass | 20 | 80 | 90 | 150 | 200 | 400 |
| Diet type | B | O | BG | BG | B | G |
| Gut type | R | NR | R | R | R | R |

DOI: https://doi.org/10.7554/eLife.44937.022

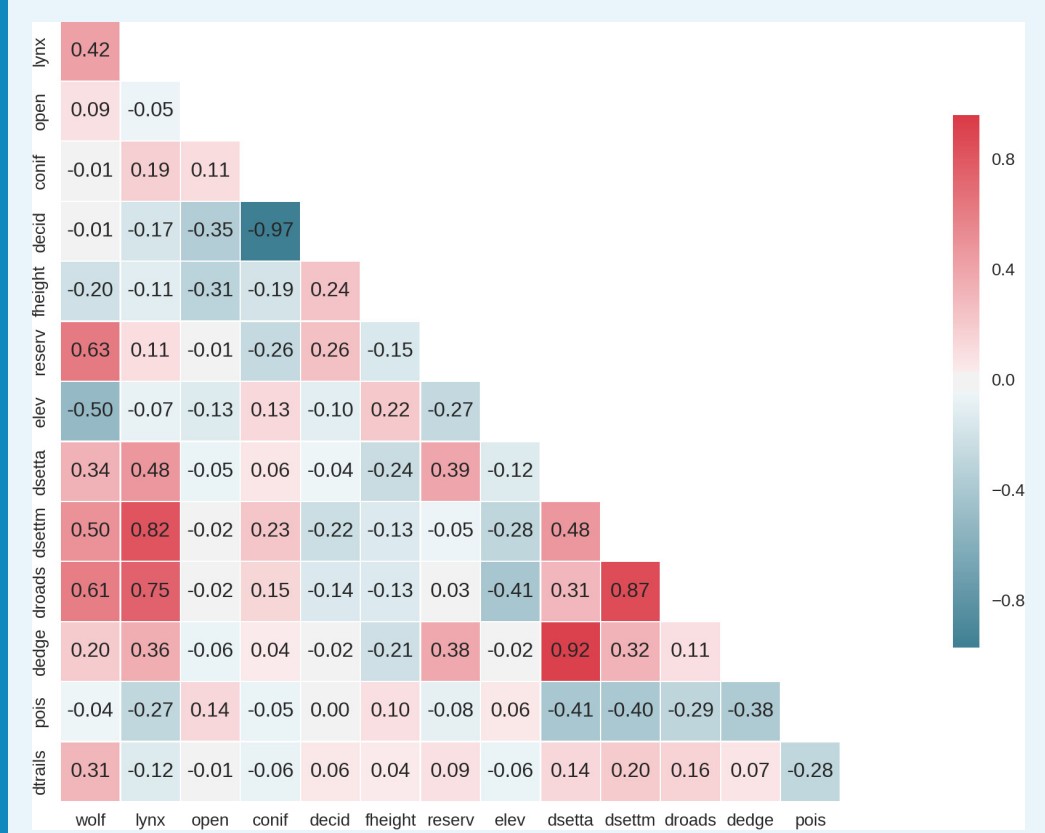

**Appendix 1—Figure 1.** Heatmap of pairwise Pearson correlation values of all candidate landscape-scale covariates for the ungulates models; *wolf*: predicted use of the landscape by the wolf, *lynx:* predicted use of the landscape by the lynx, *open*: landscape openness, *conif*: % share of coniferous tree species, *decid*: % share of deciduous tree species, *fheight*: forest height, *reserv*: density of protected areas, *elev*: elevation, *dsetta*: distance to all settlements, *dsettm*: distance to major settlements, *droads*: distance to main roads, *dedge*: distance to forest edge, *pois*: density of tourist infrastructure. When selecting covariates for the final models we ensured that the Pearson correlation value for all pairs of included covariates was lower than 0.7.

DOI: https://doi.org/10.7554/eLife.44937.023

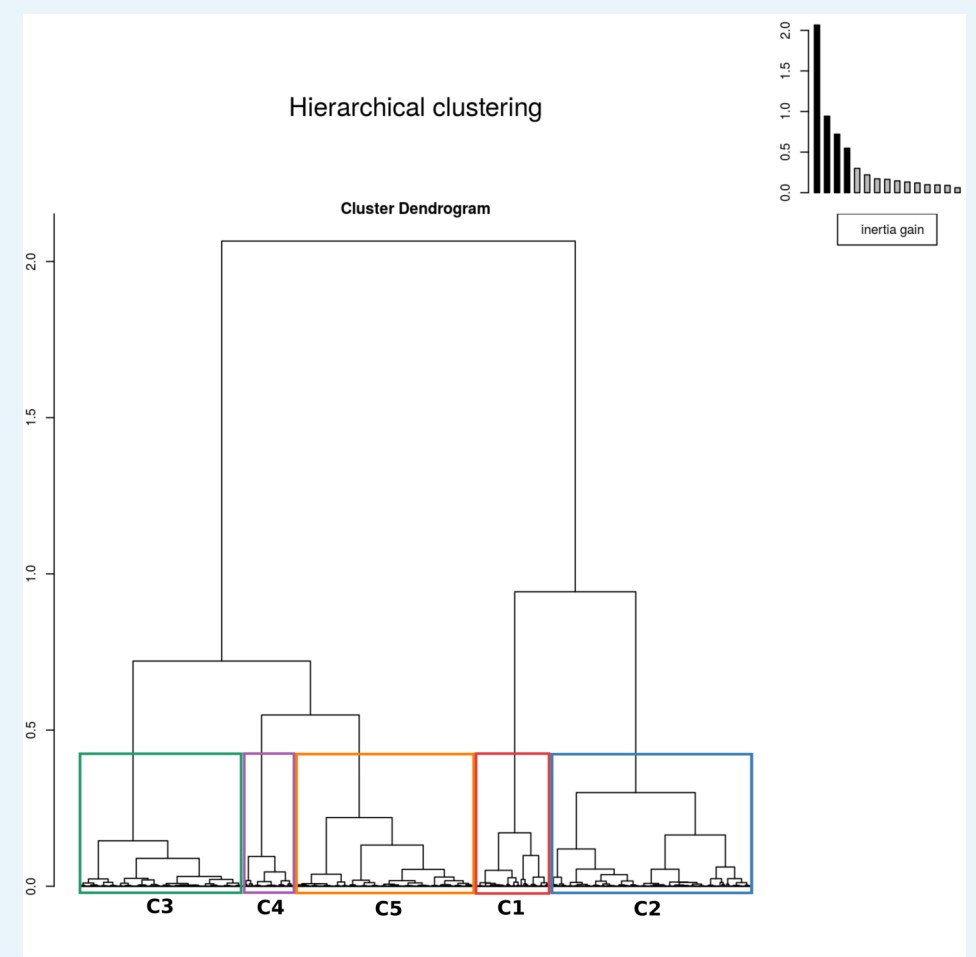

**Appendix 1—Figure 2.** The results of hierarchical clustering on the principal component analysis. The five clusters were identified by assessing the inertia (i.e. change in within cluster homogeneity) gained by cutting the tree at different levels and the ecological interpretability of the resulting clusters. The colors of the clusters correspond to those in *Figure 10* in the main text.

DOI: https://doi.org/10.7554/eLife.44937.024

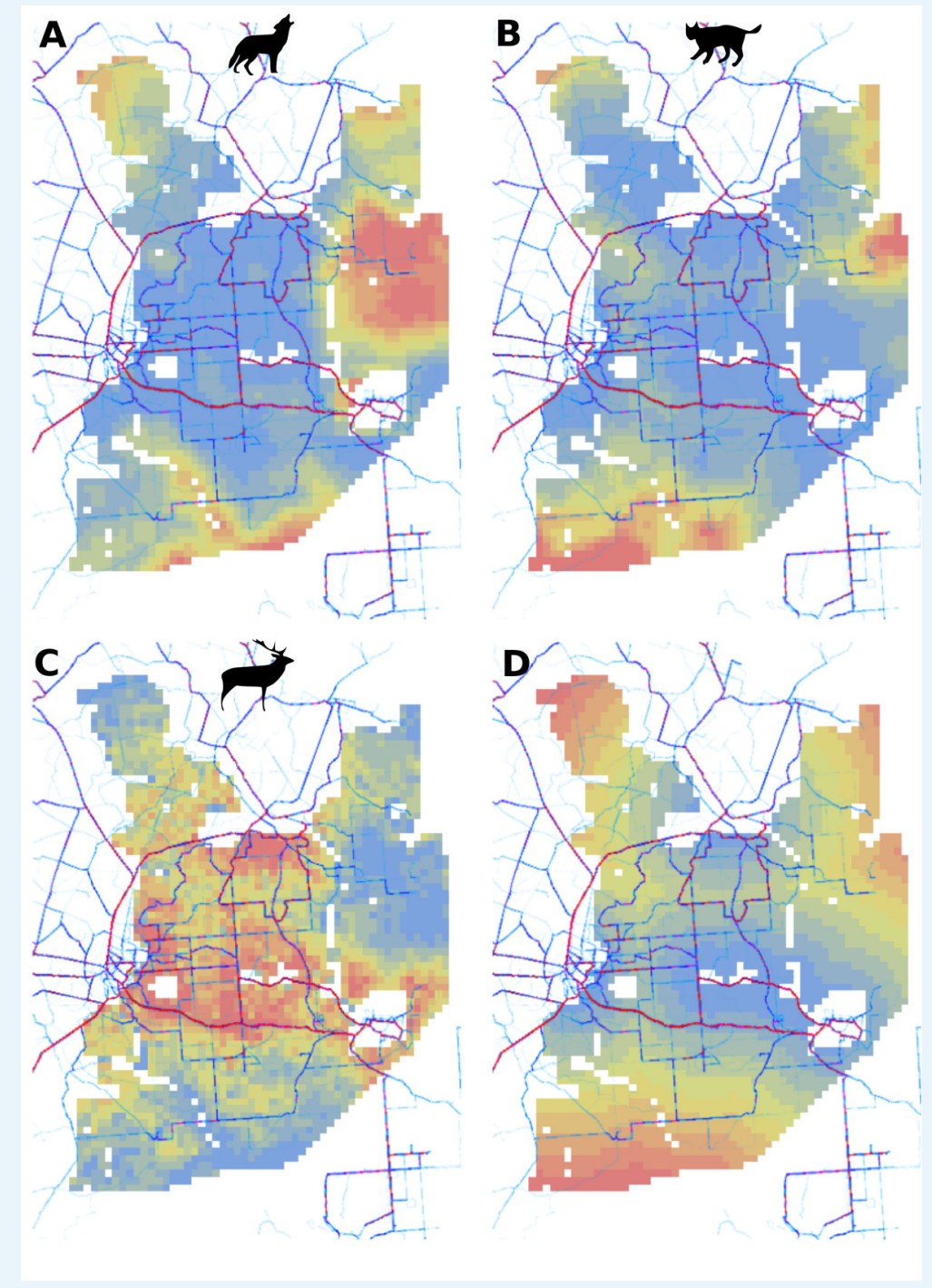

**Appendix 1—Figure 3.** Spatial patterns of human activity in Białowieża Forest derived from the publicly available world-wide data provided by STRAVA (https://www.strava.com/heatmap) combined with (**A**) predicted use of the landscape by the wolf, (**B**) predicted use of the landscape by the lynx, (**C**) predicted use of landscape by the red deer (both sexes) and (**D**) distance to major settlements used in the models for both large carnivores. The dark-red thick lines indicate high human activity and the light-blue thin lines indicate low human activity. For (A), (B) and (C) the color gradient from blue to red corresponds to the gradient from low to high use of the landscape by the presented species.

DOI: https://doi.org/10.7554/eLife.44937.025

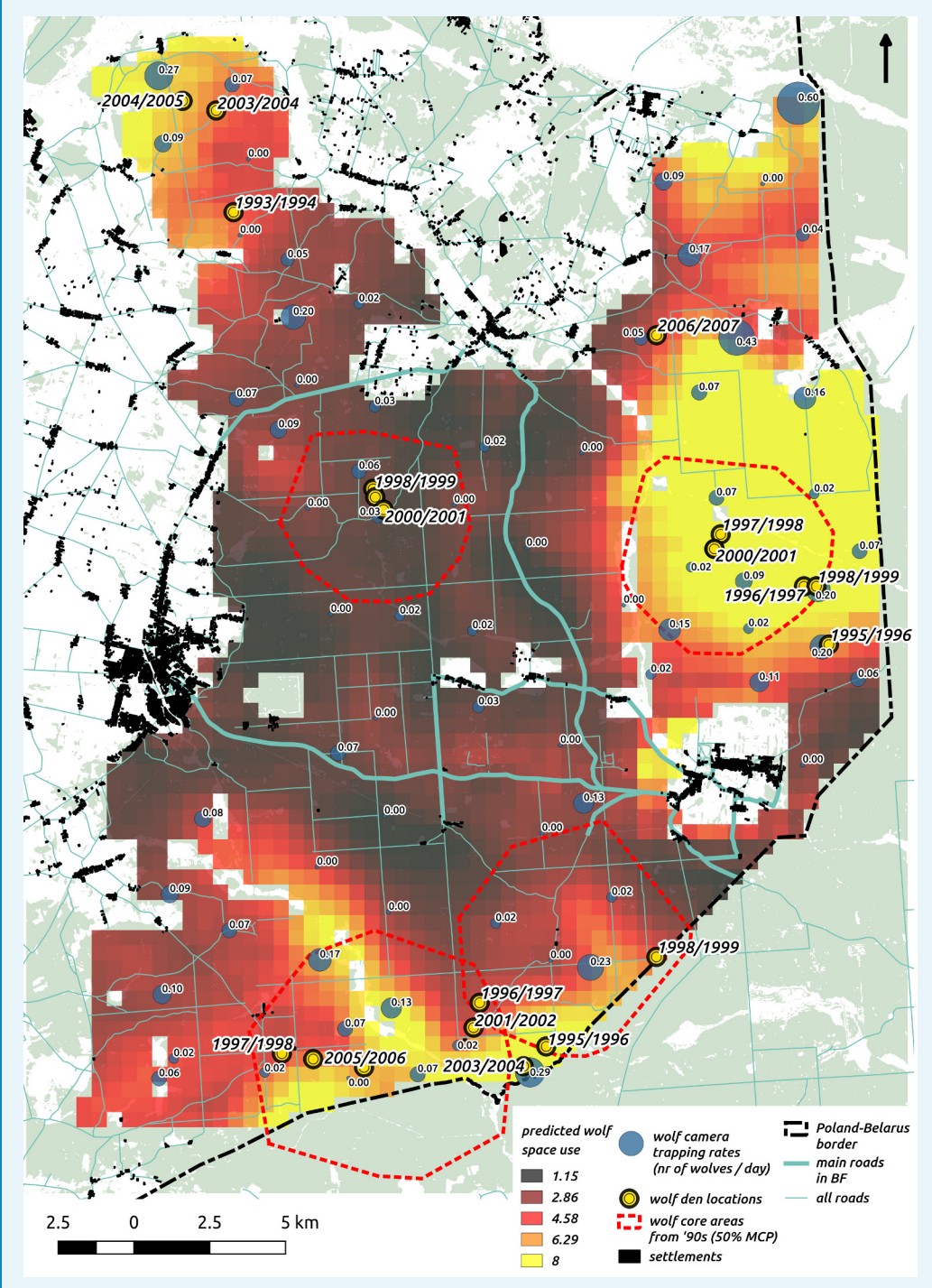

**Appendix 1—Figure 4.** Comparison of predictions of wolf space use with existing radio-tracking data from collared wolves in Białowieża Forest (period 1994–1999, *Jędrzejewski et al., 2007*) and observations of known den locations (period 1993–2007, MRI PAS unpublished data). In the background we plotted the raster map of wolf space use predicted by our model together with the raw camera trapping data ('bubbles' with trapping rates at each location). On top of it we plotted two datasets: 1) the locations of the core areas (dashed red circles) of the annual territories of the four wolf packs present in the area (50% MCP based on telemetry) and 2) den locations.

DOI: https://doi.org/10.7554/eLife.44937.026

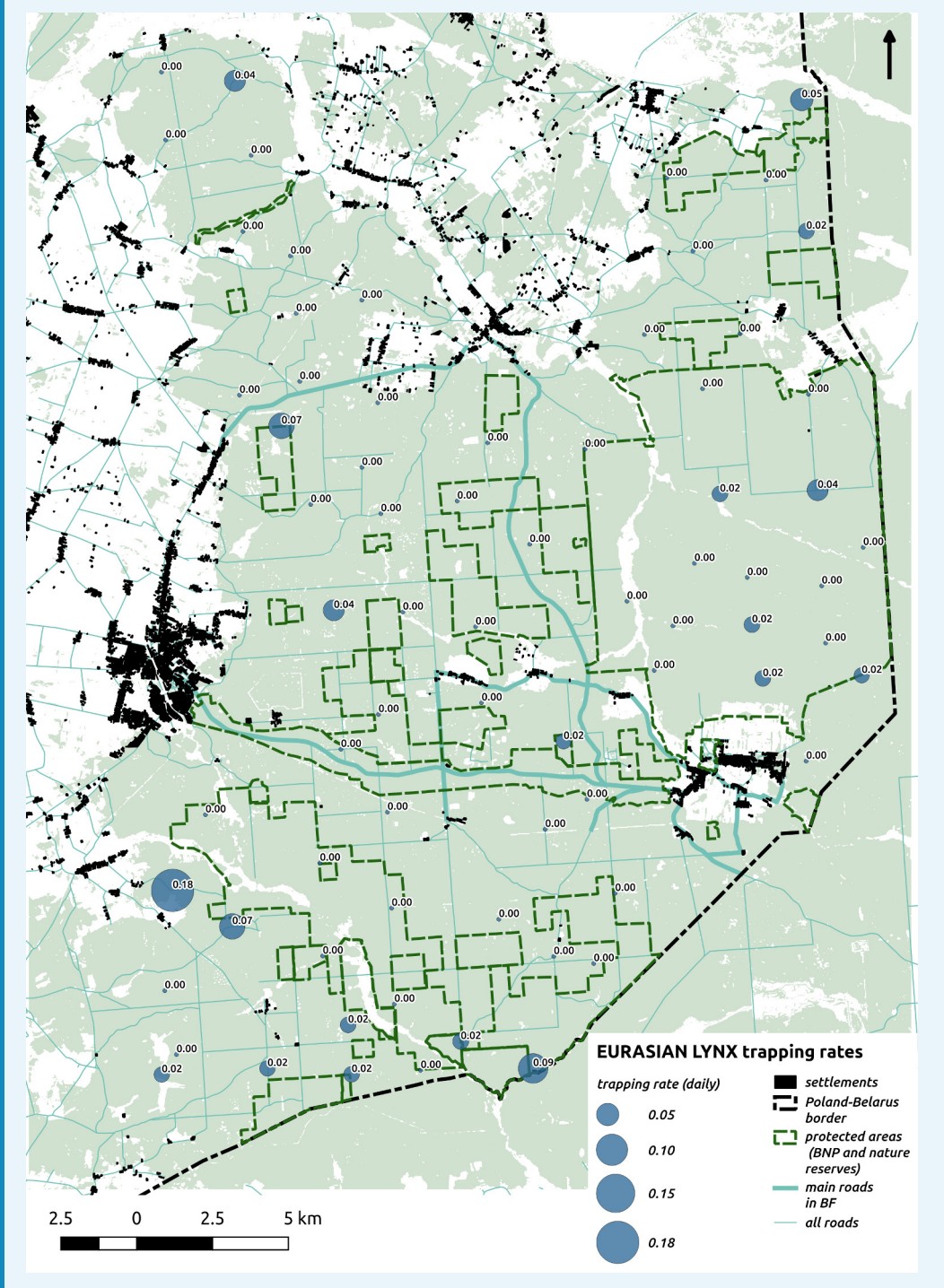

**Appendix 1—Figure 5.** Bubble plot presenting raw camera trapping data for lynx collected during the carnivore survey. Each bubble is a camera trap location and its size is proportional to the daily trapping rate (i.e. no. of individuals observed/no. of days).

DOI: https://doi.org/10.7554/eLife.44937.027

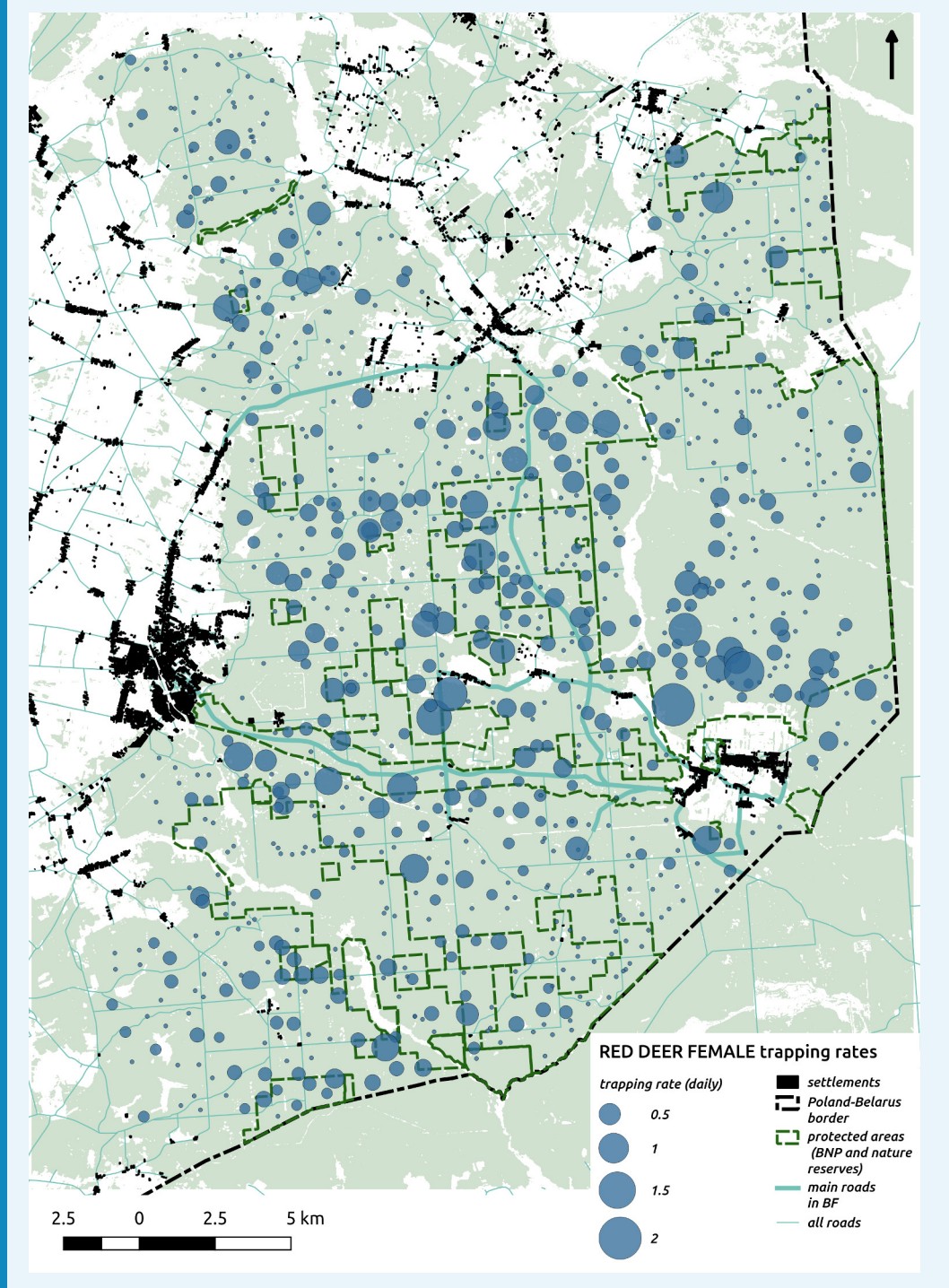

**Appendix 1—Figure 6.** Bubble plot presenting raw camera trapping data for red deer female collected during the ungulate survey. Each bubble is a camera trap location and its size is proportional to the daily trapping rate (i.e. no. of individuals observed/no. of days).

DOI: https://doi.org/10.7554/eLife.44937.028

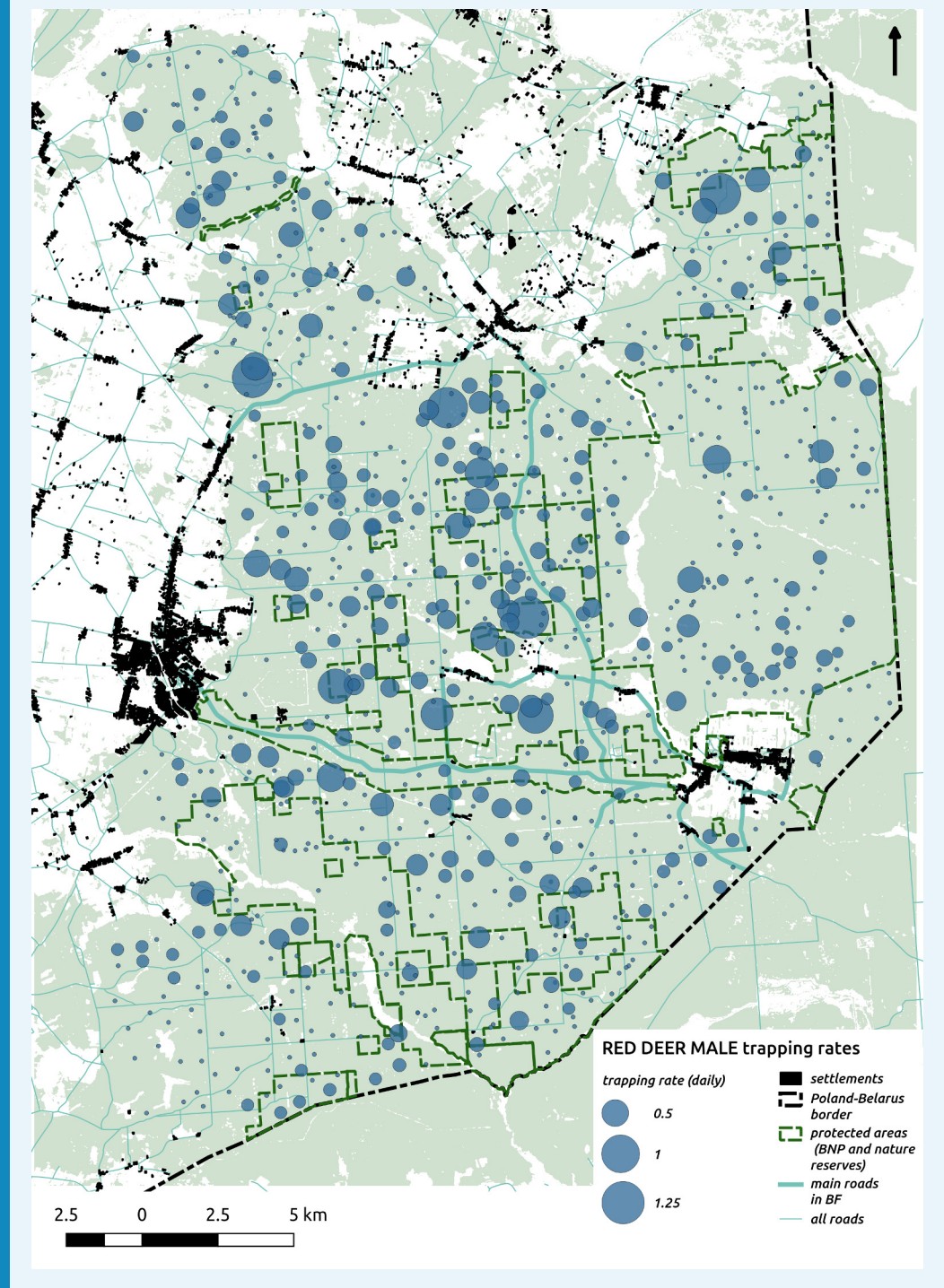

**Appendix 1—Figure 7.** Bubble plot presenting raw camera trapping data for red deer male collected during the ungulate survey. Each bubble is a camera trap location and its size is proportional to the daily trapping rate (i.e. no. of individuals observed/no. of days).

DOI: https://doi.org/10.7554/eLife.44937.029

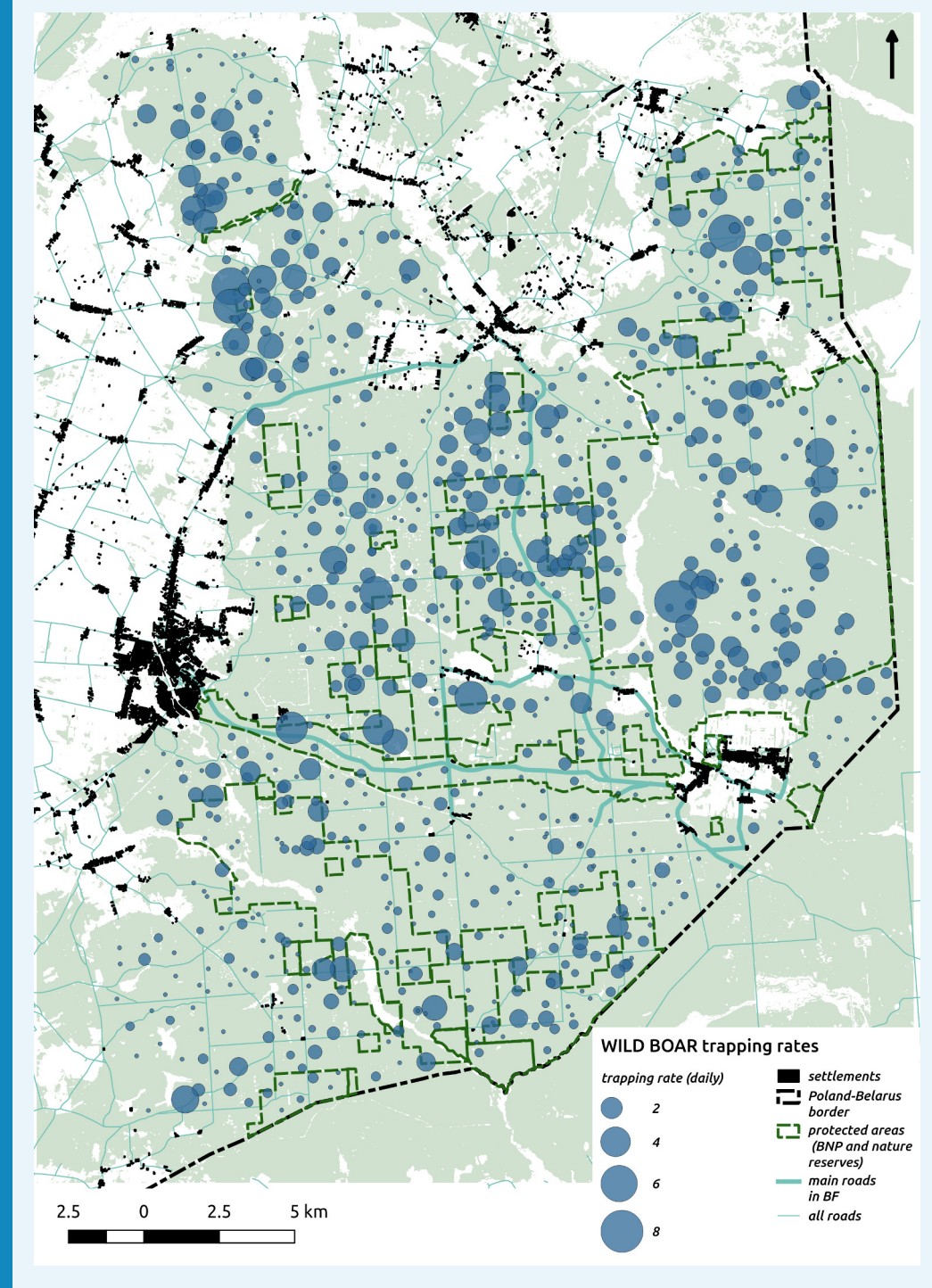

**Appendix 1—Figure 8.** Bubble plot presenting raw camera trapping data for wild boar collected during the ungulate survey. Each bubble is a camera trap location and its size is proportional to the daily trapping rate (i.e. no. of individuals observed/no. of days).

DOI: https://doi.org/10.7554/eLife.44937.030

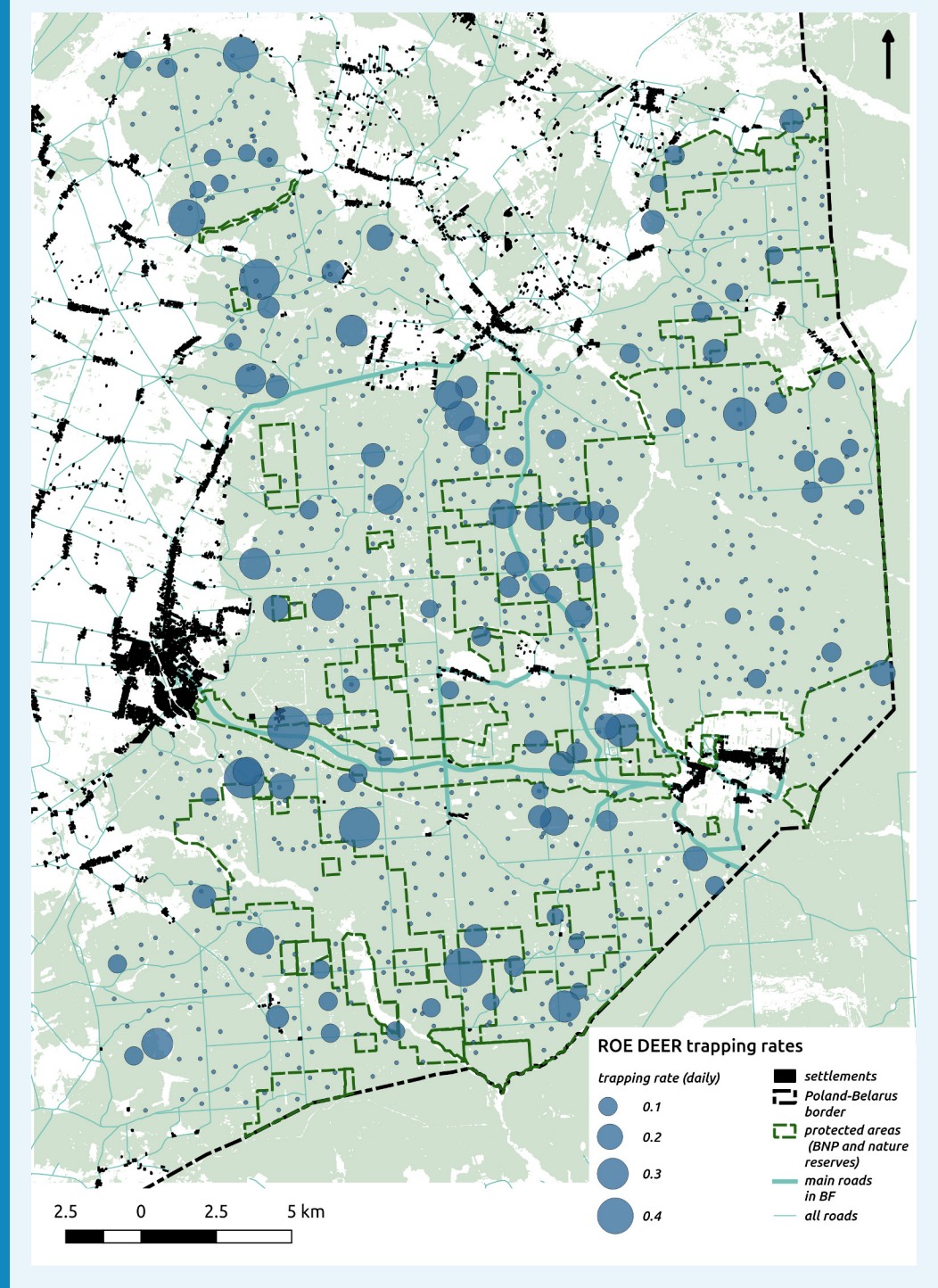

**Appendix 1—Figure 9.** Bubble plot presenting raw camera trapping data for roe deer collected during the ungulate survey. Each bubble is a camera trap location and its size is proportional to the daily trapping rate (i.e. no. of individuals observed/no. of days).

DOI: https://doi.org/10.7554/eLife.44937.031

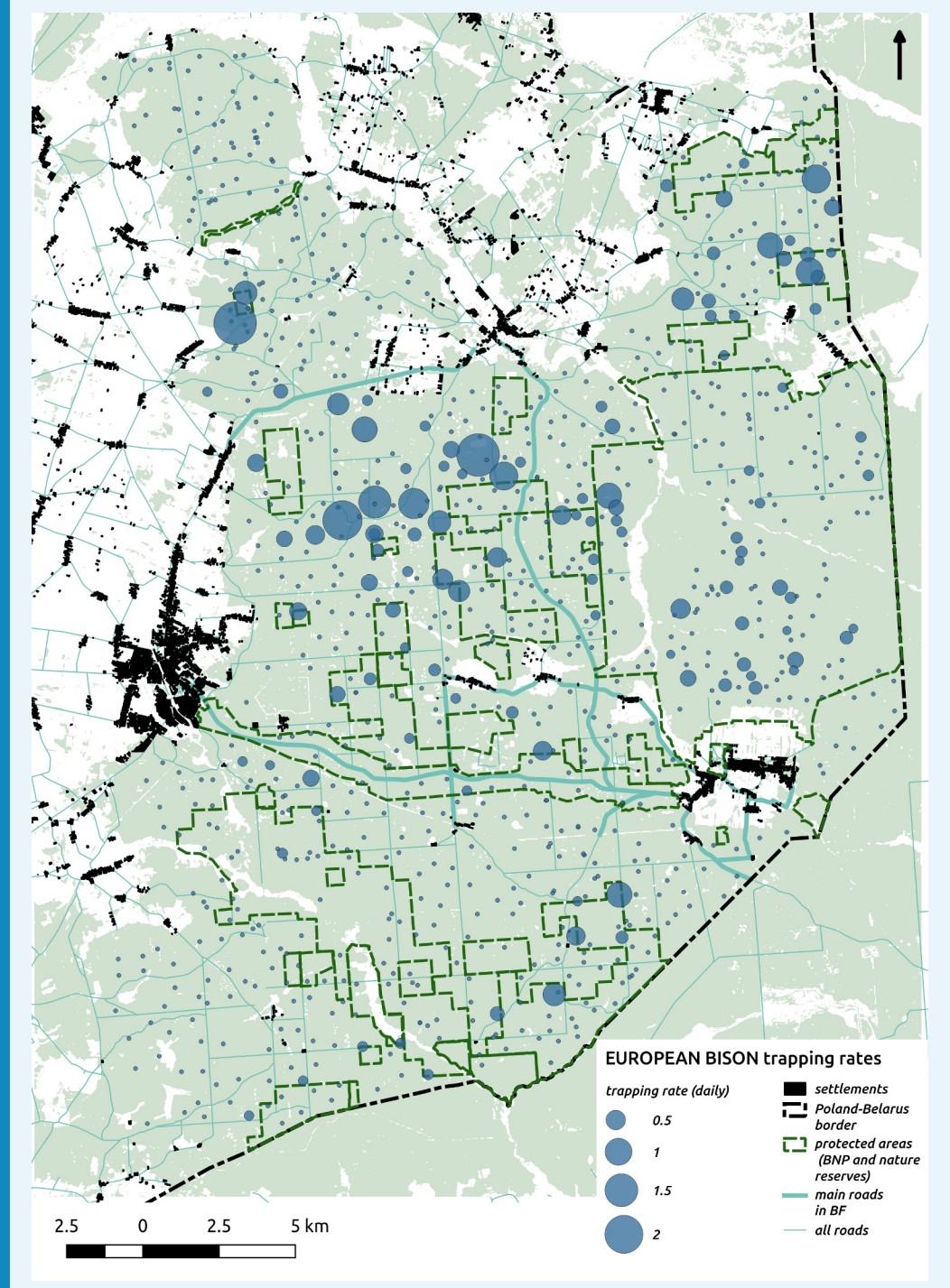

**Appendix 1—Figure 10.** Bubble plot presenting raw camera trapping data for bison collected during the ungulate survey. Each bubble is a camera trap location and its size is proportional to the daily trapping rate (i.e. no. of individuals observed/no. of days).

DOI: https://doi.org/10.7554/eLife.44937.032

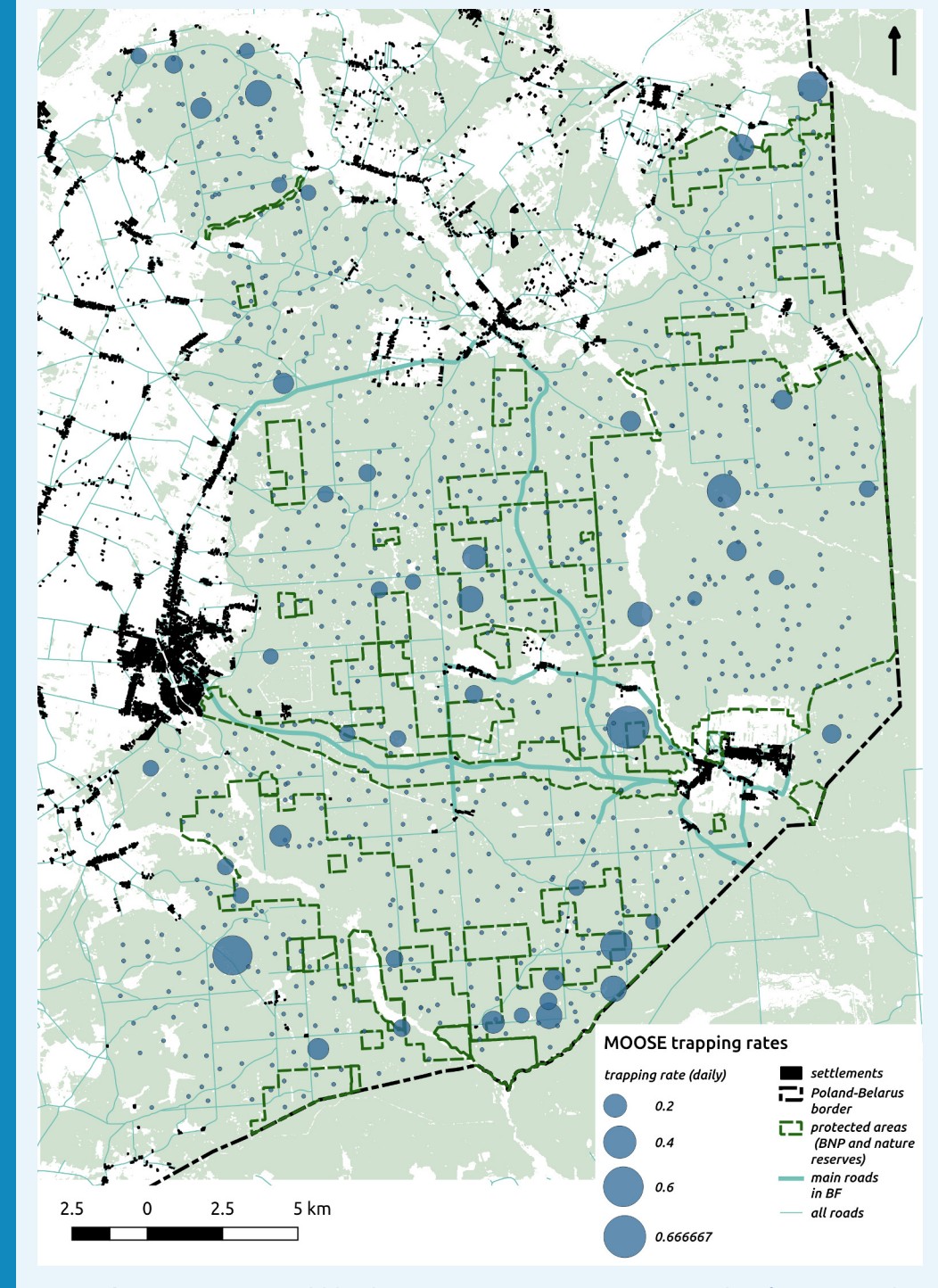

**Appendix 1—Figure 11.** Bubble plot presenting raw camera trapping data for moose collected during the ungulate survey. Each bubble is a camera trap location and its size is proportional to the daily trapping rate (i.e. no. of individuals observed/no. of days).

DOI: https://doi.org/10.7554/eLife.44937.033

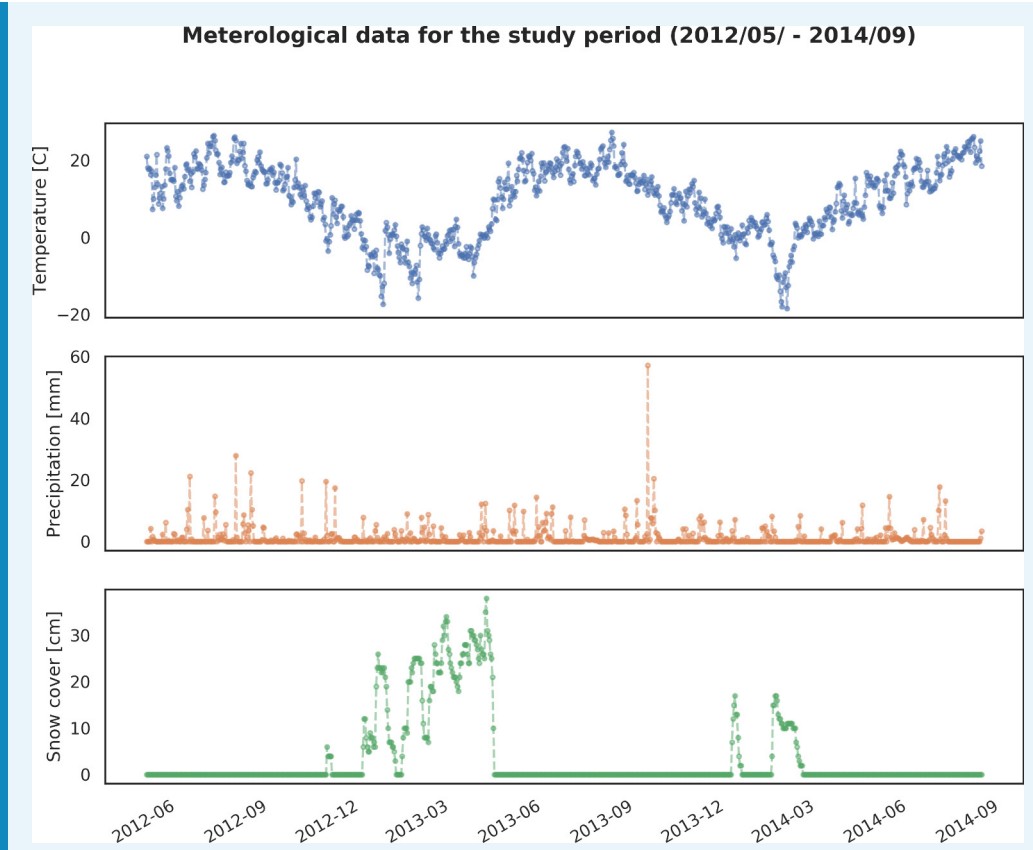

**Appendix 1—Figure 12.** Basic meteorological data for the study period (05/2012 – 09/2014): 1) averaged daily temperature [C], 2) precipitation [mm] and 3) snow cover [cm].

DOI: https://doi.org/10.7554/eLife.44937.034

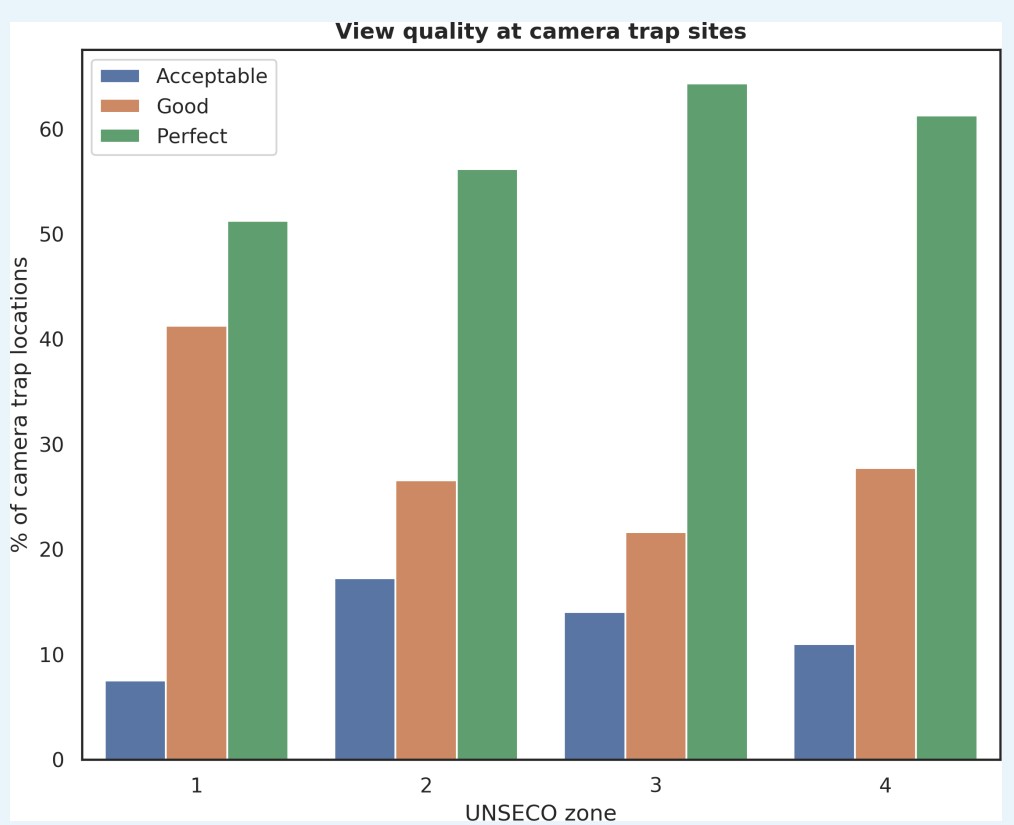

**Appendix 1—Figure 13.** Distribution of camera trap locations grouped by quality of view in front of a camera and different UNESCO zones in Białowieża Forest (1: Strict Reserve of BNP, 2: BNP and nature reserves, 3: valuable unprotected forest stands, 4: managed forest stands). Each camera location was manually tagged with a 4-level categorical label ('Exclude', 'Acceptable', 'Good', 'Perfect') describing the quality of view in front of the camera. Locations labeled with the 'Exclude' were not included in further analysis and are not shown on this figure.

DOI: https://doi.org/10.7554/eLife.44937.035

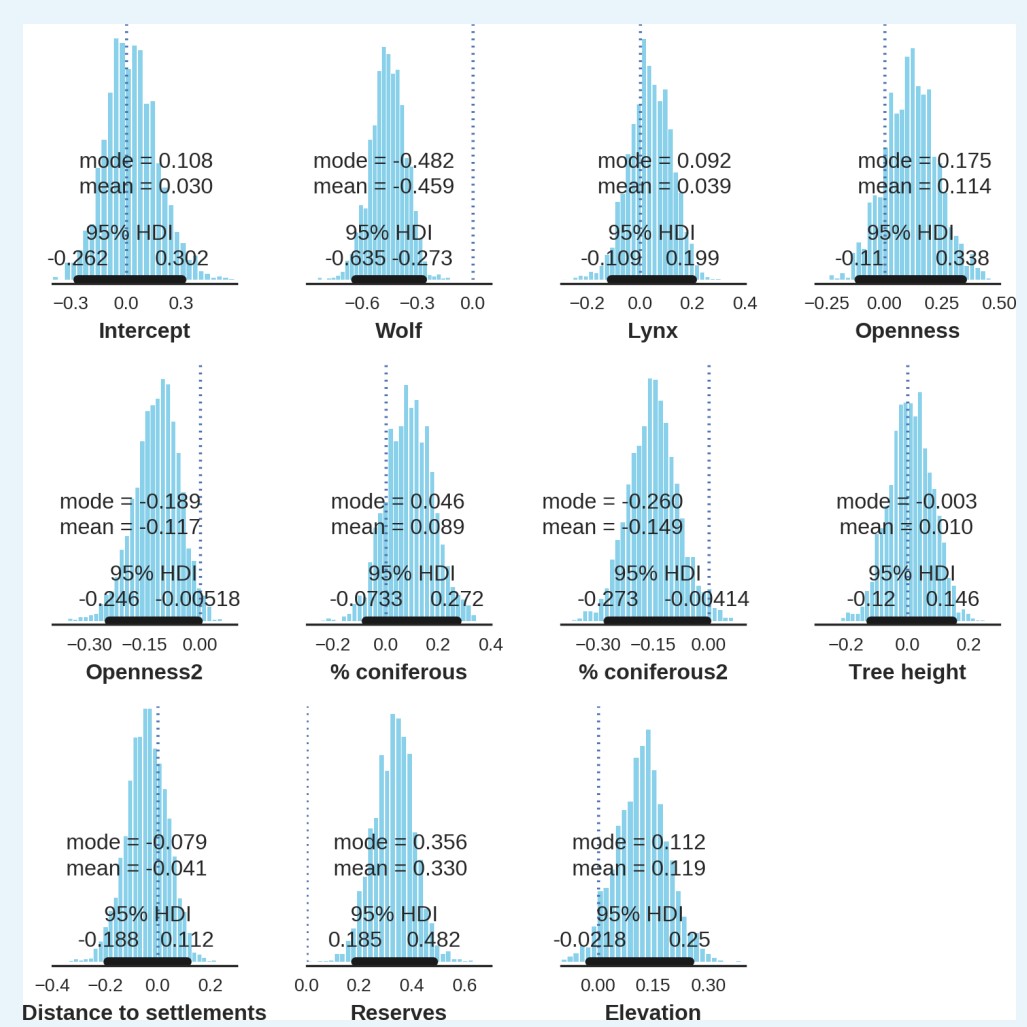

**Appendix 1—Figure 14.** Red deer female – landscape use; the posterior distributions of all parameters.

DOI: https://doi.org/10.7554/eLife.44937.036

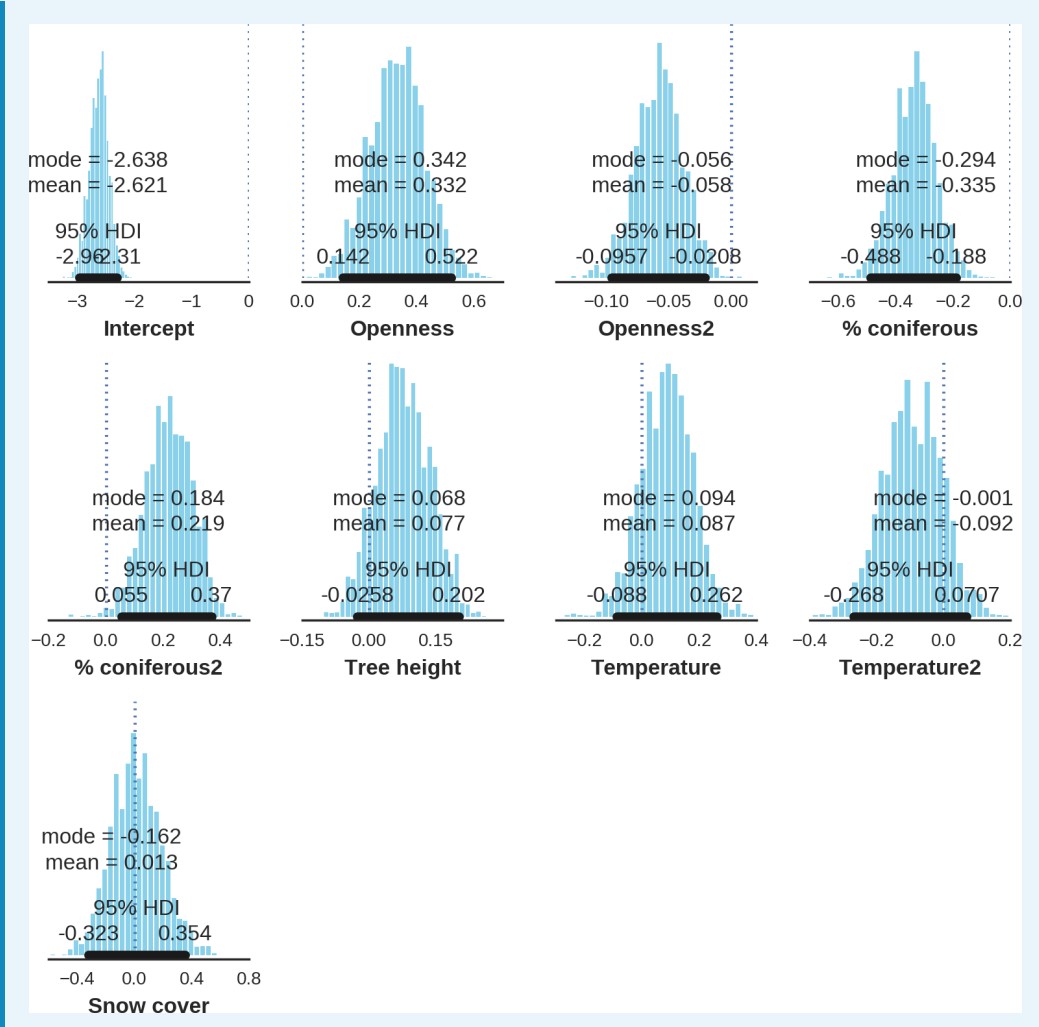

**Appendix 1—Figure 15.** Red deer female – detection rate; the posterior distributions of all parameters.

DOI: https://doi.org/10.7554/eLife.44937.037

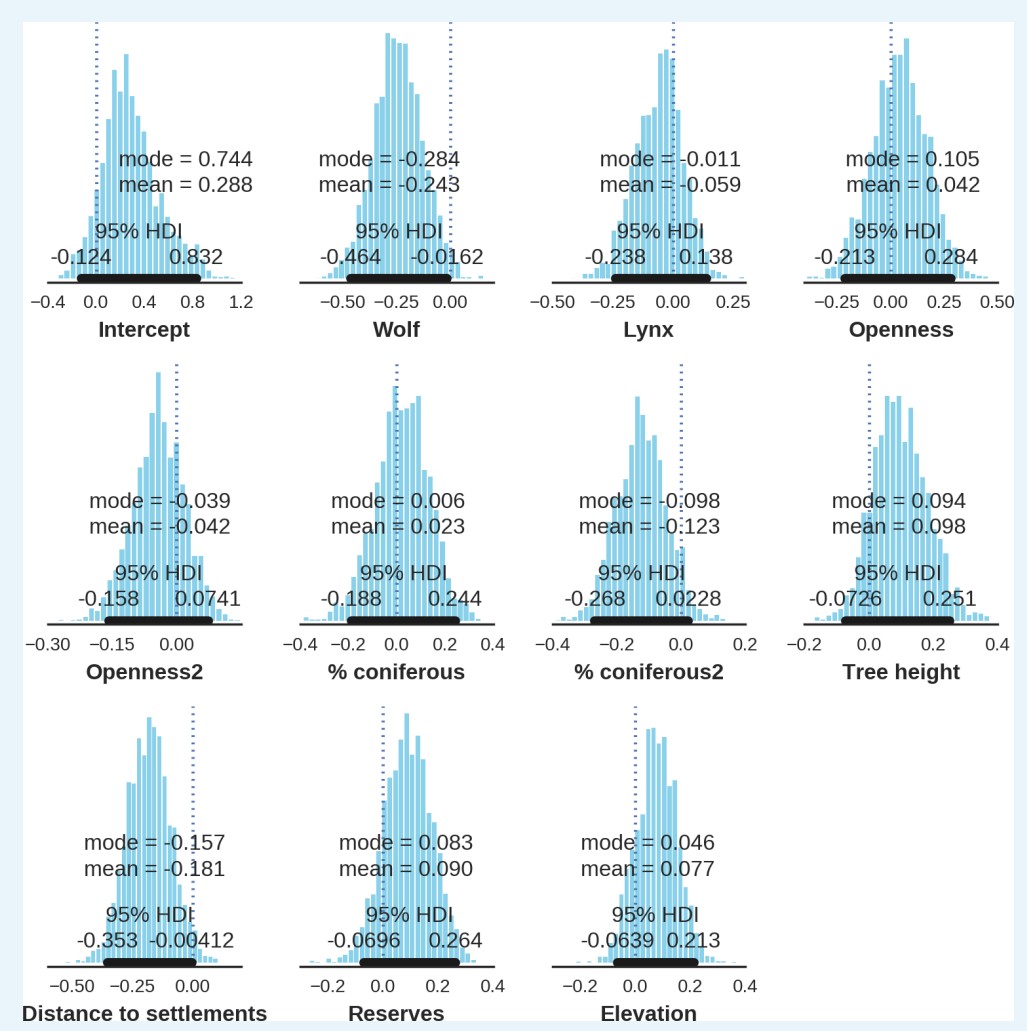

**Appendix 1—Figure 16.** Red deer male – landscape use; the posterior distributions of all parameters.

DOI: https://doi.org/10.7554/eLife.44937.038

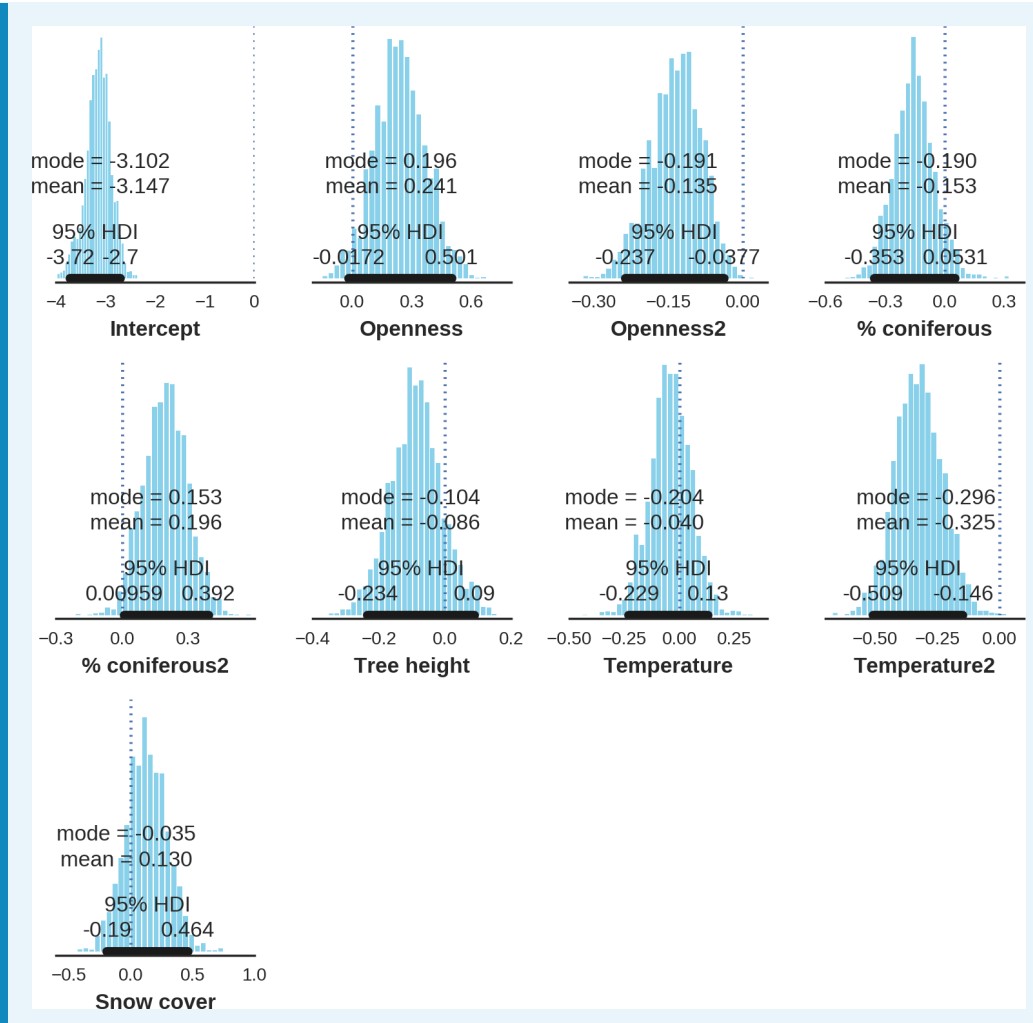

**Appendix 1—Figure 17.** Red deer male – detection rate; the posterior distributions of all parameters.

DOI: https://doi.org/10.7554/eLife.44937.039

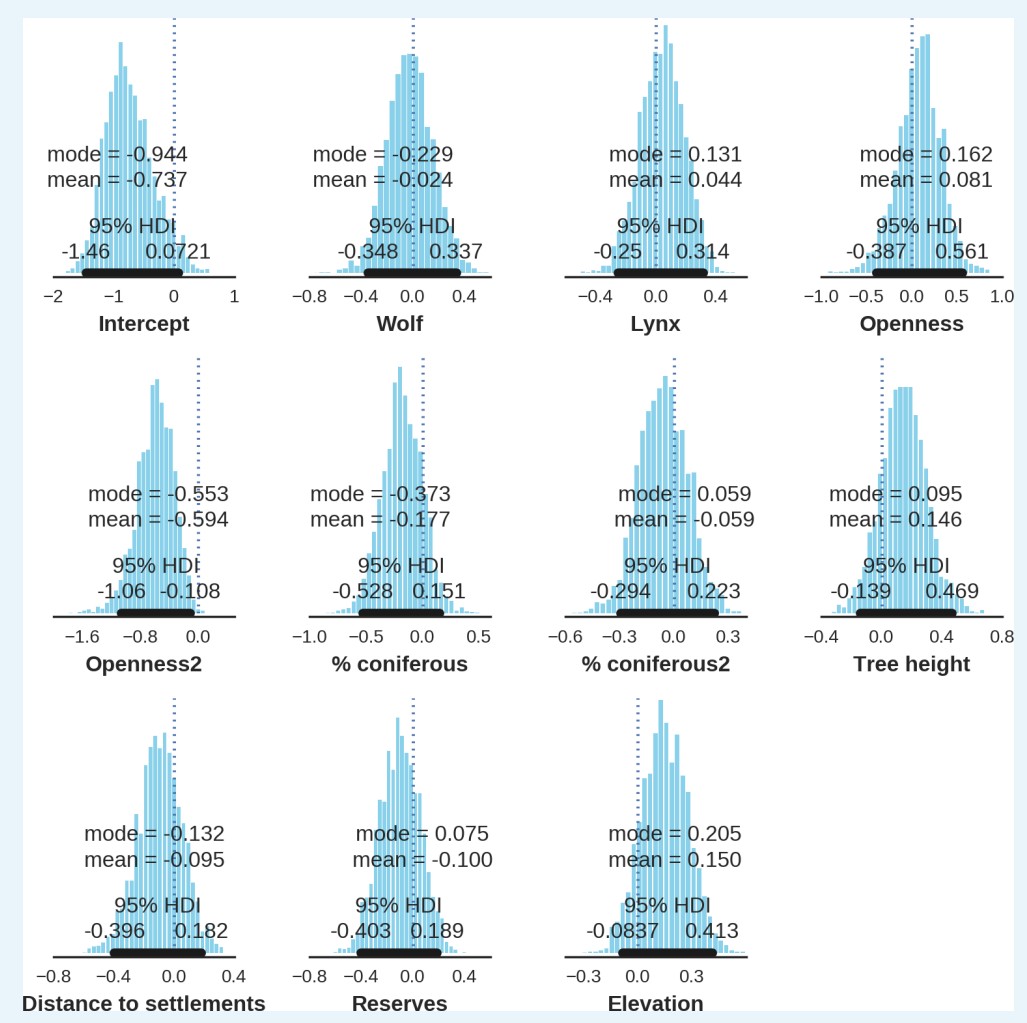

**Appendix 1—Figure 18.** Roe deer – landscape use; the posterior distributions of all parameters.

DOI: https://doi.org/10.7554/eLife.44937.040

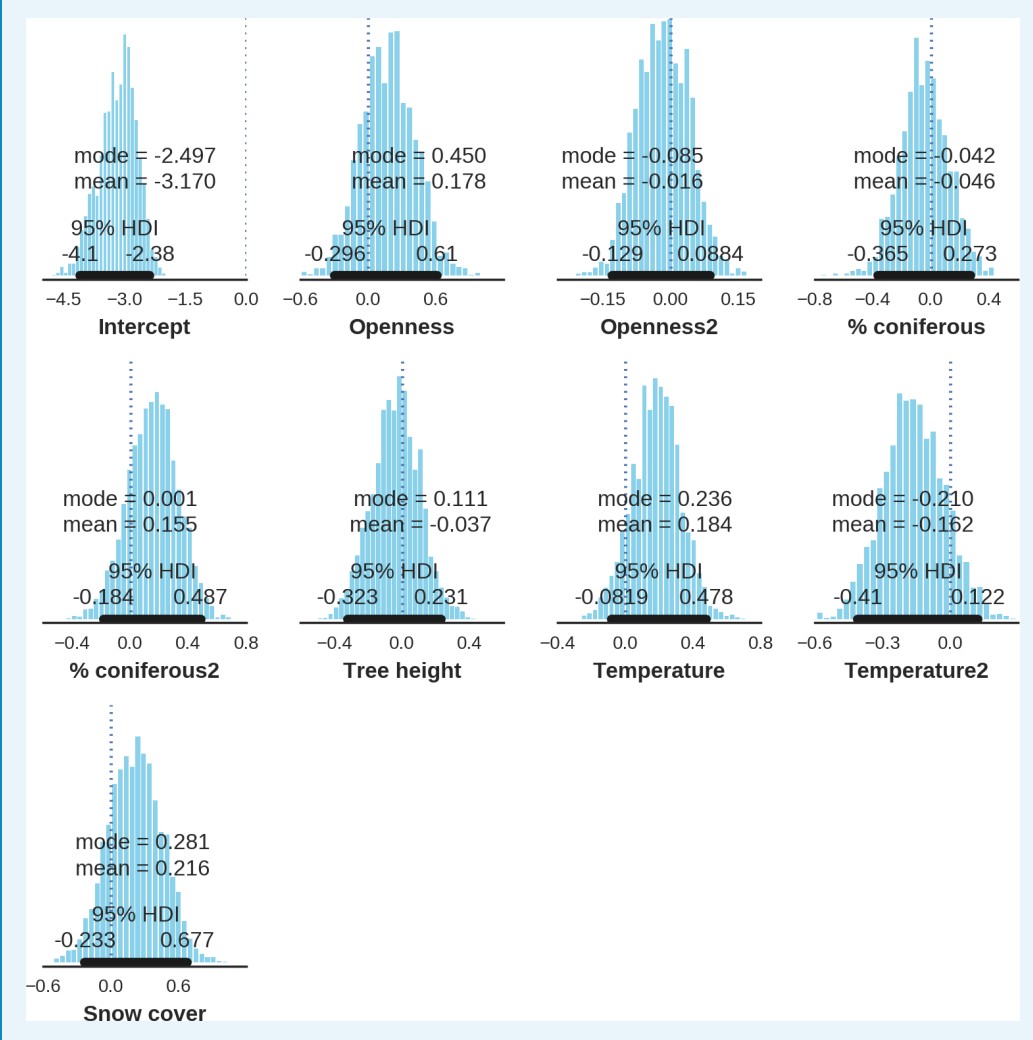

**Appendix 1—Figure 19.** Roe deer – detection rate; the posterior distributions of all parameters.

DOI: https://doi.org/10.7554/eLife.44937.041

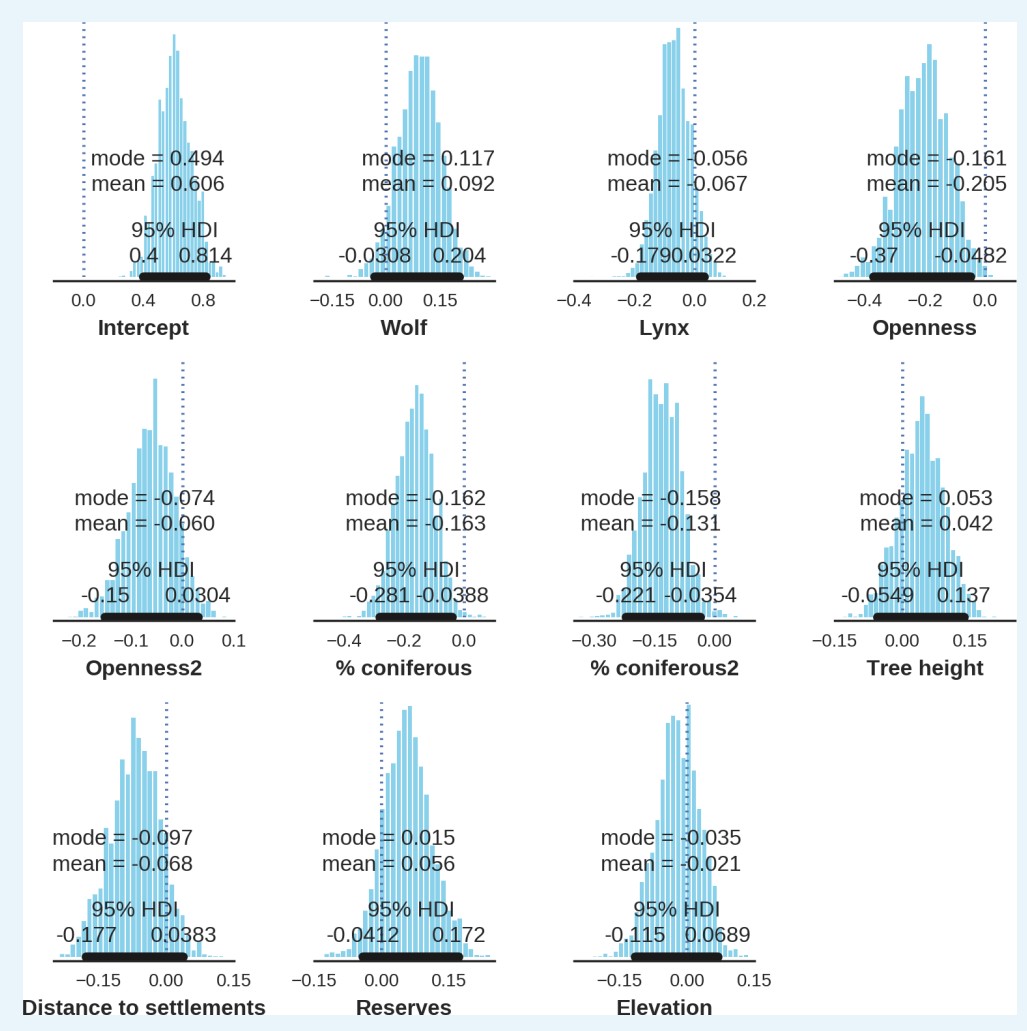

**Appendix 1—Figure 20.** Wild boar – landscape use; the posterior distributions of all parameters.

DOI: https://doi.org/10.7554/eLife.44937.042

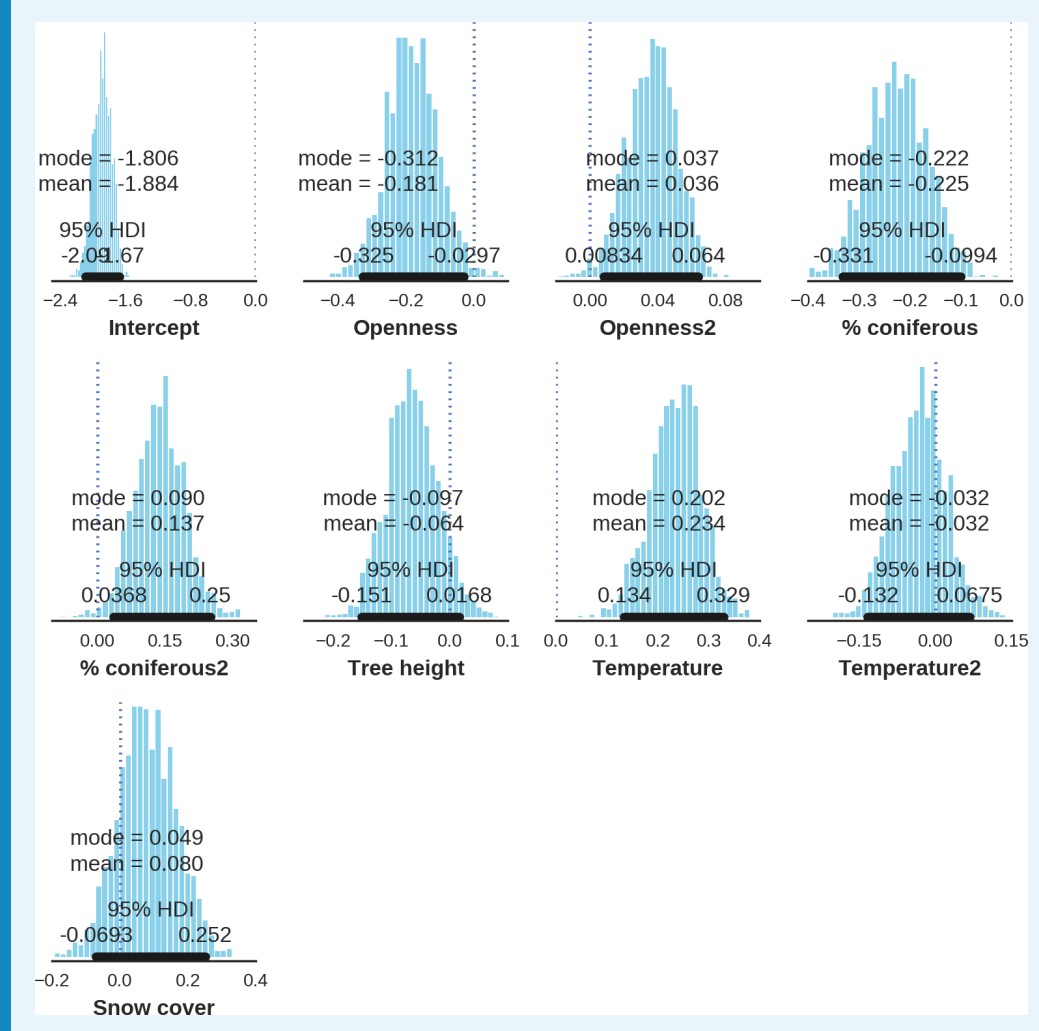

**Appendix 1—Figure 21.** Wild boar – detection rate; the posterior distributions of all parameters.

DOI: https://doi.org/10.7554/eLife.44937.043

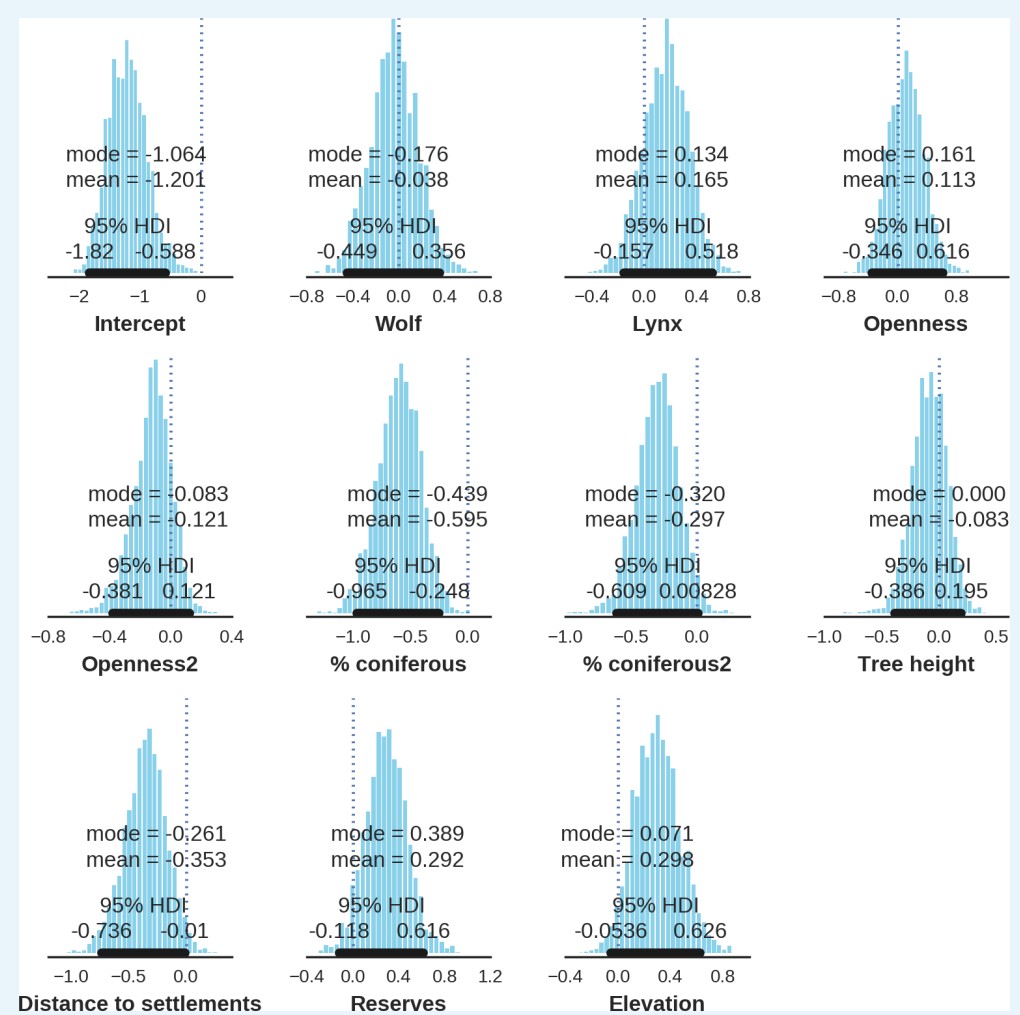

**Appendix 1—Figure 22.** European bison – landscape use; the posterior distributions of all parameters.

DOI: https://doi.org/10.7554/eLife.44937.044

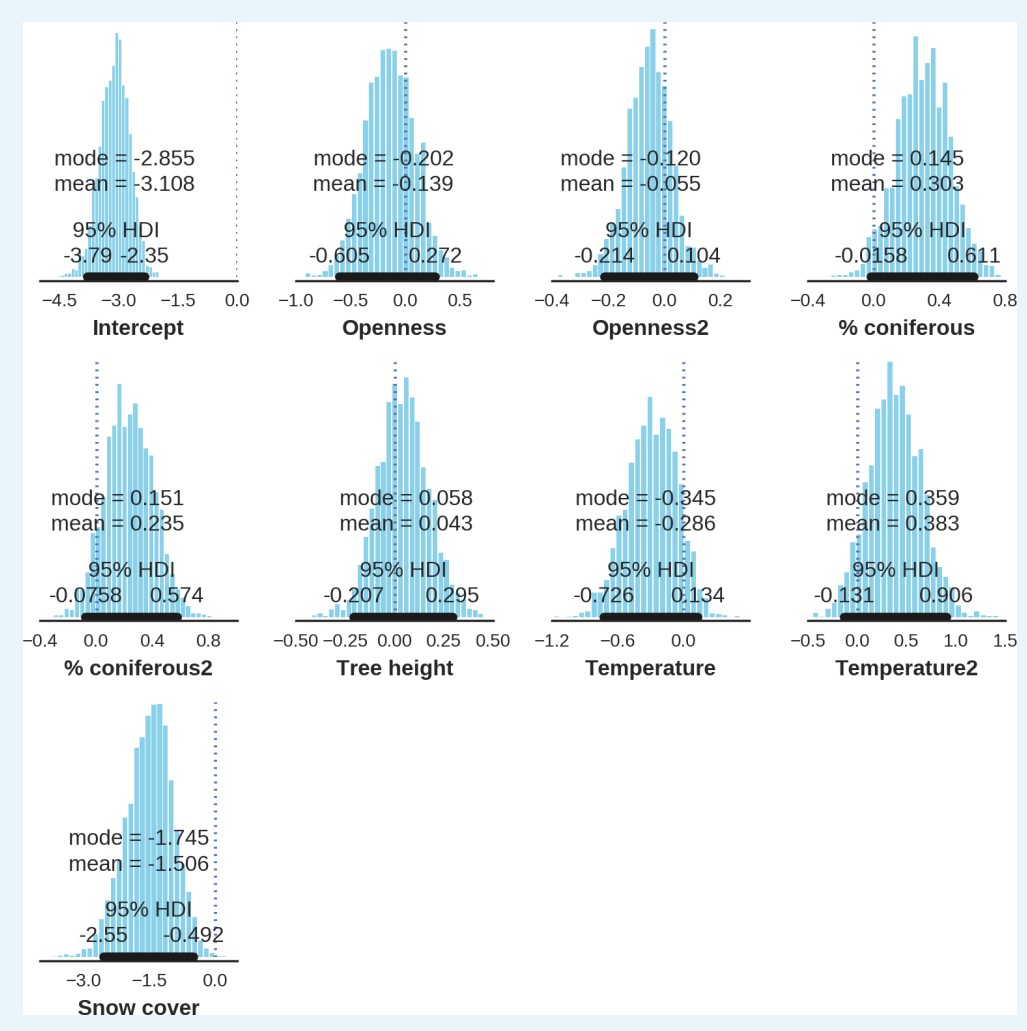

**Appendix 1—Figure 23.** European bison – detection rate; the posterior distributions of all parameters.

DOI: https://doi.org/10.7554/eLife.44937.045

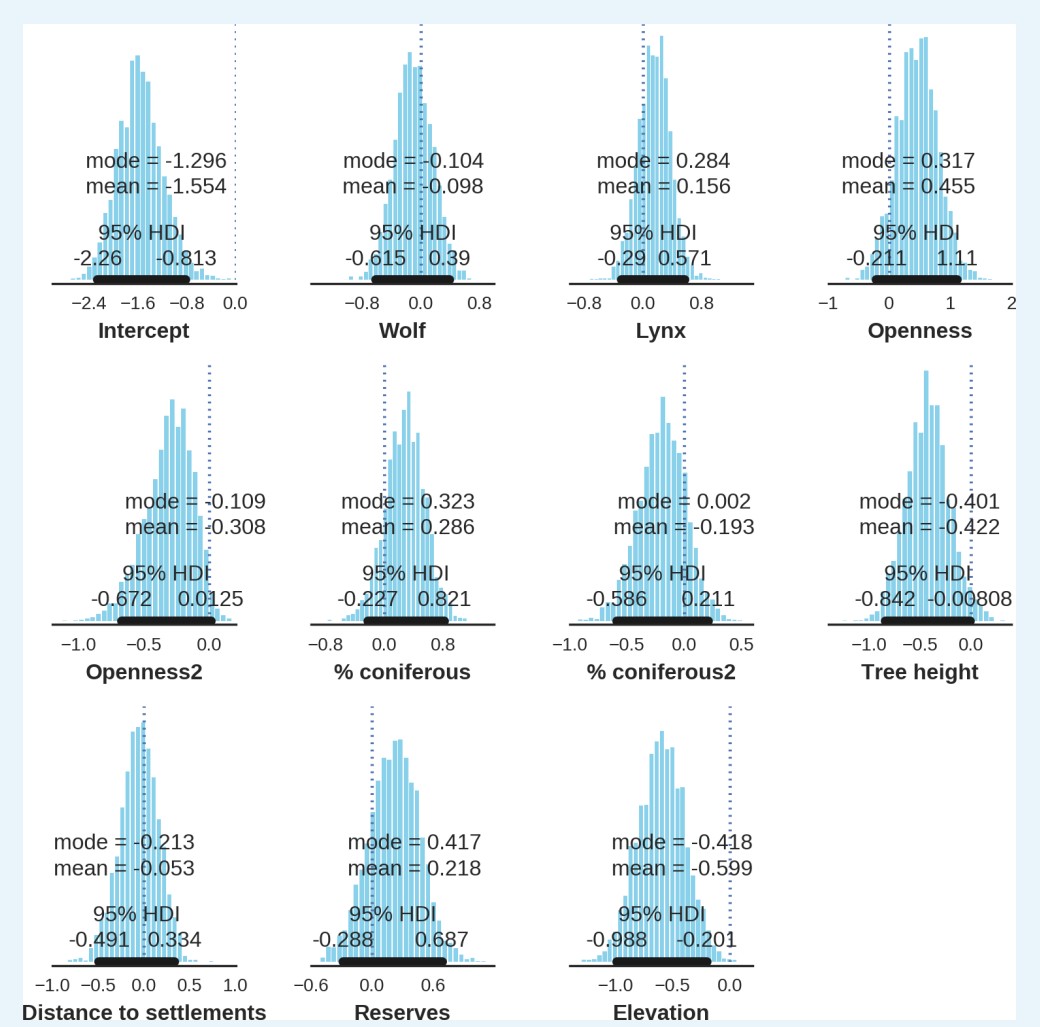

**Appendix 1—Figure 24.** Moose – landscape use; the posterior distributions of all parameters.

DOI: https://doi.org/10.7554/eLife.44937.046

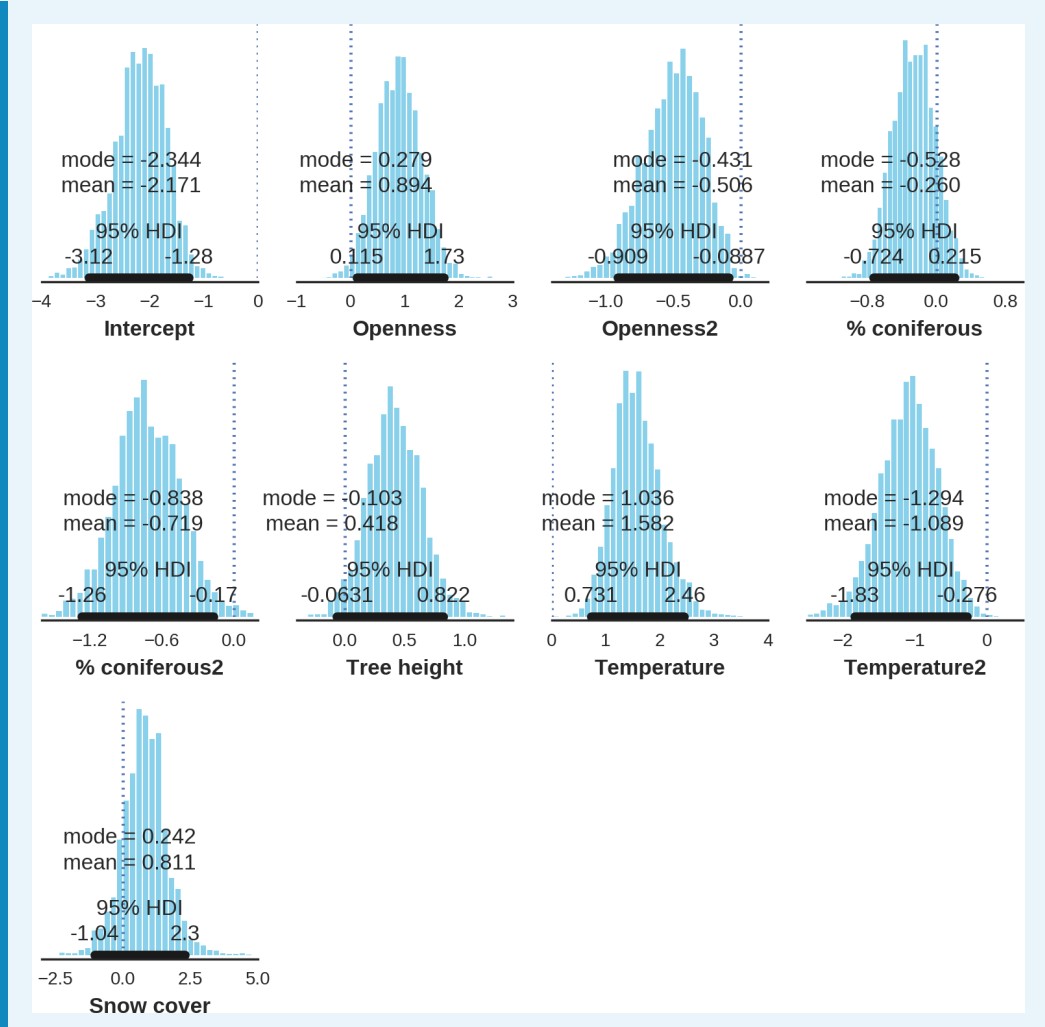

**Appendix 1—Figure 25.** Moose – detection rate; the posterior distributions of all parameters.

DOI: https://doi.org/10.7554/eLife.44937.047

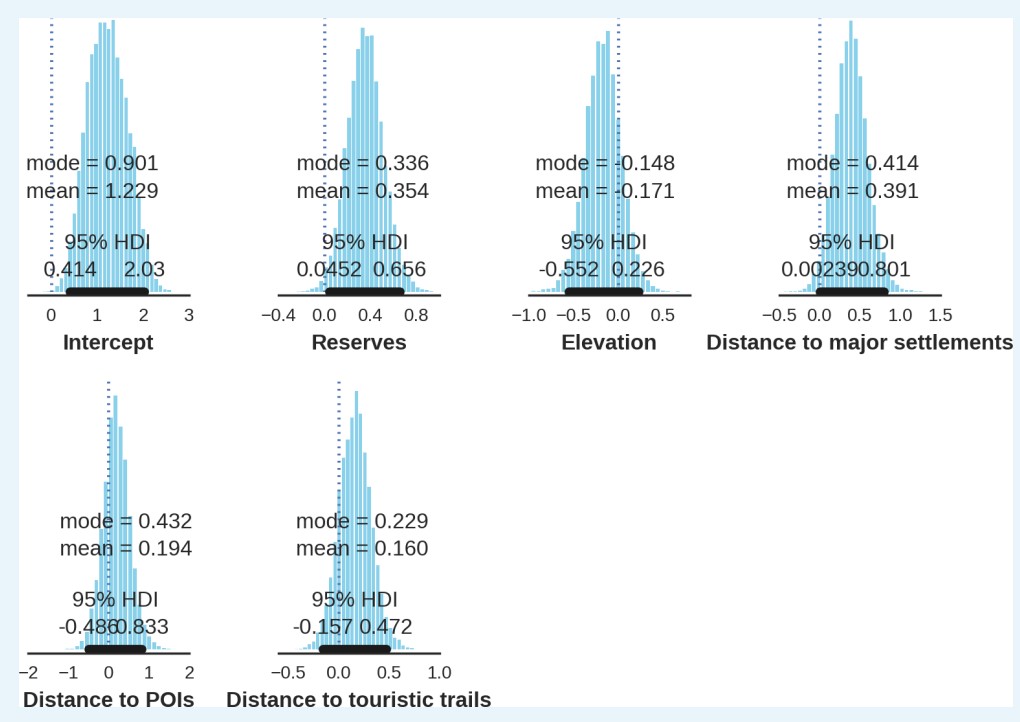

**Appendix 1—Figure 26.** Wolf – landscape use; the posterior distributions of all parameters.

DOI: https://doi.org/10.7554/eLife.44937.048

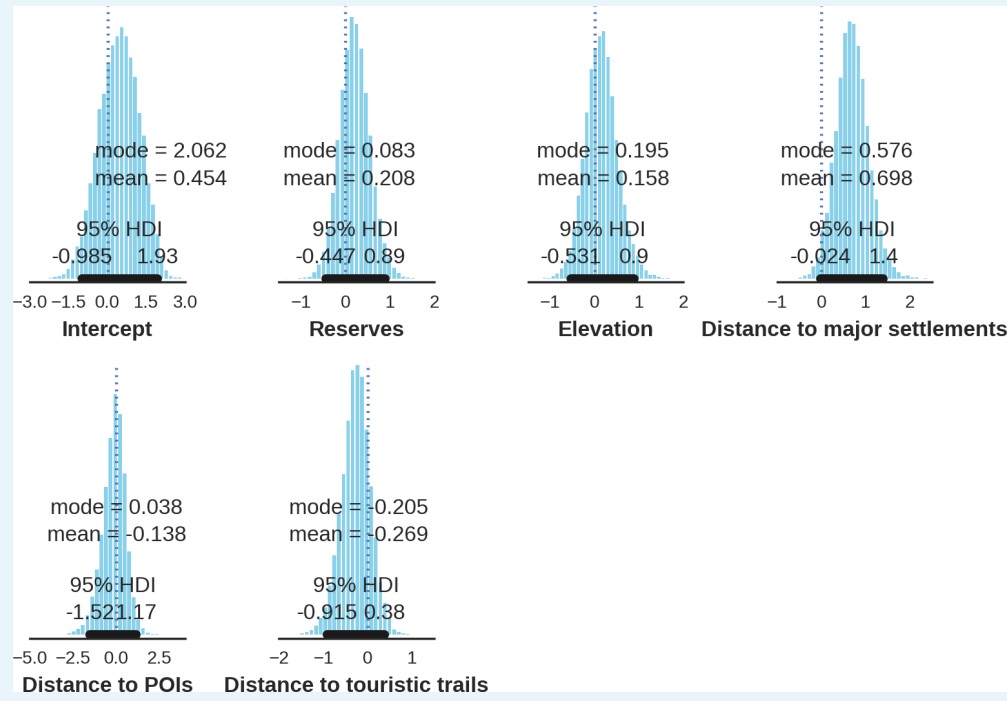

**Appendix 1—Figure 27.** Eurasian lynx – landscape use; the posterior distributions of all parameters.

DOI: https://doi.org/10.7554/eLife.44937.049

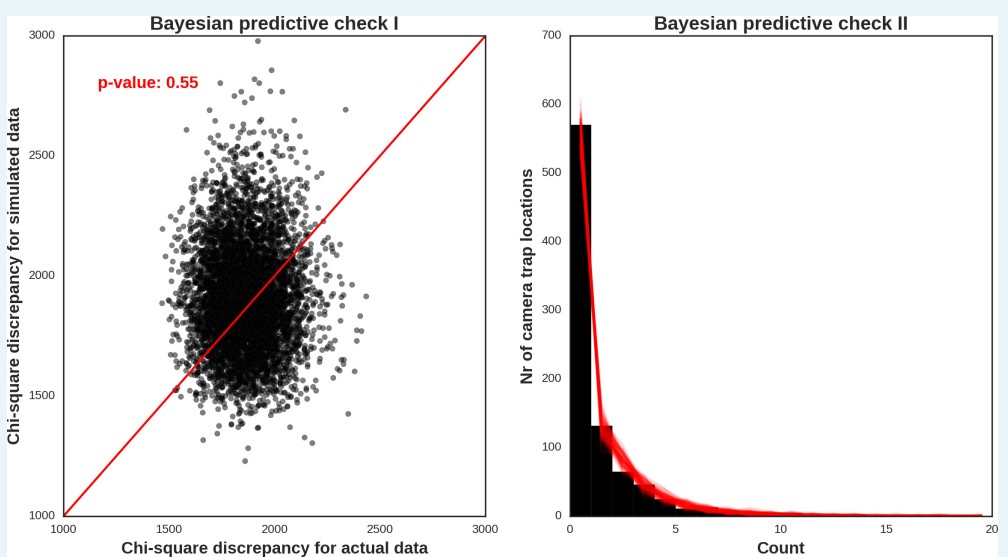

**Appendix 1—Figure 28.** Red deer female – model diagnostic plots.

DOI: https://doi.org/10.7554/eLife.44937.050

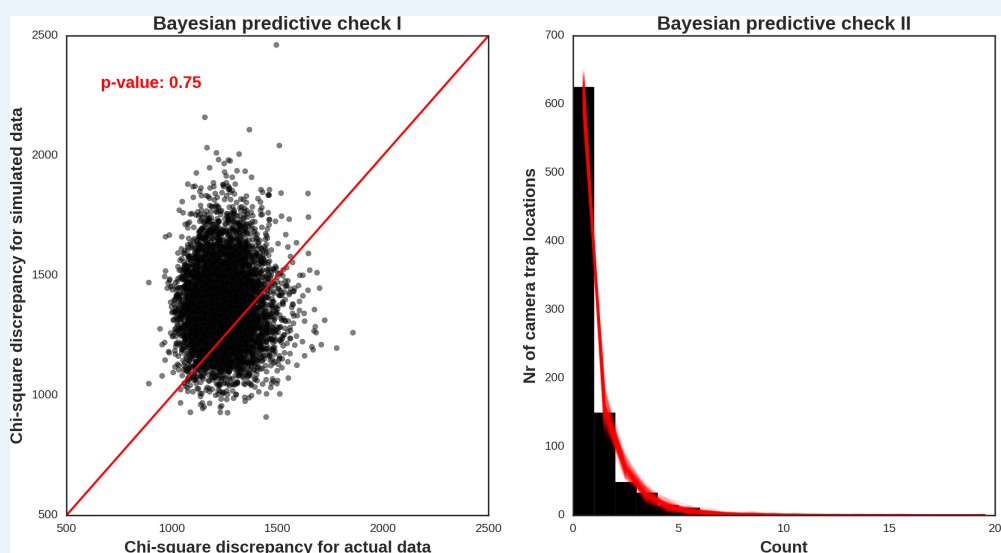

**Appendix 1—Figure 29.** Red deer male – model diagnostic plots.

DOI: https://doi.org/10.7554/eLife.44937.051

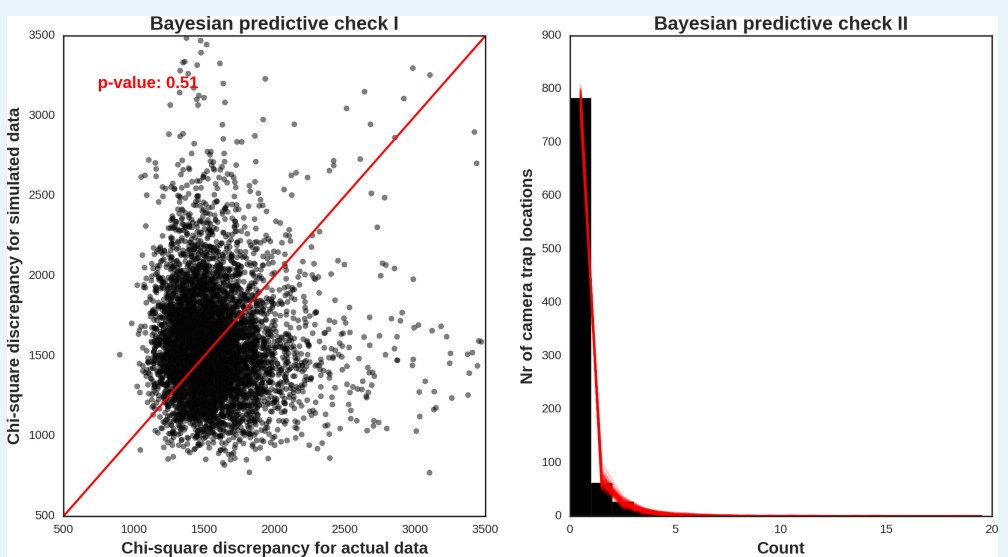

**Appendix 1—Figure 30.** Roe deer – model diagnostic plots.

DOI: https://doi.org/10.7554/eLife.44937.052

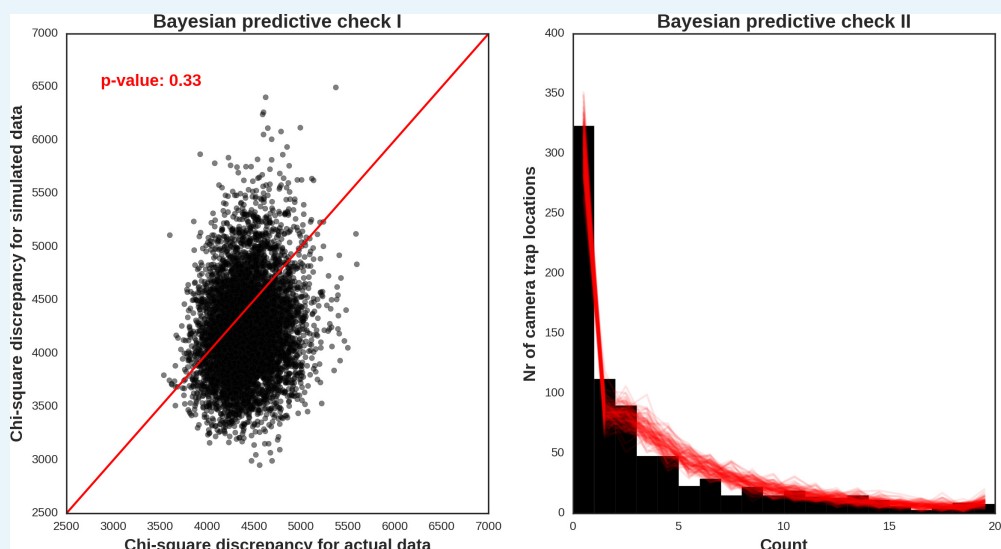

**Appendix 1—Figure 31.** Wild boar – model diagnostic plots.

DOI: https://doi.org/10.7554/eLife.44937.053

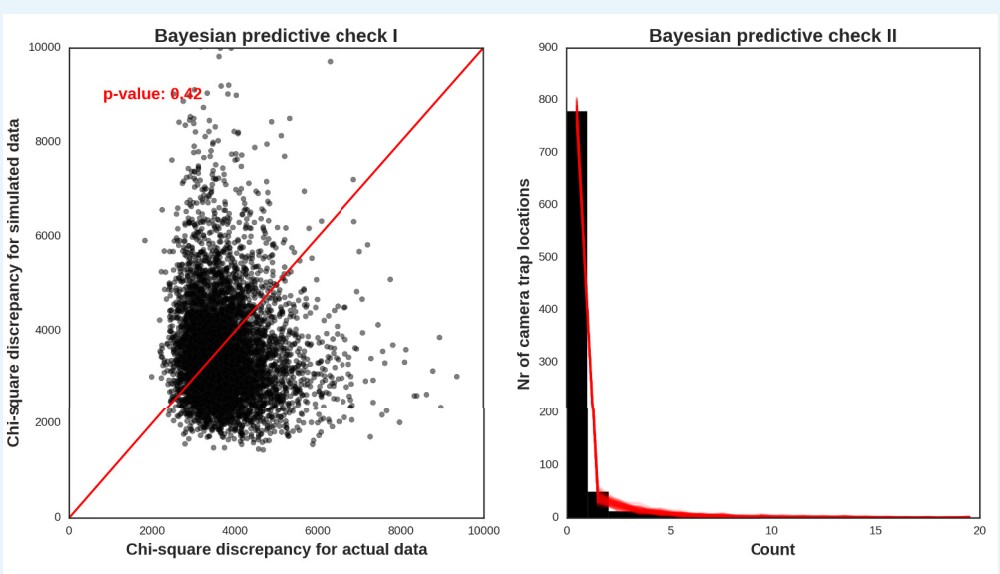

**Appendix 1—Figure 32.** European bison – model diagnostic plots.

DOI: https://doi.org/10.7554/eLife.44937.054

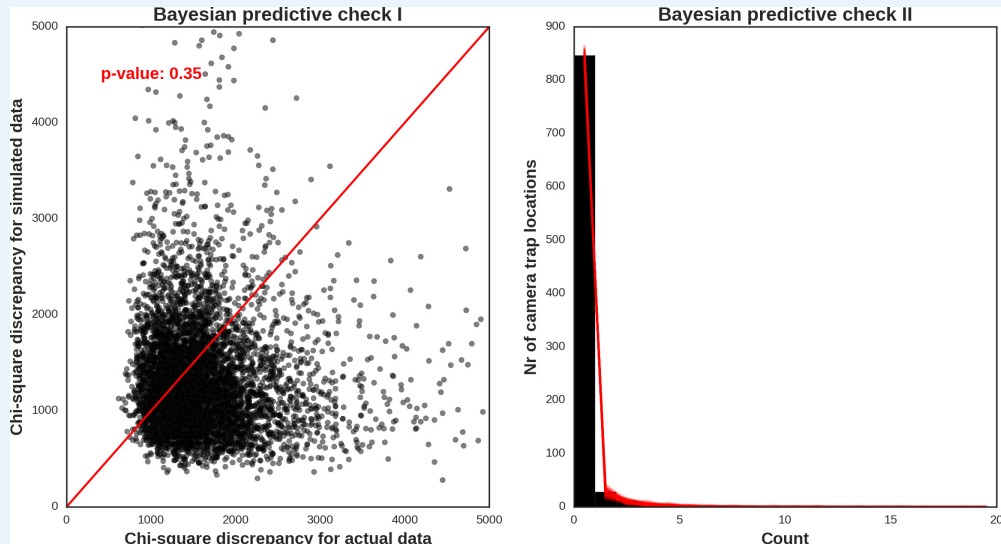

**Appendix 1—Figure 33.** Moose – model diagnostic plots.

DOI: https://doi.org/10.7554/eLife.44937.055

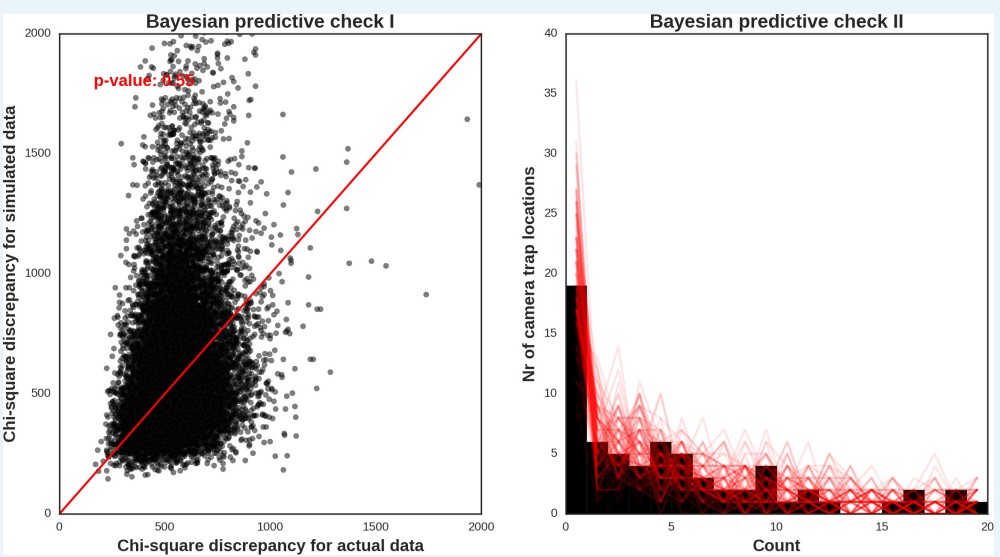

**Appendix 1—Figure 34.** Wolf – model diagnostic plots.

DOI: https://doi.org/10.7554/eLife.44937.056

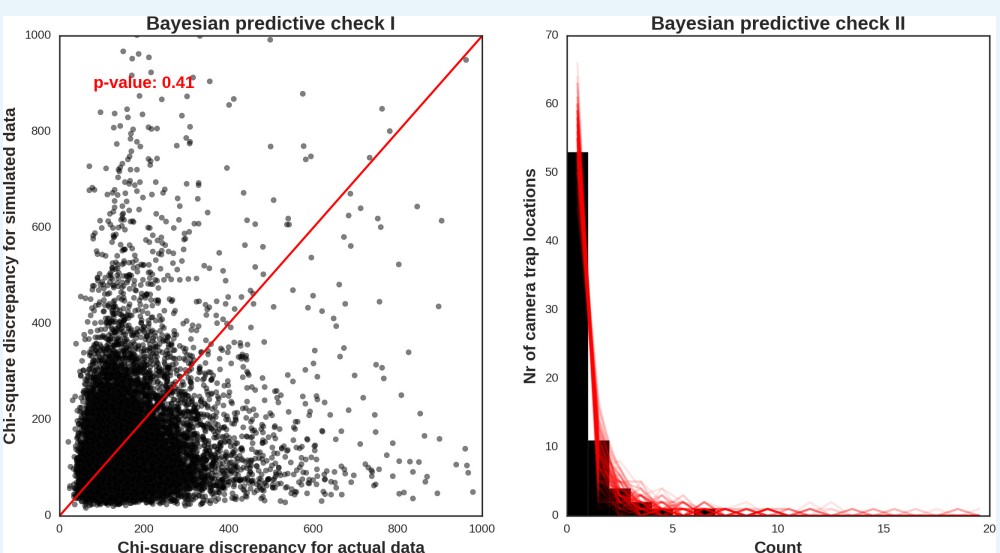

**Appendix 1—Figure 35.** Eurasian lynx – model diagnostic plots.

DOI: https://doi.org/10.7554/eLife.44937.057

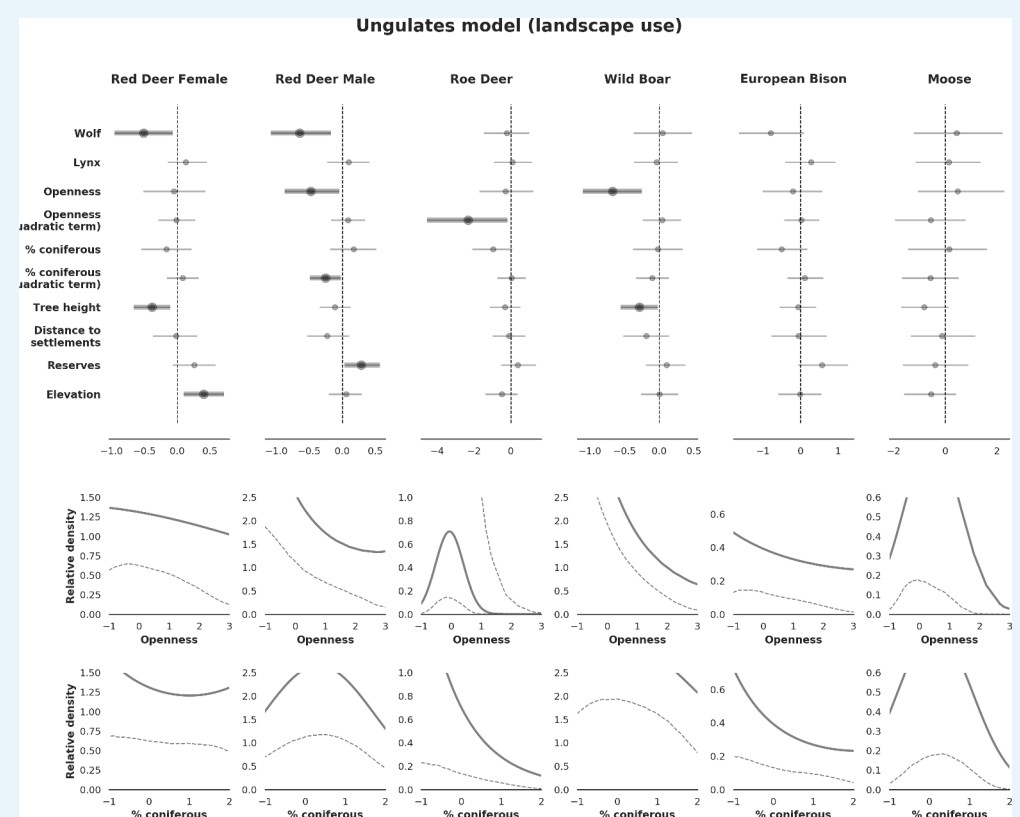

**Appendix 1—Figure 36.** Estimated effects of the covariates from the model for all ungulate species using only a subset of camera trap data covering a three month period (August - October) closely matching the period of the carnivore survey. The covariates explain the spatial variation in the parameter $\lambda$, which is the expected number of individuals using a given landscape grid cell (25 ha pixel) during the sampling period (i.e. the relative density; see the general and formal description of the model in the Materials and methods section).

DOI: https://doi.org/10.7554/eLife.44937.058

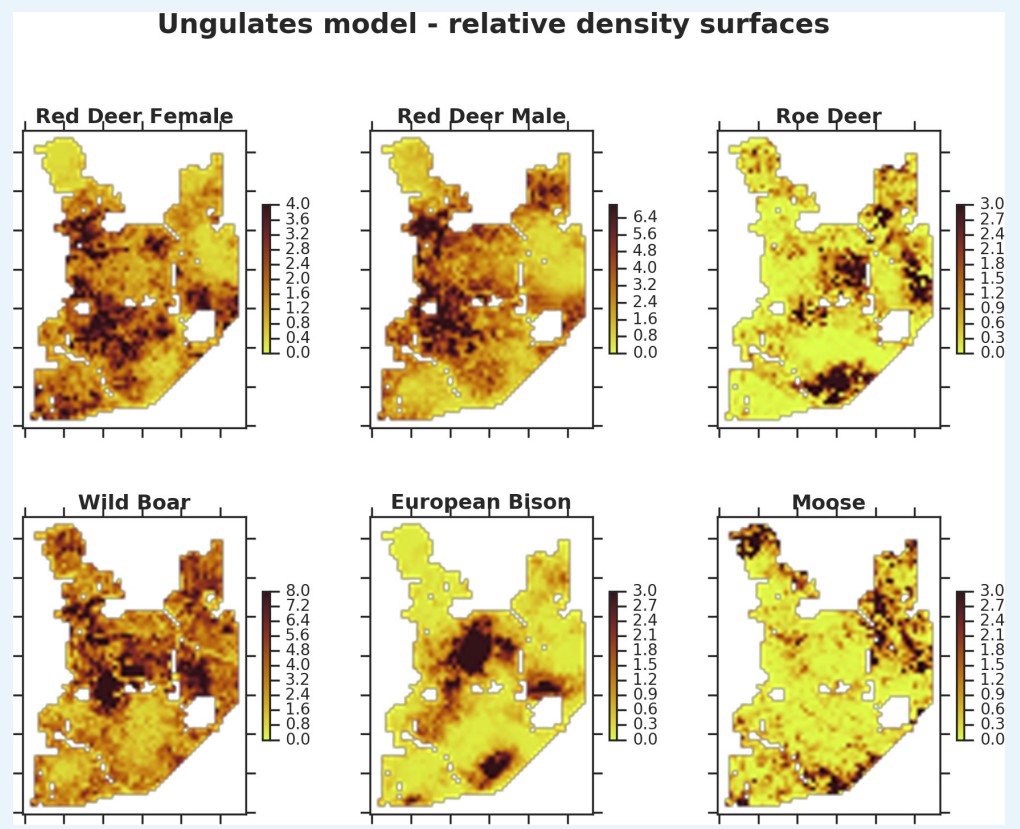

**Appendix 1—Figure 37.** The spatial predictions for the parameter $\lambda$, which is the expected number of individuals using a given landscape grid cell (25 ha pixel) during the sampling period (see the general and formal description of the model in the Materials and methods section). The predictions are based on the model for all ungulate species using only a subset of the camera trap data covering a three month period (August - October) closely matching the period of the carnivore survey.

DOI: https://doi.org/10.7554/eLife.44937.059

