## [Decision Letter]

Thank you for submitting your article "Linking spatial patterns to trophic interactions in terrestrial herbivore communities" for consideration by *eLife*. Your article has been reviewed by three peer reviewers, including Christian Rutz as the Reviewing Editor and Reviewer #1, and the evaluation has been overseen by Ian Baldwin as the Senior Editor. The following individual involved in review of your submission has agreed to reveal their identity: Joern Theuerkauf (Reviewer #3).

The reviewers have discussed the reviews with one another and the Reviewing Editor has drafted this decision to help you prepare a revised submission.

Summary:

This study potentially makes a valuable contribution to our understanding of the processes structuring herbivore communities. While generally a well-studied topic, the study provides rare data from a temperate forest system, identifies functionally-distinct 'herbiscapes' across the landscape, and attempts to measure the influence of human activity. That said, the reviewers have raised a relatively large number of major concerns that need to be carefully addressed in a revision.

Essential revisions:

1) Data coverage. One of the main concerns is that the datasets produced by your camera-trapping approach (as summarised in Appendix 1—table 1) may be too small to enable robust statistical inferences. Counts are relatively small for all but two of the ungulate species (wild boar and red deer), and only few detections were made for the two carnivores (even during the repeat survey). In fact, the data for lynx were so sparse, with just 35 detections, that we had little confidence in the relationships shown in Figure 3D-F, and we actually remained unconvinced that this species should be considered a major predator of the ungulates studied (except, perhaps, for their young, but this would require further justification and analyses). While a better sample was achieved for wolves, with 471 detections, we were concerned that this may still be insufficient to map the species' ranging behaviour. In fact, we were surprised that you had not used data from extensive earlier radio-tracking projects in the study area. To address these concerns, please: (a) remove the lynx from your species set (toning down claims accordingly, about this being a 'multi-predator' system); (b) provide further analyses demonstrating that detection rates for all remaining species were sufficient to allow robust statistical inferences; and (c) explicitly compare your camera-trap data for wolves to earlier radio-tracking data, and assess whether using the latter yields similar conclusions overall.

2) Earlier work. We were surprised to find no mention of a previous study in the Bialowieza Forest, which was very similar in scope (Theuerkauf J and Rouys S 2008, Habitat selection by ungulates in relation to predation risk by wolves and humans in the Bialowieza Forest, Poland. Forest Ecology and Management 256, 1325-1332). Please provide a detailed comparison of your results to those of this earlier study (which had combined pellet counts on transects, with radio-tracking data). It would be reassuring if both studies independently reached similar conclusions, based on different methodology, and if there are disagreements, these should be highlighted and explored further.

3) Camera-trapping methodology. Detection rates are expected to be highly dependent on the placement of camera traps, which can affect inferences about species' habitat use. The area covered by the camera is usually larger in habitats with low ground and scrub cover (not the canopy cover; remote sensing can therefore not detect these differences), which can bias habitat-selection estimates towards areas with high visibility. For example, looking at the distribution map for red deer (Appendix—figure 3), we were surprised that the highest densities were not observed in the strict reserve. Since ground vegetation cover seems higher in the strict reserve than the commercial forest, as noted by one of the reviewers, the camera-trapping method could have biased the distribution of red deer towards the commercial forest. There have been extensive radio-tracking studies on wolves, lynx, bison and red deer in the Bialowieza Forest, and these should be consulted for formal, quantitative comparison.

4) Correlative evidence. It is important to acknowledge that your study draws inferences from correlative evidence, while establishing causality would require experimental manipulation of key parameters. Throughout the manuscript, terms are used that imply causality (to drive, determine, affect etc.), and these should be replaced with appropriate alternatives (to correlate, be associated with etc.) or combined with suitable qualifiers (may, seems, appears to be etc.). Two analyses in particular require more nuanced wording. First, the 'herbiscape' analysis is identified as a primary point of novelty, and while conceptually this is correct, the evidence provided is largely correlative in nature. In the absence of direct experimental evidence of the effects of herbivores on the specific plant species examined (through controlled exclosure experiments), it is impossible to conclude that the spatial distribution of herbivores is causing the observed vegetation patterns. We recognise that such evidence is difficult to obtain, and we are not suggesting that additional work is required, but we think that this issue needs to be acknowledged (e.g., in subsection “Spatial heterogeneity in the landscape distribution of large herbivores” of the Discussion). Second, while it is clear from the data that red deer did not use the same areas as wolves, it is not possible to infer that they 'avoided' those areas (which implies causality). In fact, wolves may only select areas of high red deer density during periods of hunting; the negative correlation between deer and wolf distributions could therefore arise if areas used by wolves when they are not hunting are not preferred habitats of deer. Besides, as red deer are the major prey of wolves in the study area, why would wolves then not select areas of higher red deer density? We appreciate that experimental manipulation of this system is impossible, but note that analyses of spatio-temporal patterns (e.g., from simultaneous radio- or GPS-tracking of wolves and deer) would have considerably strengthened inferences. Please tone down your discussion of wolf-deer relationships accordingly.

5) Presentation. This is a complex study, both in terms of the biological processes studied and the analysis techniques used, and it is critical to help readers follow the narrative. This is especially important as *eLife*'s readership is very broad, and papers should be intelligible to non-specialists. To improve the presentation of the study, we suggest three main revisions. (a) Technical terms should be avoided where possible, or at least clearly defined at the outset. For example, the Introduction talks about "cascading effects", "hyper-keystone species" etc. While these may be frequently-used terms in this research field, without further explanation, many non-specialist readers may struggle to understand the intended meaning. In general, the Introduction could be more focussed and reader-friendly. (b) Since the Results and Discussion sections precede the Materials and methods, some aspects of the study system and methodology remain unclear until much later in the narrative. For example, the species under investigation haven't been mentioned until they are referred to in various parts of the Results section, and 'herbiscapes' are discussed in the Results before the concept is explained in the Discussion. There are two possibilities for dealing with this. Either the Materials and methods are moved forward, for a traditional I-M-R-D structure, or the front end of the narrative is carefully revised, to equip the reader with all the information they need to understand the results and their interpretation. (c) We think it would really help readers if there was a 'graphical abstract' that illustrates, in two separate panels: (i) how data were collected and processed; and (ii) the biological relationships uncovered by the analyses (humans  wolves  prey  vegetation). Panel (ii) could provide explicit call-outs to figures and materials in the online supplement that provide support for a given link, helping readers navigate these various materials.

6) Human activity. Did your camera traps record human activity (on/off established tracks)? If so, why were these data not included in the models, instead of (or in addition to) your chosen proxy variable (distance to settlements)? Appendix—figure 3 shows human activity data, but it is unclear how these were obtained, and "combined" with other datasets, and whether any of this was used in the models. Incidentally, human detections should enable a very useful assessment of the reliability of your camera-trapping method (the distribution of human observations is expected to match the known distribution of human activity in the forest); see also comment #3 above.

7) Sampling periods. The analyses assessing patterns for a subsampled ungulate dataset (for the months covered by the carnivore survey) are very useful. But, it is not clear why an extra month was included here (Aug-Oct; subsection “Carnivores survey”). Why not use Sept-Oct, to make it strictly comparable? More importantly, you say that the "results were similar to those based on the full dataset", yet careful comparison between Figures 4/5 and Appendix 1—figures 36/37 reveals important differences. This should be acknowledged, further investigated, and discussed. Which of your main conclusions remain unchanged when you use this subsampled dataset, and which ones are affected?

8) Broader implications. The study makes a notable contribution to our understanding of key processes in community ecology, but we felt it would be useful to discuss some broader implications in the Conclusions section. Specifically, are any of the findings relevant for ongoing conservation/management efforts, in the study area or elsewhere?

9) Transparent reporting. Some of the points raised in *eLife*'s guidance document require attention (e.g., power analyses, which relates to comment #1 above).

[Editors' note: further revisions were suggested prior to acceptance, as described below.]

Thank you for submitting your article "Linking spatial patterns of terrestrial herbivore community structure to trophic interactions" for consideration by *eLife*. Your article has been reviewed by two peer reviewers, and the evaluation has been overseen by Christian Rutz as the Reviewing Editor and Ian Baldwin as the Senior Editor. The following individuals involved in the review of your submission have agreed to reveal their identity: Joern Theuerkauf (Reviewer #3).

The reviewers have carefully evaluated your revised manuscript and response letter. While reviewer #2 was happy with the revisions overall, apart from a few minor points that still need addressing, reviewer #3 remained concerned about data quality, and specifically, the mismatch in timing of the predator and prey datasets. The reviewers' original reports are appended below for reference. We share reviewer #3's concerns, but are prepared to consider a final revision that addresses these points more explicitly and tones down claims further. It would also be important to better acknowledge throughout that correlations between spatial species distributions were examined, not interactions (as for example implied by the use of the term “cascading effects”).

Reviewer #2:

All of my major and minor comments have been addressed in this revised version of the manuscript. I believe the revision is much-improved and would recommend publication after addressing the few minor comments below.

Minor Comments:

Introduction paragraph two: Unsure of the accuracy of this claim that this subject has often been ignored – "The strong spatial variation in herbivory impact, resulting from the space use of by different functional groups of herbivores has often been ignored." Suggest re-wording to simply state the effect, as it has been shown elsewhere, e.g. "There is often strong spatial variation in herbivory impact resulting from the space use of by different functional groups of herbivores."

Subsection “Spatial heterogeneity in the landscape distribution of large herbivores” paragraph three: Avoid repeated and strong claims of novelty (as this type of study has been done in other systems, as you say). Suggest re-wording to "Similar studies in African ecosystems have shown that…"

In the same subsection: Suggest rewording to focus specifically on the benefits of this study from providing information on a temperate ecosystem – this is the major point of novelty and, in my view, what makes this study a valuable contribution to this body of literature, and should be emphasized. The current wording does not make this clear.

Reviewer #3:

The first concern regarding the manuscript was the robustness of statistical inference. In the rebuttal, the authors assume that if there is no problem with the robustness of the statistical analyses, then the results are reliable. However, the problem does not lie with the analyses but with the quality of data. If the data are not representative in the first place, then there is even no way to verify the robustness of the results. The major problem I see is the temporal mismatch of data sampling. While you sampled ungulate distribution over 2 years, you only sampled predators during a single period of 2 months. Besides, you state that "In August-September, pups begin to travel with other pack members and move more widely through their territory, returning to the core of their territory on a regular basis (Jędrzejewski et al., 2001)". This is however not completely correct. Wolves actually use core areas only during the pup rearing season but not during the rest of the year. Therefore, seasonal data is not representative for the year-round space use. You can therefore compare space use by wolves in September-October only with prey space use during the same season. Otherwise, it is possible you will get random results. Even if prey space use in September-October is representative for the whole year (using the whole year will actually create only the impression of more robust data by artificially increasing sample size, but as the predator sampling is reduced to a short period, it does not help), wolf space use in September-October is definitely not representative for the whole year and therefore also not the best period to quantify space use as you state at the end of the first paragraph in subsection “Carnivores survey”. I would therefore not agree with your statement in the rebuttal regarding seasonal differences in wolf space use: "We appreciate this idea but we did not have data on large carnivores from other seasons. Moreover, we expect little variation in wolf use of the landscape". On the contrary, there are huge differences in wolf space use comparing spring/summer and autumn/winter. This is especially important as September-October also overlap with the rut of red deer, thus the period when wolves rather select hunting male and not female deer (during this period you found that females more correlated with wolves). Given that you did also not consider any temporal information (times of hunting vs times of resting), it is not really possible to assess if the correlations are actually based on real predator-prey interactions or if they are random results. It is possible that the results actually are results of trophic relations, but the data structure does not allow explicit conclusions.

---

## [Author Response]

Essential revisions:1) Data coverage. One of the main concerns is that the datasets produced by your camera-trapping approach (as summarised in Appendix 1—table 1) may be too small to enable robust statistical inferences. Counts are relatively small for all but two of the ungulate species (wild boar and red deer), and only few detections were made for the two carnivores (even during the repeat survey). In fact, the data for lynx were so sparse, with just 35 detections, that we had little confidence in the relationships shown in Figure 3D-F, and we actually remained unconvinced that this species should be considered a major predator of the ungulates studied (except, perhaps, for their young, but this would require further justification and analyses). While a better sample was achieved for wolves, with 471 detections, we were concerned that this may still be insufficient to map the species' ranging behaviour. In fact, we were surprised that you had not used data from extensive earlier radio-tracking projects in the study area.

We appreciate the concerns of the reviewers regarding the robustness of the statistical inferences in relation to the sample size. However, this comment is too broad to address here, as the reviewer had not identified any specific technical problems with our statistical approach (relating to, e.g., a particular part of the methods, the statistical framework used, formal conception of the model).We are convinced that we can defend the structure and specification of our model and its fit to the data. We provided a very detailed description of the applied statistical framework, including a general and formal description of the model, its specification and implementation details together with the data and the source code of our analysis in Python. Our model was built on a well-established statistical framework (Guillera-Arroita et al., 2011, Guillera-Arroita et al., 2012; Hughes and Haran, 2012; Johnson et al., 2013; Kery and Royle, 2016; MacKenzie et al., 2002; Royle et al., 2007; Royle and Nichols, 2003). In the supplementary materials we have provided the standard diagnostic plots for all models of all species (both carnivores and ungulates) which show, without doubt, that our models, including the lynx model, *do* fit to the data (see Appendix 1—figures 28-35). In the supplementary materials an interested reader can also find the figures presenting the posterior distributions for all estimated parameters (for each species; Appendix 1—figures 14-27) as well as track-plots with the MCMC chains used for sampling from the posterior distributions. Therefore, we are not sure about the validity of the reviewer’s criticism on this point.

Species with a low count, such as the lynx, logically have a larger uncertainty in their parameter estimates as expressed by the larger size of the credible intervals and overlap with zero. Importantly, we want to strongly emphasize that in addition to the camera trap records of animals, all the recorded *zeros* are equally important. Our model takes all this information (both zero and non-zero counts) into account. A zero output can result from an ecological process or can be caused by imperfect detection. Our hierarchical model takes both these possibilities into account (see Materials and methods subsection “Formal description”). One model layer explains the variation in the data arising from ecological processes (spatial variation in density of animals), and another layer deals with both ecological (e.g. habitat selection, movement, seasonal activity levels) and observational processes (e.g. detection issues, camera failures). Additionally, we find it worth emphasizing once more that we used data from 894 locations resulting in 9813 trap nights from the ungulate survey, and from 73 locations resulting in 3093 trap nights from the carnivore survey (see Appendix 1—table 1). This should have resulted in robust estimates of animal space use and detection rates; see for example Parsons et al., 2018.

Taking all the above into account, we are convinced that our data and statistical approach allows for a robust statistical inference, and we therefore see no reason how it could be “insufficient to map the species’ ranging behaviour”. Below we present more specific responses to the three points raised by the reviewers.

To address these concerns, please: a) remove the lynx from your species set (toning down claims accordingly, about this being a 'multi-predator' system);

We agree that the sample size for the lynx is particularly small, but we do not see a reason to exclude it from our analysis, firstly because of the technical issues discussed above, and secondly, and even more importantly, for ecological reasons. That the Eurasian lynx *is* a relevant predator in the studied system has been shown by several studies (Jędrzejewska and Jędrzejewski, 1998; Okarma et al., 1997). These studies showed that the lynx’s major prey is the roe deer, but that the red deer also constitutes between 22 and 61% of its diet (Okarma et al. 1997, Jędrzejewski et al., 1993). The annual mortality caused by lynx constitutes 21-36% of the spring numbers of roe deer and 6-13% of the spring number of red deer (Okarma et al., 1997). Lynx predation is therefore a major mortality factor for roe deer and a secondary factor for red deer (Okarma et al., 1997, Jędrzejewski et al., 1993, Jedrzejewska et al., 1997, Jędrzejewska and Jędrzejewski, 2005). Therefore, the evidence fully supports the lynx being an important predator in this system (as we discussed in paragraph six of subsection “The effect of large carnivores on the spatial structure of the large herbivore community”), despite their low numbers. Moreover, our experimental studies have shown that red deer clearly react with anti-predatory behavior to olfactory cues of the lynx in the Białowieża Forest (Wikenros et al., 2015). As the present study shows, these effects on their prey species do not create observable spatial patterns at the landscape scale (as in the case of the wolf). This lack of a clear effect on the landscape use of its prey species likely results from the combined effects of lynx’s low density and low site fidelity (Schmidt et al., 1997. Poland. Acta Theriologica 42:289–312. DOI: 10.4098/AT.arch.97-30) and the previous observation that lynx activity is explained by even finer-scale habitat characteristics than those studied in our camera trapping survey (Podgórski et al., 2008).

We added a new paragraph in the Discussion emphasizing that lynx is a relevant predator in our system and discussing the lack of clear effect on the spatial pattern of landscape use by the prey species of lynx (paragraph six of subsection “The effect of large carnivores on the spatial structure of the large herbivore community”).

b) provide further analyses demonstrating that detection rates for all remaining species were sufficient to allow robust statistical inferences; and

We have no doubts that our statistical inference was robust. Please, see our general response above where we elaborate on this issue. However, if the reviewers are not fully convinced by our arguments we would like to refer them to a recently published paper by Parsons et al., 2018, in *eLife*: “Our sample size goal was 20 spatial replicates (equating to ~420 trap nights), which has been found to maximize precision for estimating detection rate (Kays et al., 2010; Rowcliffe et al., 2008)”. In comparison, in our study we had 894 spatial replicates resulting in 9813 trap nights for the ungulate survey (see Appendix 1—table 1). Therefore, our study is far more data-rich and we feel that the reviewers’ lack of confidence in our study is out of place in this context, especially because no formal part of our analysis was explicitly questioned.

c) explicitly compare your camera-trap data for wolves to earlier radio-tracking data, and assess whether using the latter yields similar conclusions overall.

The existing radio-tracking data from collared wolves in Białowieża Forest was collected over 20 years ago (1994-1999) and since then many things have changed in the forest. For example, the enlargement of Białowieża National Park in 1996, changes in forest management practices, the development of tourism and its infrastructure in the region and last, but not least, a mass dieback of spruce stands in recent years (Boczoń et al., 2018; Mikusiński et al., 2018). As wolves base their habitat selection on human activity, habitat characteristics and prey availability, the changes that have occurred since the end of the 1990s have likely affected wolf space use. Moreover, core areas of wolf territories (i.e. where they reproduce) are not fixed in space and time because female wolves can use different dens each year, and den locations can even change by several kilometers within a season (Schmidt et al., 2008). Therefore, the position of the current wolf core areas could have significantly changed from the telemetry studies observations from the late 1990s (see also Kuijper et al., 2015).However, previous work in Białowieża Forest has showed that human related factors are a key factor determining wolf space use (Theuerkauf et al., 2003) and we know that these factors have been present in a similar spatial arrangement for decades or even centuries (see subsection “Specification - wolf and lynx model” in the Materials and methods section and paragraph seven in subsection “The differential effect of large carnivores on the spatial structure of the large herbivore community” of our Discussion); thus, we expect a general agreement between our camera trapping and the radio-tracking data. To check this, we compiled a new map (Appendix 1—figure 14). In the background we plotted the raster map of wolf space use predicted by our model together with the raw camera trapping data. On top of it we plotted two datasets: 1) the locations of the core areas of the annual territories of four wolf packs based on telemetry studies in the 1990s (Jędrzejewski et al., 2007) and 2) observations of known den locations (period 1993–2007) and wolf howling during the reproductive season (2000–2014) (Nowak et al, 2007 and unpublished data). The map shows that there is a good spatial agreement between the previous telemetry data and the spatial predictions of our model. The only difference is the activity center of the NW wolf pack, which according to the camera trapping data has moved to the north part of Białowieża Forest when compared to the radio-tracking data. This is in line with the most recent (available) locations of dens in this area, which also occurred further north. In summary, our model fitted to the camera trapping dataset (471 wolf detections) is highly reliable as it not only conforms to the general pattern of the wolves’ space use determined by telemetry 20 years ago, but also likely reflects the wolves’ response to ongoing environmental changes in the study area. In other words, using the outdated telemetry data for this study would not be justified.

We compiled a new figure (Appendix 1-figure 4) and added a short paragraph referring to this good agreement in the main text (paragraph two in subsection “Large carnivores”). Moreover, we placed the entire text of this particular response in the appendix as we believe it is a good description of the rationale behind this comparison of our results and the radio-tracking data from ‘90s.

2) Earlier work. We were surprised to find no mention of a previous study in the Bialowieza Forest, which was very similar in scope (Theuerkauf J and Rouys S 2008, Habitat selection by ungulates in relation to predation risk by wolves and humans in the Bialowieza Forest, Poland. Forest Ecology and Management 256, 1325-1332). Please provide a detailed comparison of your results to those of this earlier study (which had combined pellet counts on transects, with radio-tracking data). It would be reassuring if both studies independently reached similar conclusions, based on different methodology, and if there are disagreements, these should be highlighted and explored further.

We have to admit that we have indeed not carefully enough considered this work, mainly due to its completely different methodology. However, this reviewer’s comment has stimulated us to re-examine the paper by Theuerkauf and Rouys, 2008, and compare our research approaches to further validate our results. We have now included a short paragraph in which we refer to this paper and discuss the additional insights that our study has yielded, see lines paragraph five of subsection “The differential effect of large carnivores on the spatial structure of the large herbivore community”. Theuerkauf and Rouys, 2008, generally found that:

1) Habitat alteration by forest exploitation and human hunting influenced the density distribution of ungulates more than predation risk by wolves.

We basically showed similar patterns, i.e. that humans shape the ecological interactions in Białowieża Forest and that these cascade down trophic levels. However, we explored these patterns in more detail, decomposing the inter-specific variation in ungulate space use by using fine-scale, spatially explicit explanatory layers. We were able to detect species specific relationships that included human-predator-ungulate interactions. Moreover, we extrapolated this to the theoretically robust ecological model of herbiscapes.

2) They also showed that (wolf) prey density was not higher in different wolf risk zones, regardless of human hunting.

We showed that the red-deer’s (the main prey of wolf) use of landscape was spatially negatively associated wolf’s high-use areas. As Theuerkauf and Rouys concluded, the “predation risk becomes more complex in multi-predator systems”, our study provided much deeper insight into this complexity showing an interactive, context-dependent character of these relationships.

In general, our results correspond to the findings of Theuerkauf and Rouys, 2008, at least in the comparable parts. However, our study was based on a much finer scale inference and different methodology and therefore there are also differences between the studies. We focused our inference on the variation in ungulate distributions over the landscape, which is a *spatial process* manifesting itself in observable *spatial patterns* emerging from interacting ecological mechanisms operating at a range of spatial scales (Levin, 1992). To gain insight into this process we recorded and reconstructed these spatial patterns by sampling randomly with respect to species and the hypothetical mechanisms driving their distributions (habitat, humans, wolf). We then tested the importance of these mechanisms via a regression-like analysis. We used an intensive large-scale camera trap study to collect detailed, spatially-explicit information on species distribution. Further, we combined these observations with GIS and high-resolution remote sensing data and used a spatially-explicit hierarchical modeling approach to explain the variation in the collected camera trapping data. As a result, our study greatly expand upon the findings of Theuerkauf and Rouys, 2008, study by adding extra resolution both in space and in species level effects. We gained additional insights due to the higher resolution of the camera trap data we used and a (close to) continuous landscape classification based on satellite images and available GIS data.

Moreover, some of the differences between Theuerkauf and Rouys and our results could be attributed to the fact that we were able to account for the contemporary, highly heterogeneous conservation zoning in BF, which has increased significantly since their work. As a result, the areas that were previously classified as “exploited” forest, in reality constitute well preserved habitats that are now excluded from exploitation.

3) Camera-trapping methodology. Detection rates are expected to be highly dependent on the placement of camera traps, which can affect inferences about species' habitat use. The area covered by the camera is usually larger in habitats with low ground and scrub cover (not the canopy cover; remote sensing can therefore not detect these differences), which can bias habitat-selection estimates towards areas with high visibility.

We agree with the reviewer that detection rates depend on many factors, including near-ground visibility in front of a camera. This could potentially have been an issue in the ungulate survey, where camera traps were placed pseudo-randomly within the forest. For this reason, to standardize recording conditions at each camera trap site, we always tried to mount a camera assuring a clear view of at least 20 m, as written in the Materials and methods section. Whenever this was impossible, we randomly looked for an acceptable place to mount a camera as close as possible to the location given by the coordinates pre-computed prior to the field work in QGIS. We have now added this clarification to the Materials and methods section. In our model we specifically took into account factors that could have affected detection probability. As such, our detection rate sub-model was designed to control for other sources of variation in the detectability of animals relating to both ecological (e.g. habitat selection, movement, seasonal activity levels) and observational processes. Please, see the Materials and methods section for more details. Additionally, in our camera trapping dataset, each camera location was tagged with a 4-level categorical label (“Exclude”, “Acceptable”, “Good”, “Perfect”) describing the quality of view in front of the camera. We used this data to prepare a new figure showing that there is no difference between the managed part of BF and BNP (Appendix 1—figure 13).

For example, looking at the distribution map for red deer (Appendix 1—figure 3), we were surprised that the highest densities were not observed in the strict reserve. Since ground vegetation cover seems higher in the strict reserve than the commercial forest, as noted by one of the reviewers, the camera-trapping method could have biased the distribution of red deer towards the commercial forest.

We would like to emphasize once more that the strict reserve and the commercial forest are not homogeneous and discrete forest patches. There is large variation in habitat structure in both areas as illustrated e.g. by our high resolution satellite data (see Figure 2). The commercial forest is also a patchwork of (often extremely) variable forest stands characterized by diverse abiotic conditions (Kwiatkowski, 1994) and forest management history. Despite intensive timber harvest in the past, many forest patches remain natural or of semi-natural character. The large variation in forest canopy cover (see Figure 2) also results in large variation in ground vegetation cover. Consequently, the density of the understory is equal or often greater in certain areas of the commercial forest than in the strict reserve.

Concerning deer densities; the southern part of the Strict Reserve was actually one of the many red deer density hotspots when considering the entire Białowieża Forest (Figure 5, 6, 9). This interesting pattern of higher densities of red deer in the south part of the strict reserve, outside the wolf core area, is in line with our previous studies based on red deer visitation rates and behaviour (Kuijper et al., 2015), and spatial patterns of browsing intensity (Kuijper et al., 2013), which also showed these clear gradients in deer abundance.

There have been extensive radio-tracking studies on wolves, lynx, bison and red deer in the Bialowieza Forest, and these should be consulted for formal, quantitative comparison.

We have noticed that a large part of the reviewers’ critique of our work is related to the quality of our camera trapping data (points #1, #3, #6) and lack of a formal and quantitative comparison of our results to telemetry-based studies (points #1, #3, #4), and so would like to briefly discuss these issues.

Camera trapping, although not a *panacea* for surveying wildlife (Wearn and Glover-Kapfer, 2019), is a well-established, non-invasive method of collecting field data on animal abundance, distribution, temporal activity and space use (Burton et al., 2015; Karanth et al., 2017; Pfeffer et al., 2017; Wearn and Glover-Kapfer, 2017). Camera trapping provides reliable data for population- and community-level inferences by itself and there is no a priori requirement to complement camera trapping with telemetry data, although the latter can provide important auxiliary information e.g. for mark-recapture studies (Dillon and Kelly, 2008; Soisalo and Cavalcanti, 2006).

We fully recognize the value of telemetry, especially GPS-based, which is a powerful method providing rich information on animal movement ecology, activity patterns, resource selection etc. However, it is also not a *panacea* for ecological studies as pointed out by Hebblewhite and Haydon, 2010, in their critical review. One of their conclusions was that one of major disadvantages of GPS-telemetry are the generally small sample sizes and poor population-level inferences. For studies of resource selection, sample size requirements of more than 30 units for robust population-level inferences likely still apply, but still this recommendation does not address representativeness of the sample, which is a function of the total size of the population (Hebblewhite and Haydon, 2010). Therefore, radio-tracking studies from Białowieża forest from 12 collared wolves (Jędrzejewski et al., 2007) out of a wolf population of ~30 (4 packs 6-7 members each) is not the same as data from 19 collared red deer (Kamler et al., 2007) out of a red deer population of ~3000 (5-6 individuals/km2 x 580km^2^) or 29 collared wild boars (Podgórski et al., 2013) out of a wild boar population of ~3000 (5-6 individuals/km^2^ x 580km^2^). Besides this generally low number of collared individuals as a proportion of the entire population (in the case of ungulates), spatial coverage is also essential. The collared individuals of both red deer and wild boar were captured and radio-tracked in a limited area of the Białowieża Forest (at the border of National Park). So the observed patterns from telemetry studies are representative for only a very small part of the population from a particular part of the Białowieża Forest landscape.

The main ecological process we studied, the distribution of five ungulate species over the BF landscape, is a *spatial process*. Our sampling scheme was generally designed to maximize the probability of capturing the overall landscape scale spatial variation in local abundances of each species. We recorded and reconstructed spatial patterns of all the study species by sampling the continuous BF landscape with camera traps placed randomly with respect to species and the hypothetical mechanisms driving their distributions (habitat structure, humans, wolf). That is, we used an intensive, large-scale and high-resolution camera trap network covering the entire study area to collect detailed, spatially-explicit information on species distributions. We argue that this kind of study is impossible to do with telemetry. To record the targeted spatial patterns, observations had to be made at the level of the *entire* population of each species and not at the level of single individuals. No matter how accurately tracked in space and time, data from GPS-collared individuals would not allow the re-constructing of approximated density-surfaces, which are the products of landscape use of the *entire* population of a species. The only exception from this is the radio-tracking data for wolves. In this case the recorded landscape use by collared individuals (c. 50% of all wolves in BF, with each pack represented) is a very good approximation of the landscape use by the entire wolf population. These are the most fundamental reasons why we could not use telemetry and why we cannot formally nor quantitatively compare our results with radio-tracking data from the past, except in the case of wolves (but please see our answer to #1). Finally, our camera trapping and analytical approach, as explained above, enabled us to obtain a reliable picture of wolf spatial patterns, comparable to previous telemetry results, while simultaneously providing high resolution data on all species of the large mammal community with the same unified methodology. We have clarified this in the Materials and methods section.

However, what we can do is a *partial* comparison of our results (i.e. from factors influencing detection rates of single species to patterns observed at the community level) with the previous studies from Białowieża forest, including those based on telemetry. The idea behind this is similar to the philosophy behind pattern-oriented modelling, where multiple patterns observed in complex systems at different hierarchical levels and scales are used systematically to optimize model complexity and reduce uncertainty (Grimm, 2005). We believe we did a good job of this in the results and Discussion sections in the main text. Here we only briefly list some examples of the *partial* results and patterns observed in our data that are directly comparable to the previous studies from Białowieża Forest, and we refer to the figures presenting these results:

1. The use of the landscape by wolves is primarily determined by human related factors, as was shown earlier by Theuerkauf et al. (2003); Figure 3,

2. The distribution of red deer, the main prey of the wolf (Jędrzejewski et al., 2002), negatively correlates with the landscape use of wolves, as was shown by Kuijper et al. (2013, 2015), but contrasts with Theuerkauf and Rouys (2008); Figures 4,5,10

3. The distribution of red deer females was more strongly negatively correlated with wolves than that of red deer males, which is consistent with the selective killing of this sex by wolves in BF (Jędrzejewski et al., 2000; Jędrzejewski et al., 2002); Figure 4

4. The presence of spatial segregation between red deer males and females was in line with Kamler et al. (2008); Figure 8

5. The Strict Reserve was one of the hotspots of ungulate biomass (Jędrzejewska et al., 1997, 1994; Theuerkauf and Rouys, 2008); Figures 5,6,9,10

6. There were higher densities of red deer (especially of females) in the south part of the Strict Reserve, outside the wolf core area (Kuijper et al., 2015, 2013); Figures 5,6

7. There were higher detection rates of red deer and moose in canopy gaps (Churski et al., 2017; Kuijper et al., 2009); Figure 7

8. Wild boar had a preference for closed canopy deciduous dominated tree stands (Jędrzejewska and Jędrzejewski, 1998; Jędrzejewska et al., 1994; Kuijper et al., 2009); Figures 4,5,10

9. Ungulates had a general spatial association with deciduous (wild boar, bison) and mixed deciduous forests (red deer) (Jędrzejewska and Jędrzejewski, 1998; Jędrzejewska et al., 1994); Figures 4,5,10

10. Landscape use by bison was strongly influenced by supplementary winter feeding, the use of open areas inside the forest which are distributed sparsely within the study area and often associated with human settlements (Kowalczyk et al., 2013, 2011); Figures 4,5,6

All these, in addition to the formal methodological tests of our models, contribute to our confidence in the final results. We would like to repeat after (Grimm, 2005): “Useful patterns need not be striking; qualitative or’ weak’ patterns can be powerful in combination.”

4) Correlative evidence. It is important to acknowledge that your study draws inferences from correlative evidence, while establishing causality would require experimental manipulation of key parameters. Throughout the manuscript, terms are used that imply causality (to drive, determine, affect etc.), and these should be replaced with appropriate alternatives (to correlate, be associated with etc.) or combined with suitable qualifiers (may, seems, appears to be etc.). Two analyses in particular require more nuanced wording.

We agree about the general correlative nature of our results. We have carefully checked the manuscript and corrected the text where necessary, see our uploaded manuscript with tracked changes.

First, the 'herbiscape' analysis is identified as a primary point of novelty, and while conceptually this is correct, the evidence provided is largely correlative in nature. In the absence of direct experimental evidence of the effects of herbivores on the specific plant species examined (through controlled exclosure experiments), it is impossible to conclude that the spatial distribution of herbivores is causing the observed vegetation patterns. We recognise that such evidence is difficult to obtain, and we are not suggesting that additional work is required, but we think that this issue needs to be acknowledged (e.g., in subsection “Spatial heterogeneity in the landscape distribution of large herbivores” of the Discussion).

We agree on this point. Although the results of our study are in general correlative and based on non-manipulative observations of the system via remote sensing tools and field measurements, we have built our inference using other studies from Białowieża Forest, including our previous experimental studies on herbivory effects on vegetation (Churski et al., 2017; Kuijper et al., 2010) and quasi-experimental studies on the spatial patterns of ungulates browsing in relation to perceived risk effects (Kuijper et al., 2013). The studies by Kuijper et al., 2010, and Churski et al., 2017, were controlled exclosure experiments that provided the direct experimental evidence of the effects of herbivores on the specific plant species we examined in the current study (i.e. *Acer platanoides* and *Carpinus betulus*). The observed compositional changes in regenerating trees across herbiscapes were in accordance with these earlier experimental studies. Please, see the Discussion section for more details (subsection “Landscape scale herbivory regimes and their potential consequences for vegetation”). We believe that in combination with these previous exclosure studies our results allowed us to (at least) discuss the potential causalities as suggested by the observed patterns in the data and modelled correlations. However, at the same time we do not claim that the fine-scale patterns observed in the exclosure studies automatically imply corresponding causality at the landscape scale. Therefore we toned down our statements in the entire text, see uploaded manuscript with tracked changes.

Second, while it is clear from the data that red deer did not use the same areas as wolves, it is not possible to infer that they 'avoided' those areas (which implies causality). In fact, wolves may only select areas of high red deer density during periods of hunting; the negative correlation between deer and wolf distributions could therefore arise if areas used by wolves when they are not hunting are not preferred habitats of deer. Besides, as red deer are the major prey of wolves in the study area, why would wolves then not select areas of higher red deer density? We appreciate that experimental manipulation of this system is impossible, but note that analyses of spatio-temporal patterns (e.g., from simultaneous radio- or GPS-tracking of wolves and deer) would have considerably strengthened inferences. Please tone down your discussion of wolf-deer relationships accordingly.

We agree that our results are correlative in nature and do not present causality in the deer-wolf relationship. Thus, following the reviewer’s request we toned-downed our discussion accordingly. However, we believe, our camera-trapping system with random distribution and 24-h operation allows the monitoring of wolves both during hunting and non-hunting periods.

Answering the question of the reviewer “why would wolves then not select areas of higher red deer density?” we can cite her/his own answer: “In fact, wolves may only select areas of high red deer density during periods of hunting”. The reported average daily movement distance of wolves (based on telemetry) is 22-27 km (Jędrzejewski et al., 2001) and their average annual territories cover an average of 200 km^2^ (Jędrzejewski et al., 2007), which makes all our indicated red deer hotspots (based on camera trapping) within the daily range and within the territory of the present wolf packs. We would argue that wolves select breeding dens mainly based on human-related factors (in line with Sazatornil et al., 2016. Biological Conservation 201:103–110. DOI: 10.1016/j.biocon.2016.06.022) and then go to hunt in areas rich in deer.

The red deer is present across the entire landscape, although at much lower densities in the wolf high-use areas. In our opinion, it is very unlikely that this pattern is driven by the habitat preference of red deer, as suggested by the reviewer, as the wolf high-use areas also contain high-quality deciduous dominated and mixed deciduous forest stands intensively used by other ungulates (wild boar, bison, moose; Figure 2, Figure 5, also see Kwiatkowski, 1994). Covariates related to habitat quality were included in our models together with relative wolf encounter probability (our proxy for perceived predation risk). Our model results clearly point to the space use of the wolf being the main factor negatively influencing deer distribution rather than habitat factors. Whether these effects result from density- or behaviourally- mediated effects, or a combination of both, is less clear, as we discussed in subsection “The differential effect of large carnivores on the spatial structure of the large herbivore community”.

5) Presentation. This is a complex study, both in terms of the biological processes studied and the analysis techniques used, and it is critical to help readers follow the narrative. This is especially important as eLife's readership is very broad, and papers should be intelligible to non-specialists. To improve the presentation of the study, we suggest three main revisions. a) Technical terms should be avoided where possible, or at least clearly defined at the outset. For example, the Introduction talks about "cascading effects", "hyper-keystone species" etc. While these may be frequently-used terms in this research field, without further explanation, many non-specialist readers may struggle to understand the intended meaning. In general, the Introduction could be more focussed and reader-friendly.

We carefully edited the main text and added all missing definitions of more technical terms. We made sure that the main text is not overloaded with a technical jargon and added all missing definitions of more technical terms.

b) Since the Results and Discussion sections precede the Materials and methods, some aspects of the study system and methodology remain unclear until much later in the narrative. For example, the species under investigation haven't been mentioned until they are referred to in various parts of the Results section, and 'herbiscapes' are discussed in the Results before the concept is explained in the Discussion. There are two possibilities for dealing with this. Either the Materials and methods are moved forward, for a traditional I-M-R-D structure, or the front end of the narrative is carefully revised, to equip the reader with all the information they need to understand the results and their interpretation.

We moved the methods forward as suggested.

c) We think it would really help readers if there was a 'graphical abstract' that illustrates, in two separate panels: (i) how data were collected and processed; and (ii) the biological relationships uncovered by the analyses (humans  wolves  prey  vegetation). Panel (ii) could provide explicit call-outs to figures and materials in the online supplement that provide support for a given link, helping readers navigate these various materials.

We prepared the’ graphical abstract’ as requested. We want to thank the reviewers for this suggestion! We think that it was a great idea and we had a lot of fun working on it together with a specialist in this field and our friend Lisa Sanchez.

6) Human activity. Did your camera traps record human activity (on/off established tracks)? If so, why were these data not included in the models, instead of (or in addition to) your chosen proxy variable (distance to settlements)? Appendix 1—figure 3 shows human activity data, but it is unclear how these were obtained, and "combined" with other datasets, and whether any of this was used in the models. Incidentally, human detections should enable a very useful assessment of the reliability of your camera-trapping method (the distribution of human observations is expected to match the known distribution of human activity in the forest); see also comment #3 above.

Yes, our camera traps recorded humans and we originally intended to use this data as a variable representing the human impact. However, it appeared that the design of both our camera trapping surveys (for ungulates and carnivores) precluded obtaining an unbiased measure of human activity over the BF landscape. A preliminary exploration of the raw data showed a very random pattern of the human record distribution in both surveys, and we considered it unlikely to reliably reflect the true spatial pattern of human interference in BF. In the ungulate survey, the cameras were placed pseudo-randomly in the forest and within a distance of 100 m from the nearest roads (both paved and unpaved), large clearings and settlements, and provided very few human records. In contrast, the carnivore survey, with cameras placed on forest roads, provided more abundant human data, but with records biased heavily by vehicle traffic. The reason for this lies partly in our selection of forest roads as locations for camera traps – we used both very small forest roads not utilized by cars and larger roads used e.g. by forestry transportation machines and border guards, who patrol the area regularly in their vehicles. Also, we did not have cameras monitoring the major (paved) roads in BF that are open for the public and cars (see Figure 1) (i.e. the roads with heavy traffic in the study area).

For these reasons, we decided to express the landscape scale human impact using spatial covariates derived directly from the available GIS data. We believe that the chosen distance-based covariates (distance to all settlements and distance to major settlements) represent human (especially non-motorised) pressure well, as they consider permanent human-made structures and the resulting enduring effect of human interference with wildlife. The distance to major roads was highly correlated with the distance to major settlements (0.87; Appendix 1—figure 1). Another proxy describing the landscape-scale variation in human pressure was the density of protected areas, which are areas of minimal human activity due to exclusion of most forestry practices, hunting and motorized vehicles. We believe that the combination of these variables provided more accurate information on landscape scale and long term human pressure on wildlife populations than our camera trapping surveys, which were not specifically designed for the purpose of collecting this kind of data. As a visualization we included the available data from STRAVA (the global “heatmap” of human activity, https://www.strava.com/heatmap#10.75/23.64564/52.74927/hot/all), see Appendix 1—figure 3. The obtained picture of human activity (people using the STRAVA mobile application) fits well to our covariates describing landscape scale human impact.

7) Sampling periods. The analyses assessing patterns for a subsampled ungulate dataset (for the months covered by the carnivore survey) are very useful. But, it is not clear why an extra month was included here (Aug-Oct; subsection “Carnivores survey”). Why not use Sept-Oct, to make it strictly comparable? More importantly, you say that the "results were similar to those based on the full dataset", yet careful comparison between Figures 4/5 and Appendix 1—figures 36/37 reveals important differences. This should be acknowledged, further investigated, and discussed. Which of your main conclusions remain unchanged when you use this subsampled dataset, and which ones are affected?

The differences between the results based on the full dataset and its subset covering only three months was to be expected. With the full dataset covering all seasons we obtained a picture of the yearly-averaged responses of ungulates to factors affecting their distribution over the landscape. It is to be expected that responses can somewhat differ for some species and factors when we limit the observation window to a short period of only 3 months. Considering the logistics of the entire field work we planned to assure a good, close-to-uniform spatial coverage of the entire study area for the entire study period (2 years). Additionally, we aimed to ensure a similar level of spatial effort for two distinct seasons, i.e. green-on and green-off. Thus, a three-month-only subset of data would not have been representative in terms of the spatial coverage. From a technical point of view, our spatial models are (very) data hungry and we thus chose to use the full dataset, which provided a larger sample size and better spatial coverage of the study area, both of which are needed for making robust inferences using complex hierarchical spatial models such as ours (see Statistical model section). For these technical reasons we included three months in this extra analysis to ensure convergence of the models. When limiting ourselves to only two months, we had too little data to fit the models, which did not converge.

In our opinion, this extra analysis is not critical for confidence in our analysis and results (see our responses to points #1 and #3), especially when taking into account the fact that our model of wolf space use matches well the telemetry and den location data from the past (see our response to point #1). All this indicates the high level of stability in the way that wolves use the Białowieża Forest landscape.

8) Broader implications. The study makes a notable contribution to our understanding of key processes in community ecology, but we felt it would be useful to discuss some broader implications in the Conclusions section. Specifically, are any of the findings relevant for ongoing conservation/management efforts, in the study area or elsewhere?

Following this suggestion we have added a new paragraph to the Discussion. Please see paragraph six of subsection “Landscape scale herbivory regimes and their potential consequences for vegetation”.

9) Transparent reporting. Some of the points raised in eLife's guidance document require attention (e.g., power analyses, which relates to comment #1 above).

Please see our response to point #1. Also, we are clear and transparent about this issue in the submitted transparent reporting form:

“No explicit power analysis was used prior to our field data collection using camera traps. As our study was of an observational, rather than experimental nature, and as we investigated complex spatial interactions between many species representing multiple trophic levels, we were unable (to our knowledge) to run an explicit power analysis. However, in our methods we ensured a large enough number of sampled locations to successfully fit our complex spatial models (diagnostics are presented in Appendix 1). Our sampling methodology, together with a short discussion of all the arbitrary decisions we had to take, are described in detail in the main text in the Materials and methods section.”

We would also like to refer to the work by Parsons et al., 2017, recently published in *eLife*: “No explicit power analysis was used to predetermine sample size. Our sample size goal was 20 spatial replicates (equating to ~420 trap nights), which has been found to maximize precision for estimating detection rate (Kays et al., 2010; Rowcliffe et al., 2008). Camera sites are biological replicates, parallel measurements capturing random biological variation. This study did not include technical replicates.”

[Editors' note: further revisions were suggested prior to acceptance, as described below.]

The reviewers have carefully evaluated your revised manuscript and response letter. While reviewer #2 was happy with the revisions overall, apart from a few minor points that still need addressing, reviewer #3 remained concerned about data quality, and specifically, the mismatch in timing of the predator and prey datasets. The reviewers' original reports are appended below for reference. We share reviewer #3's concerns, but are prepared to consider a final revision that addresses these points more explicitly and tones down claims further. It would also be important to better acknowledge throughout that correlations between spatial species distributions were examined, not interactions (as for example implied by the use of the term “cascading effects”).

Reviewer #2:

All of my major and minor comments have been addressed in this revised version of the manuscript. I believe the revision is much-improved and would recommend publication after addressing the few minor comments below.Minor Comments:Introduction paragraph two: Unsure of the accuracy of this claim that this subject has often been ignored – "The strong spatial variation in herbivory impact, resulting from the space use of by different functional groups of herbivores has often been ignored." Suggest re-wording to simply state the effect, as it has been shown elsewhere, e.g. "There is often strong spatial variation in herbivory impact resulting from the space use of by different functional groups of herbivores."

We have changed this sentence according to the reviewer's suggestion.

Subsection “Spatial heterogeneity in the landscape distribution of large herbivores” paragraph three: Avoid repeated and strong claims of novelty (as this type of study has been done in other systems, as you say). Suggest re-wording to "Similar studies in African ecosystems have shown that…"

We have changed this sentence according to the reviewer's suggestion.

In the same subsection: Suggest rewording to focus specifically on the benefits of this study from providing information on a temperate ecosystem – this is the major point of novelty and, in my view, what makes this study a valuable contribution to this body of literature, and should be emphasized. The current wording does not make this clear.

We have reworded this sentence according to the reviewer's suggestion:

“Our contribution to all the above studies is a high-resolution picture of the spatial structure of an entire community of large herbivores that incorporates both bottom-up and predation- and human-related top-down factors. We believe one of our major points of novelty is in providing information on these spatial interactions for a temperate ecosystem.”

Reviewer #3:

The first concern regarding the manuscript was the robustness of statistical inference. In the rebuttal, the authors assume that if there is no problem with the robustness of the statistical analyses, then the results are reliable. However, the problem does not lie with the analyses but with the quality of data. If the data are not representative in the first place, then there is even no way to verify the robustness of the results. The major problem I see is the temporal mismatch of data sampling. While you sampled ungulate distribution over 2 years, you only sampled predators during a single period of 2 months. Besides, you state that "In August-September, pups begin to travel with other pack members and move more widely through their territory, returning to the core of their territory on a regular basis (Jędrzejewski et al., 2001)". This is however not completely correct. Wolves actually use core areas only during the pup rearing season but not during the rest of the year. Therefore, seasonal data is not representative for the year-round space use. You can therefore compare space use by wolves in September-October only with prey space use during the same season. Otherwise, it is possible you will get random results. Even if prey space use in September-October is representative for the whole year (using the whole year will actually create only the impression of more robust data by artificially increasing sample size, but as the predator sampling is reduced to a short period, it does not help), wolf space use in September-October is definitely not representative for the whole year and therefore also not the best period to quantify space use as you state at the end of the first paragraph in subsection “Carnivores survey”. I would therefore not agree with your statement in the rebuttal regarding seasonal differences in wolf space use: "We appreciate this idea but we did not have data on large carnivores from other seasons. Moreover, we expect little variation in wolf use of the landscape". On the contrary, there are huge differences in wolf space use comparing spring/summer and autumn/winter. This is especially important as September-October also overlap with the rut of red deer, thus the period when wolves rather select hunting male and not female deer (during this period you found that females more correlated with wolves). Given that you did also not consider any temporal information (times of hunting vs times of resting), it is not really possible to assess if the correlations are actually based on real predator-prey interactions or if they are random results. It is possible that the results actually are results of trophic relations, but the data structure does not allow explicit conclusions.

The critique of reviewer #3 essentially comes down to the question of whether the wolf data collected by our camera-traps in September-October is representative of the year-round and landscape-scale space use of wolves. The reviewer states that it is not, and that the temporal mismatch of camera trap-based sampling of large carnivores (in autumn) and ungulates (year-round) could ultimately lead to “random results”. The reviewer specifically argues that “wolves actually use core areas only during the pup rearing season but not during the rest of the year”. Below we argue that our assumption is defendable and that our camera-trapping carnivore-data can be regarded representative:

1) Firstly, the reviewer’s statement that wolves use core areas only during the pup rearing season but not during the rest of the year is contrary to extensive earlier studies from the study area. The year-round, rotational use of core areas by wolves, including outside the pup-rearing period, was shown by Jędrzejewski et al., 2001, on the basis of extensive radio-telemetry work (Figures 2 and 3). The presence of clearly defined core areas during bimonthly intervals throughout the year was also shown by Jędrzejewski et al., 2004, in their detailed study on the process of wolf pack splitting. Both Jędrzejewski et al., 2007, and Schmidt et al., 2009, showed that on a yearly basis core areas are the territory areas used most intensively by wolves. Furthermore, Zub et al., 2003 also showed that in winter (study period from November till April) the density of territorial markings is concentrated in core areas due to more intensive use of core areas by packs. Finally, the study of Nowak et al., 2007, found that spontaneous howling activity in summer and late autumn is strictly connected to the core areas of their territories, where the breeding den had been located, and not from the peripheries. In summary, the existing literature clearly shows that core areas are the most intensively used parts of the annual wolf territory throughout the year and many wolf activities additional to breeding and pup-rearing concentrate in this area.

That wolves continue to use the core area of their annual wolf territory also after pup rearing, although with different intensity, was also shown by Theuerkauf et al., 2003. The core areas are highly selected in both summer and winter (see Figure 2 in Theuerkauf et al., 2003). We generally agree with the reviewer that there are seasonal changes whereby the summer season (including the pup-rearing period) is characterized by a stronger concentration in the core areas. However, these seasonal changes in territory use do not change the overall pattern of concentrated use in some parts of their territories.

2) The seasonal changes in wolf movements are most clearly marked in adult breeding and sub-adult females. Indeed, during the pup-rearing period their movements are restricted to the vicinity of the den. However, other members of the pack (adult males, adult non-breeding females, subadult males) do not show such “huge differences”. Please also note, that camera traps collect data on wolf presence and activity randomly, irrespective of the sex or breeding status. The period from September to October was chosen to monitor the activity of carnivores, considering that both wolves (of all reproductive status) and lynx then resume travelling long distances within their home ranges after the breeding season. According to previous telemetry studies conducted on both predators in the study area wolves covered the largest distances and ranges during this period as compared to other months (Jędrzejewski et al., 2001), and lynx (all except breeding females) were equally mobile year-round (Jędrzejewski et al., 2002). We thus believe that as we were unable to conduct a year-round survey of carnivores, September – October is the most suitable period for sampling the activity of carnivores at the scale of the whole forest complex as it reflects the general annual activity of large carnivores most reliably.

3) In our previous response to this particular critique, we already showed that the non-uniform space use of wolves as predicted by our camera trap study, is in close correspondence with earlier radio-telemetry studies (showing the year-round space use of wolves) and with the distribution of known locations of breeding dens (see Appendix 1—figure 4). These independent data sources all show that the landscape use of wolves is neither uniform nor random and core areas (with a higher intensity of use) are clearly “visible” at the landscape scale. Moreover, the core areas and spatial arrangements of territories of the four wolf packs in Białowieża forest (BF) have been relatively stable over the last decades. We elaborated on this in the main text (subsection “Specification - ungulates model” in the Materials and methods and subsection “Landscape scale herbivory regimes and their potential consequences for vegetation” in the Discussion) and in our previous response letter (points #1c).

4) We showed in our previous response letter that restricting our ungulate dataset to only part of the year, to create a better temporal match between wolf and ungulate data, does not change the model outputs and conclusions of our study related to large carnivores (see Appendix 1—figures 36 and 37).

Conclusions:

We do not disagree with the reviewer that wolves show seasonal changes in their territory use, but the arguments provided above illustrate that there is no rationale for expecting these changes to undermine the patterns we found in the study. Based on existing knowledge and our data, we can expect continuous higher relative use of core areas than of other parts of their territories. Thus the spatial patterns of wolf landscape use observed in autumn, when they clearly concentrate in their core areas but also widely use their entire territories, gives a representative, ‘averaged’ picture of the relative use of the BF landscape throughout the year. Therefore, we feel confident that our collected wolf data is representative for showing year-round patterns of wolf space use, despite the seasonal variation in territory use over the year (see for example Theuerkauf et al., 2003) referred to by the reviewer.

Our analyses show that these core areas have an effect on the space use of red deer. This does not exclude the possibility that additional finer-scale and more direct interactions occur between wolves and their prey during instances of actual wolf hunting activity. Much of the hunting activity takes place outside the wolf core areas and all red deer hot spots predicted by our models are within the daily range of each wolf pack in BF. We elaborate on this topic in our previous response to the reviewers (see point #4b). However, our data suggest that the continuous presence and higher relative use of wolf core areas has a landscape-level effect on the distribution of their main prey species (red deer) on a yearly basis.

If the reviewer continues to disagree with the arguments we have provided, we feel it may be more appropriate to continue this discussion on the basis of a scientific comment to our study that we would be happy to reply to.

References:

Jędrzejewski W, Schmidt K, Theuerkauf J, Jędrzejewska B, Kowalczyk R. 2007. Territory size of wolves Canis lupus: Linking local (Bia?Owie?A Primeval Forest, Poland) and Holarctic-scale patterns. *Ecography***30**:66–76. doi:10.1111/j.2006.0906-7590.04826.x

Nowak S, Jędrzejewski W, Schmidt K, Theuerkauf J, Mysłajek RW, Jędrzejewska B. 2007. Howling activity of free-ranging wolves (canis lupus) in the białowieża primeval forest and the western beskidy mountains (poland). *Journal of Ethology***25**:231–237. doi:10.1007/s10164-006-0015-y

Zub K, Theuerkauf J, Jędrzejewski W, Jędrzejewska B, Schmidt K, Kowalczyk R. 2003. Wolf pack territory marking in the Bialowieza Primeval Forest (Poland). *Behaviour***140**:635–648. doi:10.1163/156853903322149478